# Global genetic diversity, introgression, and evolutionary adaptation of indicine cattle revealed by whole genome sequencing

Indicine cattle, also referred to as zebu (*Bos taurus indicus*), play a central role in pastoral communities across a wide range of agro-ecosystems, from extremely hot semiarid regions to hot humid tropical regions. However, their adaptive genetic changes following their dispersal into East Asia from the Indian subcontinent have remained poorly documented. Here, we characterize their global genetic diversity using high-quality whole-genome sequencing data from 354 indicine cattle of 57 breeds/populations, including major indicine phylogeographic groups worldwide. We reveal their probable migration into East Asia was along a coastal route rather than inland routes and we detected introgression from other bovine species. Genomic regions carrying morphology-, immune-, and heat-tolerance-related genes underwent divergent selection according to Asian agro-ecologies. We identify distinct sets of loci that contain promising candidate variants for adaptation to hot semi-arid and hot humid tropical ecosystems. Our results indicate that the rapid and successful adaptation of East Asian indicine cattle to hot humid environments was promoted by localized introgression from banteng and/or gaur. Our findings provide insights into the history and environmental adaptation of indicine cattle.

The domestication of aurochs (*Bos primigenius*) gave rise to two distinct but cross-fertile cattle subspecies, humpless taurine cattle (*B. taurus taurus*) and humped indicine or zebu cattle (*B. t. indicus*)[1]. These events were important in human history, with extensive implications for the diet, culture, and socioeconomic structure of the farming populations across the Old World. Taurine cattle were domesticated ~10,000 years before present (YBP), followed by the domestication of indicine cattle 2000 years later in the Indus Valley of modern Pakistan[2]. Indicine cattle are recognized by their thoracic hump, low metabolic rate, many large sweat glands, large skin surface, and short smooth coat[3]. They are often resilient to local ticks and capable of tolerating the hot and/or humid climates of the semiarid and tropical regions[3]. Thus, they can experience a much larger complementary thermal stress spectrum than taurine cattle, which are notably absent in the tropical areas of Asia. The latter are indeed largely confined to temperate to cold environments, with the exception of the West African

taurine cattle living in humid and subhumid, tsetse fly-infested, and tropical environments[4].

Indicine cattle are the most abundant and important livestock species in South Asia, East Asia, and Africa[5], and they represent more than half of all cattle populations worldwide[3]. The successful global and agro-ecological dispersal of indicine cattle is unique among domestic bovine species. It has been essential for the development of local agricultural lifestyle and economy that have shaped modern societies in subtropical and tropical regions[3,4]. Their adaptation to the hot climate will be increasingly important in the context of climatic changes, with increasing temperatures affecting livestock production worldwide[4].

Archaeological evidence indicates the presence of domesticated indicine cattle earlier in the Indus Valley (~8000 YBP) than in South India (~5000 YBP) and the middle Ganges (~4000 YBP)[5,6]. The global dispersal of indicine cattle started in the Indus Valley at ~5000 YBP,

✉e-mail: o.hanotte@cgiar.org; h.jianlin@cgiar.org; yu.jiang@nwafu.edu.cn; leichuzhao1118@nwafu.edu.cn

followed by their spread into Southwest and Central Asia, East Asia, and Africa between 4000 and 1300 YBP[5, 6]. An ancient DNA analysis indicated widespread male-mediated introgression of indicine cattle from the Indus Valley into the Near East from 4200 YBP. Modern DNA analyses have now well documented this male-mediated indicine admixture into African taurine cattle in the eastern, western, and southern areas of the continent[5]. There is also small but significant indicine introgression into almost all southeastern European cattle breeds[7]. The expansion of indicine cattle has continued until recent times, and indicine cattle imported in the nineteenth and twentieth centuries into America and Australia have formed large local populations[8]. Along with their global spread, admixture with local taurine cattle, wild and/or domesticated banteng and gaur, and possibly other unsampled wild bovine species supposedly led to the diversification of indicine cattle populations[5,9,10]. A common practice is to hybridize other bovine species with cattle to rapidly improve their adaptation to new environments. The establishment of stable hybrid populations is difficult because hybrid males are often sterile, but limited introgression after backcrossing several generations of female hybrids to male cattle is possible[11].

Accordingly, three major domestic indicine autosomal lineages are recognized today: (1) the source population in South Asia; (2) African indicine cattle admixed with African taurine diversity[5]; and (3) East Asian indicine cattle[12]. Global indicine diversity is further characterized by two Y chromosome haplogroups (Y3A and Y3B)[9], two major indicine mtDNA haplogroups (I1 and I2) in Asia[6], taurine mtDNA haplogroups in African, American, and Australian indicine cattle populations, and banteng mtDNA in several Indonesian indicine breeds[13]. Taken together, current autosomal, Y-chromosomal, and mitochondrial ancestries indicate complex domestication and evolutionary processes in the formation of global indicine cattle diversity.

The aim of this study was to explore the unique genomic characteristics and phylogeographic patterns of the diversity of indicine cattle using the largest indicine cattle genome dataset available to date. We present a comprehensive genomic analysis of the variations in the Y chromosome, mitogenomes, and whole nuclear genomes of 354 indigenous indicine cattle sampled from 57 breeds/populations representing the majority of indicine cattle groups. Our findings reveal a discontinuous geographic pattern of genomic diversity and extensive introgression of banteng and gaur, and provide insights into the genomic background of the unusual physiological features that enable indicine cattle to tolerate extreme environments (hot-dry and hot-humid) and a high infectious disease burden.

## Results

### Genetic diversity and differentiation of indicine cattle

A total of 297 new genomes, including 287 indicine cattle representing 42 breeds/populations and 10 taurine cattle representing three breeds, were sequenced to an average depth of 11.72×. They were combined with 198 (67 indicine and 131 taurine genomes) publicly available genomes (Fig. 1a, Supplementary Note 1, and Supplementary Data 1). Twenty-two whole genomes from other wild and/or domestic bovine species (five gaur, eight banteng, two bison, two wisent, three yak, and two swamp buffaloes) were included for introgression analysis. Sequence reads were aligned to the taurine cattle reference genome (ARS-UCD1.2) and Btau 5.0 Y chromosome with an average alignment rate of 99.50% and a coverage of the reference genome of 94.76% (Supplementary Data 1). A total of 354 indicine genomes representing 57 breeds/populations and 141 taurine genomes from 17 breeds/populations were classified as follows: African taurine (AFT, $n = 19$), European taurine (EUT, $n = 62$), Eurasian taurine (EAT, $n = 28$), Tibetan taurine (TBT, $n = 8$), Northeast Asian taurine (NEAT, $n = 24$), African indicine (AFI, $n = 111$), South Asian indicine (SAI, $n = 118$), Southeast Asian indicine (SEAI, $n = 28$), Tibetan indicine (TBI, $n = 7$), Southwest Chinese indicine (SWCI, $n = 4$), East Asian indicine (EAI, $n = 80$), and American indicine (AMI, $n = 6$) cattle (Fig. 1 and Supplementary Data 1). A total of 67,162,108 autosomal SNPs were identified (Supplementary Tables 1 and 2).

The genome-wide nucleotide diversity revealed by autosomal SNPs was generally higher in indicine cattle (0.00261–0.00337) than in taurine cattle (0.00136–0.00164). The highest value (0.00337) was observed within the EAI cattle (Supplementary Fig. 1 and Supplementary Note 2), while the average values for AFI and SAI cattle were 0.00265 and 0.00261, respectively. The level of inbreeding measured by runs of homozygosity (ROH) was lower in indicine cattle than in taurine cattle (Supplementary Fig. 2). Genetic distances estimated via the pairwise fixation index ($F_{ST}$) ranged from 0 to 0.205 between indicine breeds/populations and from 0.201 to 0.550 between indicine and taurine groups (Supplementary Fig. 3).

Principal component analysis (PCA) of the autosomal SNPs revealed clear phylogeographical differentiation, with PC1 corresponding to the contrast between indicine and taurine cattle (Fig. 1b). The PCA and phylogenetic tree almost completely separated the three indicine geographic groups of SAI, AFI, and EAI cattle (Fig. 1c and Supplementary Figs. 4–6). SWCI fell in genetically intermediate positions between SEAI and EAI cattle. The indicine cattle of TBI (Tibet, China) and Nepal were close to SAI cattle (Fig. 1c). The ADMIXTURE analysis recapitulated a similar pattern and identified three indicine and three taurine phylogeographic groups at $K = 6$ (Fig. 1d, Supplementary Fig. 7, and Supplementary Table 3). The same differential topology was observed in the maximum likelihood (ML) tree of these breeds/populations (Supplementary Fig. 8).

### Adaptation of indicine cattle

Outside the monsoon season, the Indus Valley has a semiarid climate with a high temperature, high solar radiation, and low rain fall[14, 15]. Because it is the center of origin of indicine cattle, these cattle may be expected to be particularly adapted to such environmental conditions. This has driven the successful spread of indicine cattle into the central and southern regions of the globe. To identify these ancestral adaptations at the genome level, we combined SAI, AFI, and EAI populations for a comparison with taurine cattle by using $F_{ST}$, $\theta_\pi$ ratio, and cross-population extended haplotype homozygosity (XP-EHH) approaches (Table 1, Supplementary Note 3, Supplementary Table 4, and Supplementary Figs. 9 and 10). A total of 156 nonoverlapping windows of 50 kilobase (kb) in size were detected using all three approaches. They overlapped with 117 candidate genes (Supplementary Table 4).

The top selection signatures were in two regions on *Bos taurus* autosome (BTA)7, together spanning 4.46 megabases (Mb) (43.04–44.67 and 50.14–52.97 Mb) (Supplementary Fig. 11). This region was previously identified to be associated with host immunity, environmental thermal stresses, and reproduction in African humped cattle[5] (Table 1), supporting its ancestral indicine origin. Another strong selection signature for a gene-rich region was located on BTA19, spanning 1 Mb and covering genes related to antiviral immunity (*SPAG7*[16]), neurodegenerative disease (*KIF1C*[17]), skeletal development (*PFN1*[18]), cardiac growth (*CAMTA2*[19]), and muscle development and glycogen storage (*ENO3*[20]) (Supplementary Fig. 12). We also identified a functional gene, *LIPH*, on BTA1 (81.58–81.69 Mb) (Supplementary Fig. 13), which was linked to hair growth deficiency in humans[21], implying its potential contribution to the heat tolerance of indicine cattle via the control of coat hair length and/or thickness. Seven of the remaining 75 genes in the topmost significant sweep regions are functionally associated with heart development, blood circulation, DNA damage, and light response (Table 1 and Supplementary Table 4). However, further research is warranted to test their roles in heat adaptation or other differences between indicine and taurine cattle.

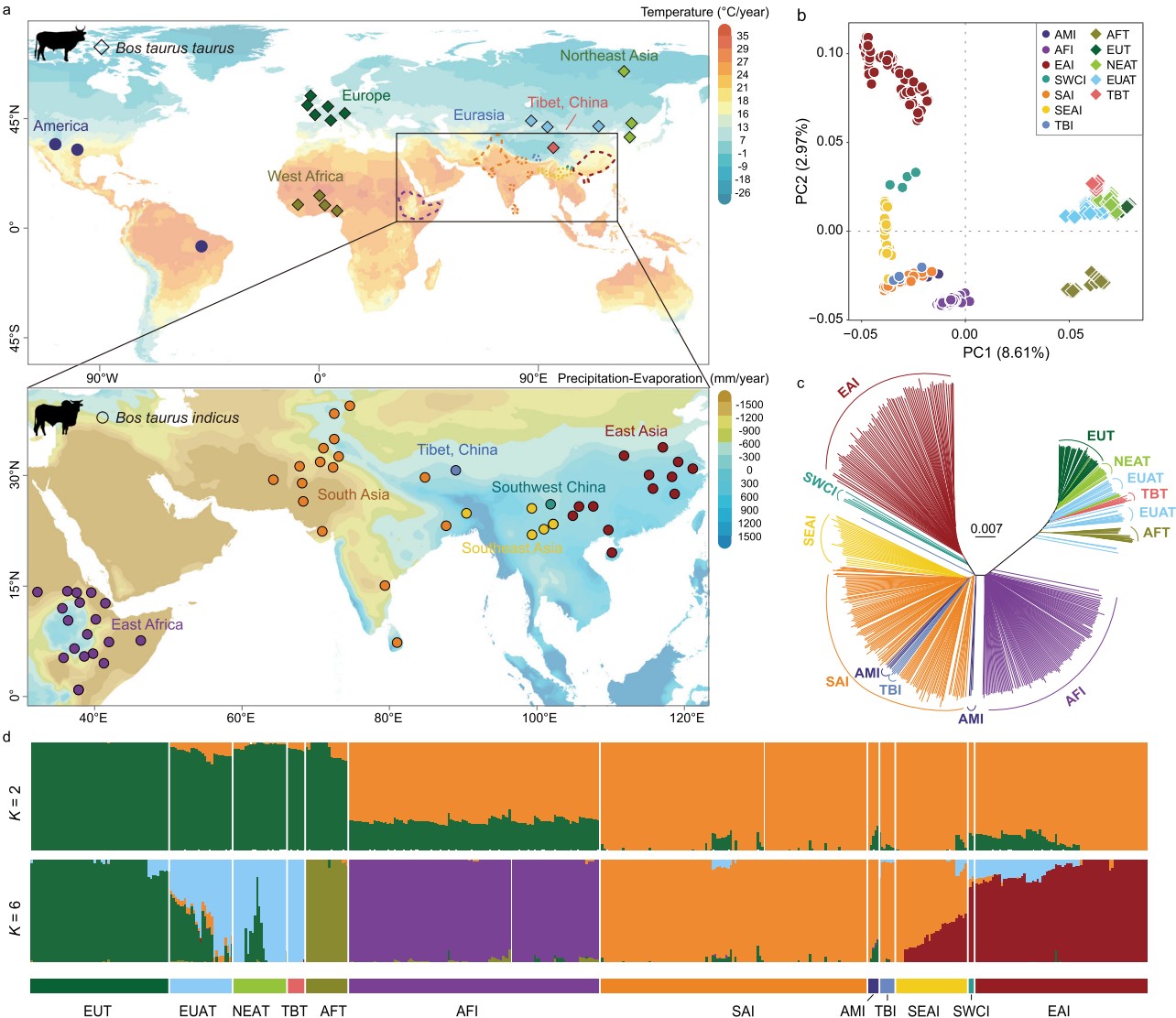

**Fig. 1 | Population genomic structuring and relationship among indicine cattle. a** Geographical locations of 74 taurine and indicine cattle breeds/populations included in this study. The map images were created by authors using https://impactlab.org/map. **b** Principal component analysis (PCA) showing PC1 versus PC2 of all cattle breeds/populations. **c** Neighbor-joining tree constructed using whole-genomic autosomal SNP data. **d** Admixture patterns for $K = 2$, dividing these cattle breeds/populations into indicine and taurine cattle, and $K = 6$, revealing each of the taurine and indicine phylogeographic groups.

## Adaptation of South Asian indicine cattle

Throughout the history of migration and admixture of indicine cattle, genomic regions under selection might have been lost in specific indicine groups. We therefore also performed a test for positive selection signatures in SAI cattle using $\theta_{\pi}$, *iHS*, CLR, and $F_{ST}$ estimates based on the comparison between SAI and non-SAI groups (Supplementary Data 2). SAI comprised ancestral indicine cattle that have adapted to the harsh conditions with high temperatures and solar radiation but low rain fall. Gene Ontology (GO) category and Kyoto Encyclopedia of Genes and Genomes (KEGG) pathway analyses revealed major enrichments of positively selected genes (PSGs) involved in environmental adaptation, including arachidonic acid metabolic process (GO:0019369) and infectious disease (human papillomavirus infection, bta05165) (Supplementary Table 5). Notably, the arachidonic acid metabolic pathway has been considered as an important molecular mechanism for desert adaptation[22,23] via efficient water reabsorption in sheep[24], chickens[23], and Bactrian camels[22], implying a deep convergent evolution among indicine cattle and other species survival over high salt diets in similarly arid and hot environments. We also detected several PSGs in SAI cattle, to be associated with fluid homeostasis (*APELA*[25]) and sensory ability (*CALB2*[26])

(Supplementary Data 2). Therefore, this comparison revealed a variety of important genes, pathways, and GO categories associated with the genetic adaptation of SAI cattle to semiarid environments (Supplementary Table 5).

Additionally, we compared the light- and dark-coated SAI breeds, such as the white-coated Bhagnari and Dajal cattle from Pakistan, by using $F_{ST}$ and $\theta_{\pi}$ ratio estimates. We identified shared selective sweeps around pigmentation loci, e.g., *LEF1* and *ASIP*, in the light-coated indicine breeds (Fig. 2). This selection pressure may have been favored or driven by high temperatures and intense solar radiation and/or human preferences. Across the whole genomes, the CLR and *iHS* analyses revealed 368 regions overlapping with 477 genes present in AFI and SAI cattle (Supplementary Data 3), supporting that the ancestral adaptations of SAI cattle were equally important for AFI cattle.

## The indicine adaptation to the tropical, humid Asian environment

Although domestic indicine cattle emerged in South Asia[6], different phylogeographic groups have evolved to adapt to local environments

**Table 1 | Summary of major genes under selection in indicine cattle**

| BTA | Region (Mb) | $F_{ST}$ | $\theta_\pi$ | XP-EHH | Gene identified | Association | Reference |
|---|---|---|---|---|---|---|---|
| 1 | 81.58–81.69 | 0.74 | 0.23 | 3.42 | *LIPH* | Hair development | 21 |
| 7 | 43.18–43.29 | 0.69 | 0.49 | 2.62 | *FGF22* | Hair development | 79 |
| 7 | 50.14–50.31 | 0.83 | 0.57 | 0.78 | *LRRTM2, CTNNA1, SIL1, MZB1, PROB1, PAIP2, SLC23A1* | Brain development, muscle development, antiviral immunity, reproduction, vitamin C transporters | 5 |
| 7 | 50.64–51.15 | 0.84 | 0.58 | 2.26 | *SPATA24, DNAJC18, TMEM173, UBE2D2, ECSCR, CXXC5, PSD2, NRG2* | Fertility, reproduction, heat stress | 5 |
| 8 | 53.22–53.27 | 0.64 | 0.32 | 2.64 | *VPS13A* | Blood circulation | 80 |
| 16 | 50.50–50.67 | 0.74 | 0.22 | 3.02 | *PRKCZ, FAAP20* | Light response, DNA damage | 81, 82 |
| 19 | 26.38–26.45 | 0.72 | 0.34 | 3.62 | *SPAG7, PFN1, KIF1C, CAMTA2, ENO3* | Antiviral immunity, skeletal development, neurodegenerative disease, cardiac growth, muscle development, | 16–18, 20, 83,19 |
| 19 | 27.40–27.61 | 0.74 | 0.94 | 2.96 | *WRAP53, TMEM88* | DNA damage, heart development | 84, 85 |
| 22 | 55.80–55.85 | 0.64 | 0.15 | 2.20 | *TAMM41* | Heart valve development | 86 |

during their global dispersal, including nonnative humid, tropical regions[27]. The agroecologies of southern East Asia are referred to as mixed subhumid or humid systems in contrast to the mixed arid or semiarid system of South Asia[28]. Southern East Asian agroecologies are characterized by high humidity and rain fall, as well as a relatively high incidence of tropical diseases[10,28].

We first assessed whether any ancestral indicine genomic regions under selection were present in EAI cattle. We used the population branch statistic (PBS) with banteng as an outgroup to detect recent selection signatures in EAI cattle, while avoiding the effect of introgression from other bovine species on the selection of EAI cattle (Fig. 3 and Supplementary Table 6).

The longest gene-rich region of 410 kb under selection was observed on BTA22 (Fig. 3), which contained 14 protein-coding genes. Two of these genes are related to the host immune system (*MST1R*[29,30] and *MON1A*[31]) and four tumor suppressor genes are related to lung cancer (*SEMA3B*[32], *GNAI2*[33], *SEMA3F*[34], and *RBM5*[35]) (Fig. 3). Moreover, *TRPA1* on BTA14 was highly differentiated in EAI cattle and may be associated with thermal avoidance behavior (Fig. 3b). Other PSG associated with circannual clock (*FAM204A*[36]) may help EAI cattle adapt to seasonal changes in day length by modifying their behavior such as grazing at night. In addition, PSGs involved in the immune system (Supplementary Table 6) could confer tolerance to parasitic pathogens in EAI cattle.

We then assessed whether the local adaptation of EAI cattle may have involved introgression from other Asian bovine species. Increasing evidence for past introgressions between bovine species distributed in East Asia and Southeast Asia is available[9,10,12,37]. Such introgression may have facilitated the rapid adaptation of indicine cattle to humid regions[9]. Previously, we reported the introgression from banteng into southern Chinese cattle[9], while another study suggested admixture from an unknown source into EAI cattle[12]. In the present study, we used TreeMix[38], RFmix[39], the *D* statistic[40], and phylogenetic analyses of candidate introgressed fragments to investigate the gene flow from banteng and gaur into 97 EAI and SWCI genomes (Supplementary Note 4, Supplementary Table 7, Supplementary Figs. 14–16, and Supplementary Data 4 and 5). The proportions of banteng and gaur ancestries ranged from 1.13% to 10.21% and from 2.06% to 9.98% in the EAI genomes, respectively (Fig. 4a and Supplementary Data 6 and 7). EAI cattle in the southeastern coastal region of China showed the highest level of banteng and gaur ancestries (Supplementary Fig. 17). We used the *U20* statistic to identify frequently introgressed genes in the EAI genomes[41]. We calculated $U20_{SAI, EAI, banteng\ or\ gaur}$ (1%, 20%, and 100%) to be equal to the number of alleles where banteng or gaur had a particular allele fixed, while its frequency was less than 1% in SAI but greater than 20% in EAI genomes (Supplementary Fig. 18)[41]. We found 1267 genes in the EAI genomes to be of banteng origin and 1488 genes to be of gaur origin, with 921 genes

shared by both banteng and gaur (Supplementary Fig. 19). GO analysis revealed significant overrepresentations of introgressed genes involved in biological processes contributing to environmental adaptation, the nervous system, and the endocrine system (Fig. 4b and Supplementary Table 8).

Using a higher cutoff for the frequencies of banteng- or gaur-derived alleles in the EAI genomes ($U50_{SAI, EAI, banteng\ or\ gaur}$ (1%, 50%, 100%)), 70 introgressed genes in 32 candidate regions were shortlisted (Fig. 4c and Supplementary Tables 9 and 10) and then validated by phylogenetic analyses (Supplementary Figs. 20–25). Notably, one region on BTA1 (66.70–66.80 Mb) harbored genes relevant to water transport (*ILDR1*)[42], blood homeostasis (*CASR*)[43], and skin disease (*CSTA*)[44] (Fig. 4c and Supplementary Fig. 26). Another region on BTA25 (0.21–0.26 Mb) also demonstrated a clear pattern of introgression in EAI cattle (Fig. 4d–g). The geographic distribution of haplotypes in global cattle populations showed that the introgressed haplotype had the highest frequency in EAI cattle, which was supported by phylogenetic analysis (Fig. 4d), $F_{ST}$ values (Fig. 4e), and the degree to which the haplotypes were shared (Fig. 4f). This region contained a cluster of genes (*HBM, HBA, HBA1,* and *HBQ1*) involved in biologically relevant oxygen transport (GO:0015671) (Fig. 4 and Supplementary Table 8), which were also associated with resistance to severe malaria in humans[45]. Within the genes in the hemoglobin family cluster, 11 missense mutations showed significantly altered frequencies of specific alleles between the EAI and other indicine groups (Fig. 4g, h).

**Uniparental dispersal of indicine cattle**

Specific uniparental lineages may be informative for the reconstruction of historical migration patterns. Here, we identified 1389 SNPs in the male-specific region of the bovine Y chromosome (MSY) in 309 males (Supplementary Data 1), which were defined as the taurine haplogroups of Y1, Y2A, and Y2B and indicine haplogroups of Y3A and Y3B reported previously[9] (Supplementary Note 5 and Supplementary Figs. 27–30). Within Y3A, we resolved two minor sub-haplogroups of Y3A1 and Y3A2 and a major sub-haplogroup of Y3A3, whereas Y3B was divided into the sub-haplogroups of Y3B1, Y3B2, Y3B3, and Y3B4 (Fig. 5a). Most of these sub-haplogroups were present in SAI cattle, supporting that South Asia was a primary center of paternal genetic diversity of indicine cattle. Following a west–to–east genetic cline, the haplogroup Y3A was predominant in EAI (88.13%) and North-Central Chinese cattle (Supplementary Fig. 29)[9]. In contrast, Y3B was dominant in SAI and AFI cattle (89.76%) (Fig. 5a). A phylogenetic tree showed the divergence of Y3A1, followed by Y3A2 and then Y3A3, which correlated with their geographic ranges: Y3A1 in SAI and SEAI; Y3A2 in TBI and SWCI; and Y3A3 as the predominant sub-haplogroup in EAI and North-Central Chinese cattle (Supplementary Data 1 and 8 and Supplementary Fig. 29). Y3A3, which occurred only in Indochina and China, may have emerged as a new sub-haplogroup during the indicine eastward

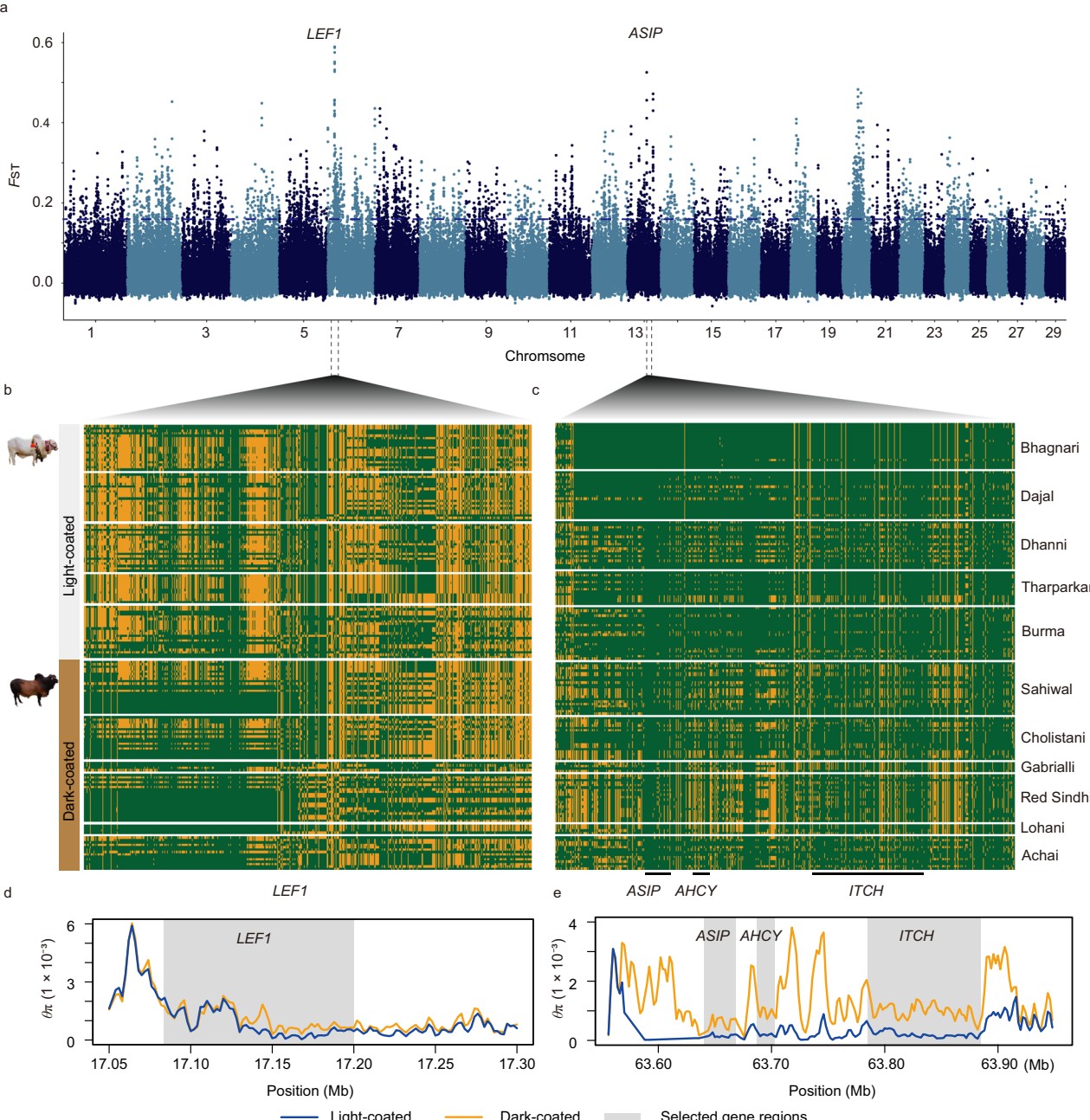

**Fig. 2 | Selective sweeps within the South Asian indicine cattle. a** Manhattan plot of the $F_{ST}$ values (y-axis) in windows of 50 kilobases (kb) using a 20 kb slide across all autosomes (X-axis). Names of genes within the highest peaks are shown. For a full list of the linked genes, see Supplementary Data 2. **b** and **c** Haplotype structures for selection regions on *Bos taurus* autosome (BTA) 6 and BTA13, respectively, in which rows represent individuals, columns represent polymorphic positions in the taurine cattle reference genome, and green and yellow indicate the alternative and reference alleles, respectively. Photos were taken by Quratulain Hanif. **d** and **e** Examples of genes with strong signals of selective sweeps in SAI cattle, in which $\theta_\pi$ estimates are plotted using a 10 kb sliding window in the *LEF1* and *ASIP* genomic regions on BTA6 and BTA13, respectively.

migration. Among the Y3B haplogroups, Y3B2 migrated to the east, while Y3B4 had a large range and was almost exclusively present in AFI cattle.

An ML tree of 347 assembled mitogenomes separated taurine and indicine cattle first and then added indicine I3 as a new haplogroup and I1a as a recent split-off of I1 (Fig. 5b and Supplementary Figs. 31–35). Similar to the Y3A3 sub-haplogroup, I1a may have emerged as a new haplogroup during the indicine eastward migration and got established in EAI and some North-Central Chinese cattle (Supplementary Data 8 and Supplementary Fig. 34)[9]. Notably, both the westward and eastward

migrations led to the fixations or near fixations of unique indicine Y-chromosomal and mitogenome haplogroups (Fig. 5a, b).

## Demographic history of indicine cattle

We used the multiple sequential coalescent Markovian model (MSMC) to infer historical changes in the effective population size and population separation of the three core indicine groups (EAI, SAI, and AFI). All indicine groups diverged from taurine groups between 251.5 and 301.2 thousand years ago (kya) and experienced common and substantial declines in $N_e$ at 20–30 kya and 7–9 kya. We observed earlier

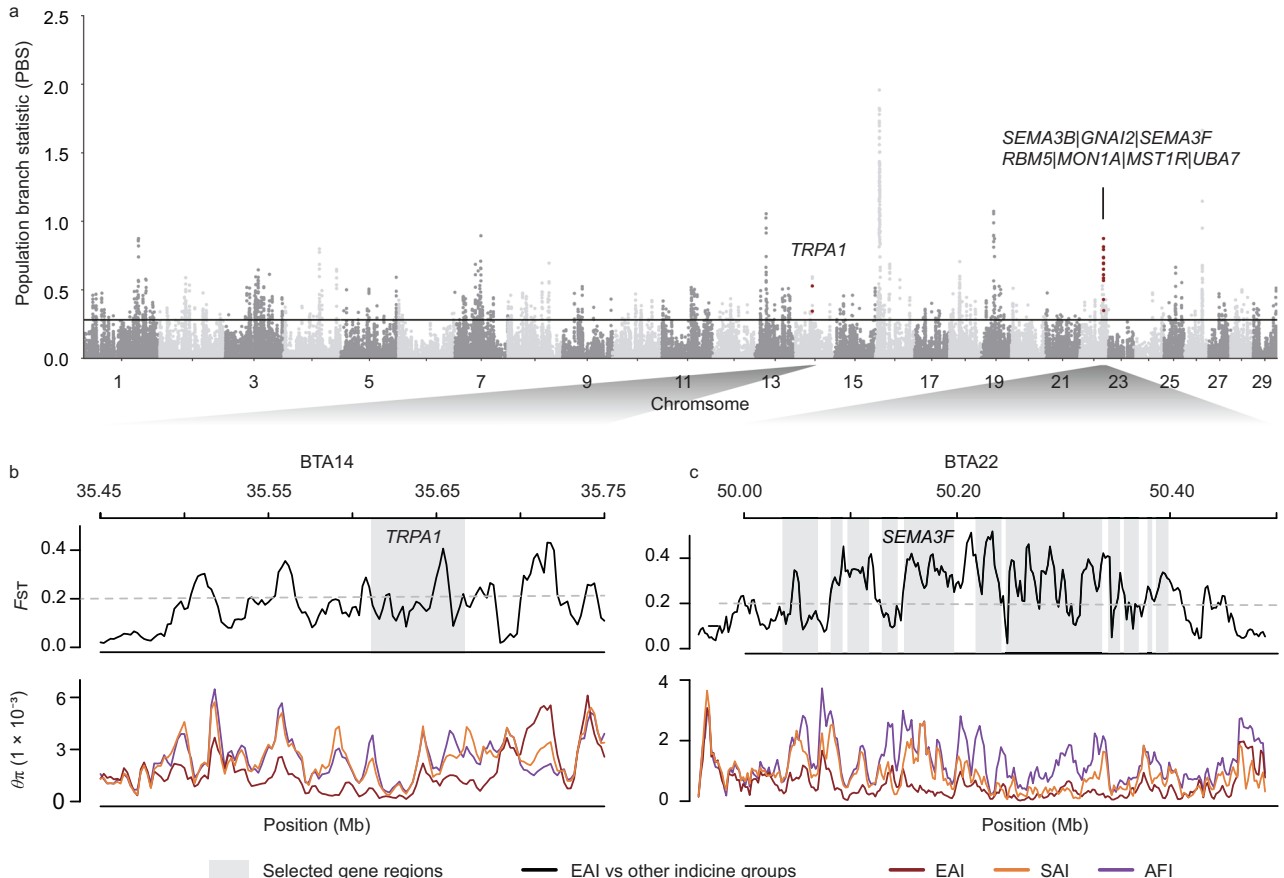

**Fig. 3 | Selective sweeps within East Asian indicine cattle. a** Manhattan plot of the population branch statistic (PBS) values (*y*-axis) in 50 kilobases (kb) windows with 20 kb steps across all autosomes (*x*-axis). Names of genes within the highest peaks are shown. A full list of the linked genes is included in Supplementary Table 6. **b** and **c** Examples of genes with strong selective sweep signals in EAI cattle, in which $F_{ST}$ and $\theta_\pi$ estimates are plotted using a 10 kb sliding window in the *TRPA1* and *SEMA3F* genomic regions on BTA14 and BTA22, respectively.

and clear splits at ~10.3 kya between EAI and SAI cattle and at ~11.8 kya between AFI and SAI cattle (Supplementary Fig. 36).

Using an empirical Bayesian approach, we detected more recent estimates of divergence times among indicine Y haplogroups than for mitogenome lineages. The divergence of Y3A from Y3B occurred at ~23.1 kya (Supplementary Fig. 30), while the newly emerged Y3A3 diverged from both Y3A1 and Y3A2 at ~20 kya (Supplementary Fig. 30). The Bayesian skyline plot (BSP) of indicine Y chromosome haplogroups displayed a slow population expansion from 10 kya, followed by an evident increase from 5 kya, which overlapped with the expansion of the post-domestication indicine populations (Supplementary Fig. 30). The divergence of sub-haplogroups within indicine mitogenomes occurred from 9.6 to 24.8 kya (Supplementary Fig. 32). The BSP of indicine mitogenomes showed a population decrease after domestication (10–8 kya) but a rapid increase from 7 kya, which overlapped with the expansion of the indicine populations after domestication (Supplementary Fig. 35). We also observed a population increase for I1a at ~3.73 kya, due probably to the expansion of I1a into the current distribution in East Asia (Supplementary Fig. 35).

**The Southeast Asian coast was the main entry point of indicine cattle into East Asia**
Indicine cattle were believed to have entered East Asia from South Asia through the inland routes[8,9]. However, we did not observe any west-to-east genetic clines, but a rather abrupt transition, as evidenced by different genetic features: (1) the drastic shift of haplogroup frequencies of both uniparental markers, with the Y-chromosomal sub-haplogroup Y3A3 and the maternal sub-haplogroup I1a exclusively

found in the Indochinese and Chinese cattle populations (Fig. 5); (2) the autosomal variation indicated relatively long-distance dispersals between SEAI and SWCI and between SWCI and EAI cattle (Fig. 1c); and (3) the extent of wild and/or domestic bovine introgression into EAI genomes did not follow a gradual west-to-east cline (Supplementary Fig. 17). Remarkably, all these transitions were geographically close to each other across Southwest China, where the mountainous landscape is traversed by three major rivers flowing from north to south (Nujiang River, Honghe River, and Lancang River; Supplementary Fig. 37). We propose that during the indicine eastward migrations, these geographic barriers were circumvented by maritime migrations along the coast (Fig. 5c), as maritime migrations also played an important role in the arrival of indicine cattle in Africa[5].

## Discussion
We conducted a comprehensive landscape genomic analysis of whole-genome sequence variations in the largest dataset available to date for indicine cattle breeds/populations across their major geographic distribution worldwide. We identified several loci that have been introgressed from banteng or gaur into EAI cattle, which may have facilitated their adaptation to the humid tropics and subsequent rapid dispersal. We also reconstructed the global indicine dispersal routes and provided estimated time frames.

Generally, indicine cattle have stronger adaptability and resistance than taurine cattle to heat, parasites, and infectious diseases in the tropics, especially in semiarid environments[3,46]. This study confirmed a few well-known adaptive loci related to heat tolerance and immunity at BTA7 in indicine cattle (e.g., *DNAJC18*, *HSPA9*, *MATR3*,

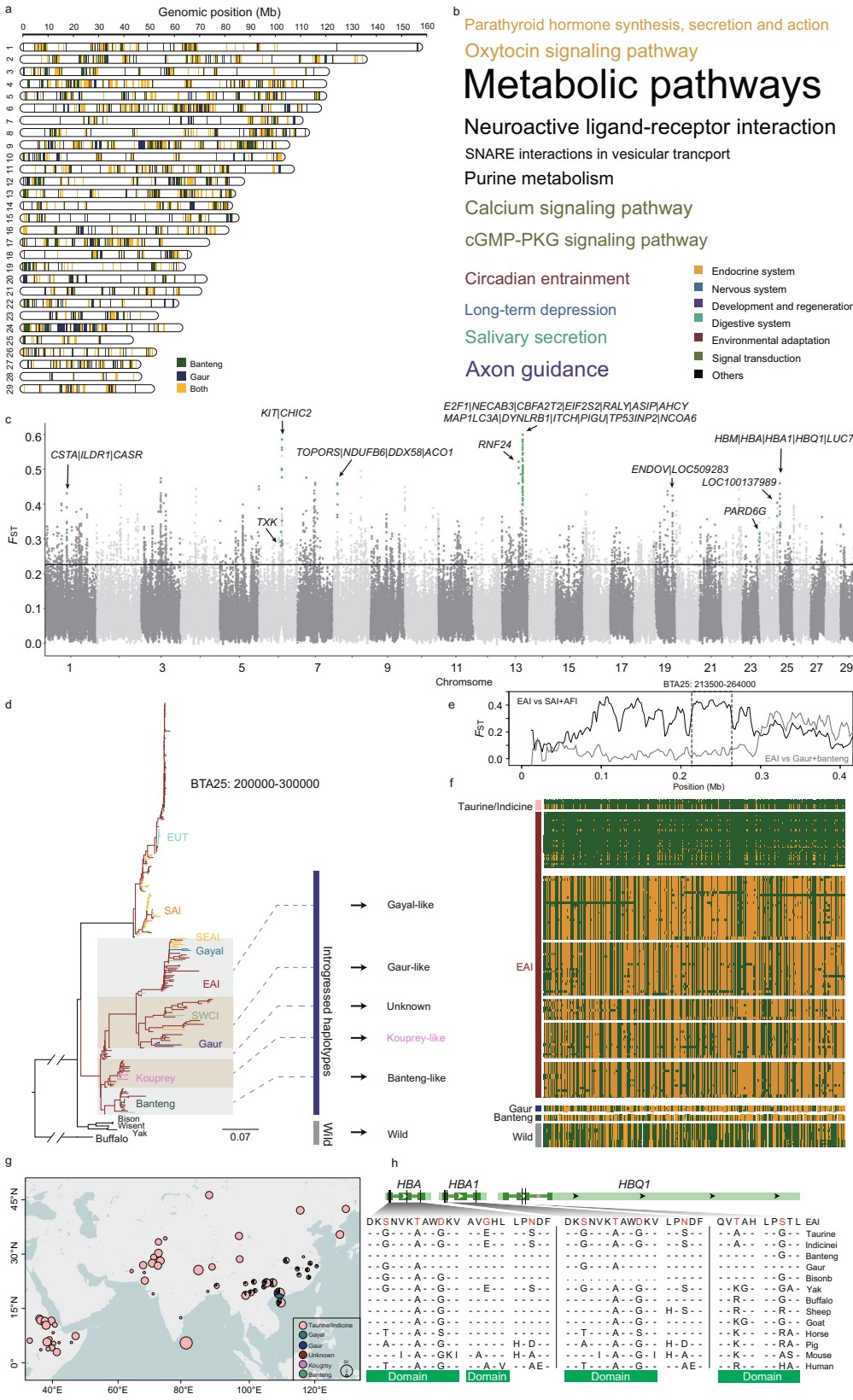

*MZB1*, and *STING1*)[5]. We also identified a selection signature near genes with functions related to hair growth, heart development, blood circulation, DNA damage repair, and solar radiation exposure, which may protect indicine cattle from heat and solar irradiation in hot regions[47]. Animal domestication likely also involved human preference for

specific traits such as coat color, which may explain the signatures of selection in *LEF1* and *ASIP* genomic regions.

From the domestication center in the Indus Valley, indicine cattle successfully expanded to other hot regions worldwide. Adaptation to new environments was accompanied by crossbreeding/hybridization

**Fig. 4 | Genome-wide introgression from banteng and gaur into East Asian indicine cattle. a** Map of the lengths and distributions of putatively adaptive introgressed segments in the EAI autosomes according to the results of the *U*20 statistic. The lengths of the colored columns are proportional to the physical lengths of the introgressed segments. **b** Introgressed segments from banteng into East Asian indicine (EAI) cattle show substantial enrichment for genes related to signal transduction and the digestive, endocrine, nervous, sensory, and circulatory systems. Word cloud color refers to the legend on the right of the terms. Font size is proportional to the gene number enriched in the pathway. **c** Plot of $F_{ST}$ values for autosomal SNPs between EAI cattle and both African indicine (AFI) and South Asian indicine (SAI) cattle based on a 10 kb sliding window. **d** Phylogenetic analysis of SNPs in introgressed genes (*HBM*, *HBA*, *HBA1*, and *HBQ1*). **e** $F_{ST}$ values of SNPs in a strong adaptive genomic region on *Bos taurus* autosome (BTA) 25 between EAI cattle and both SAI and AFI cattle as well as between EAI cattle and both banteng and gaur. **f** Different haplotypes of hemoglobin family members of banteng, gaur, gayal, kouprey, taurine, indicine, and other wild bovine species confirm the introgression from other bovine species into EAI cattle. **g** Geographical distribution of different haplotypes of hemoglobin family members in global cattle populations. **h** Amino acid sequence alignments of partial bovine HBA, HBA1, and HBQ1 along with their homologous sequences in other mammalian species. The map was drawn using the R package v4.1.0.

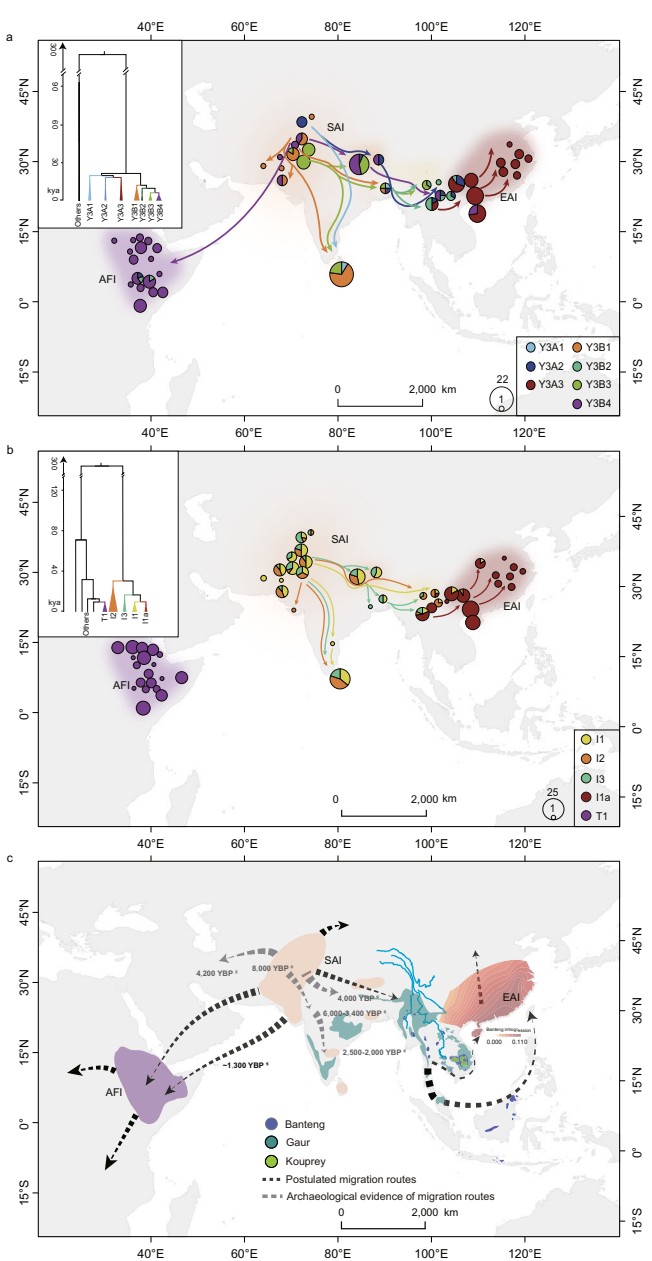

**Fig. 5 | The postulated global dispersal routes of indicine cattle.** The postulated dispersal routes of modern indicine cattle based on the distribution of Y haplogroups (**a**) and mitogenome haplogroups (**b**). The size of each circle is proportional to the number of samples per breed/population. **c** The synchronized routes and estimated times of the global dispersal of indicine cattle. The map was drawn using the ArcGIS v10.7.0.

with local bovine species, including taurine cattle in Africa and China and banteng, gaur, and gayal in East Asia. We detected clear signals of recent selection and adaptive introgression in EAI cattle from banteng and gaur, including several highly divergent loci. One of these regions contained the gene encoding the heat-sensing TRPA1 protein, which is conserved in mammals[48]. Another candidate region on BTA22 overlapped with immune- and tumor-related genes and may protect cattle against the high environmental infection burden in the tropics[30, 49]. Other candidate adaptively introgressed genes for heat adaptation were *ILDR*, which is important for paracellular water transport and the regulation of urine concentration[42], and *CASR*, which is responsible for maintaining blood $Ca^{2+}$ homeostasis[43]. We also detected an introgressed immune-related pathway involving the *HBA*, *HBA1*, *HBQ1*, and *HBM* genes, which are associated with resistance to severe malaria and anemia in humans[45] and may also confer the resistance of indicine cattle to tick-borne diseases such as tropical theileriosis[50, 51]. In addition to the introgression from banteng and gaur, other variants of hemoglobin-related genes have been introgressed from gayal, extinct kouprey or an unsampled *Bos*-like ghost species into EAI cattle (Fig. 4d)[12]. The sampling of indicine cattle from Indochina and other Asian wild bovine species may allow a better estimation of the contribution of wild bovine species to the environmental adaptation of different indicine groups.

We defined three major autosomal phylogeographic groups of SAI, AFI, and EAI cattle, within the global indicine cattle gene pool, two major paternal ancestries with six minor sub-haplogroups (Y3A1, Y3A2, Y3B1, Y3B2, Y3B3, and Y3B4) and three major mitogenome haplogroups (I1, I2, and I3), of which I1a sub-haplogroup represented a recent split from I1 (Fig. 5). Most of the mitogenome sub-haplogroups were present in SAI cattle, whereas the Y chromosomal Y3B4 and mitogenome T1 sub-haplogroups were fixed or nearly fixed in AFI cattle, while the Y chromosomal Y3A3 and mitogenome I1a sub-haplogroups were fixed in EAI cattle (Fig. 5). These observations confirmed that South Asia was the domestication center of indicine cattle.

Indicine cattle may probably entered East Asia between 3500 and 2500 YBP[6]. The phylogenetic tree showed the divergence of the Y chromosomal Y3A1, followed by Y3A2 and then Y3A3 sub-haplogroups. This pattern correlated with their geographic dispersal: Y3A1 in SAI and SEAI, Y3A2 in TBI and SWCI, and Y3A3 as the predominant sub-haplogroup in EAI and North-Central Chinese cattle. Mitogenome sub-haplogroup I1a nested within haplogroup I1 (Fig. 5) was not observed in SAI or SEAI cattle. Therefore, we propose that I1a emerged during indicine migration toward southern China. Eventually, following a founder effect, the migration from Southeast Asia to East Asia led to the establishment of the indicine Y chromosomal Y3A3 and mitogenome I1a sub-haplogroups in EAI and North-Central Chinese cattle. The estimated dates of sharp increases in both female and male populations in East Asia, as revealed by the first expansions of I1a and Y3A3, suggested the indicine entry into East Asia ~3 to 5 kya (Supplementary Figs. 30 and 35). Until now, indicine cattle are thought to have reached East Asia from South Asia through the inland trading routes[6], but our new evidence based on both uniparental and autosomal DNA variations support a coastal route for the first migration wave to Southeast Asia as the main entry point of indicine cattle into East Asia.

In conclusion, indicine cattle play an important role in the economy and culture of modern human societies. Human- and climate-mediated migration and specific wild/domestic bovine introgression have shaped the phylogeographic differentiation of mitogenome, Y-chromosomal, and autosomal DNA variations, driving unique tropical cattle herding behaviors on each continent. Our findings substantially expand the catalog of genetic variants in indicine cattle and reveal new insights into the evolutionary history and several plausible candidate genes for the unique adaptation of indicine cattle.

## Methods

### Ethics statement
Blood samples and ear tissues were collected during routine veterinary treatments with the logistical support and agreement of relevant agricultural institutions in each country. All procedures involving sample collection and experiments were approved by the Animal Ethical and Welfare Committee, Northwest A&F University (Approval No. DK2022065).

### Read mapping and SNP calling
We generated genotype data following the 1000 Bull Genomes Project Run 8 guideline (http://www.1000bullgenomes.com/) (Supplementary Note 1). We removed low-quality bases and artifact sequences using Trimmomatic v0.39[52], and all clean reads were mapped to the taurine reference assembly (ARS-UCD1.2) and Btau_5.0.1 Y using BWA-MEM v0.7.13-r1126 with default parameters[53]. We then used SAMtools v1.9[54] to sort bam files. For the mapped reads, potential PCR duplicates were identified using "MarkDuplicates" of Picard v2.20.2 (http://broadinstitute.github.io/picard). "BaseRecalibrator" and "PrintReads" of the Genome Analysis Toolkit (GATK, v.3.8-1-0-gf15c1c3ef)[55] were used to perform base quality score recalibration (BQSR) with the known variant file (ARS1.2PlusY_BQSR_v3.vcf.gz) provided by the 1000 Bull Genomes Project.

For SNP calling, we created GVCF files using "HaplotypeCaller" in GATK with the "-ERC GVCF" option. We called and selected candidate SNPs from these combined GVCF files using "GenotypeGVCFs" and "SelectVariants", respectively. To avoid possible false-positive calls, we used VariantFiltration of GATK as recommended by GATK best practices: (1) SNP clusters with "-clusterSize 3" and "-clusterWindowSize 10" options; (2) SNPs with mean depth (for all samples) < 1/3× and > 3× (×, overall mean sequencing depth across all SNPs); (3) quality by depth, QD < 2; (4) phred-scaled variant quality score, QUAL < 30; (5) strand odds ratio, SOR > 3; (6) Fisher strand, FS > 60; (7) mapping quality, MQ < 40; (8) mapping quality rank sum test, MQRankSum <−12.5; and (9) read position rank sum test, ReadPosRankSum < −8 were filtered. We then filtered out nonbiallelic SNPs and SNPs with missing genotype rates > 0.1. Imputation and phasing of SNPs were simultaneously performed using BEAGLE v4.0 with default parameters, and SNPs were filtered with DR2 < 0.9[56]. The remaining SNPs were annotated according to their positions using SnpEff v4.3[57].

### Genetic diversity and population genetic structure
The genome-wide nucleotide diversity of different cattle geographic groups was estimated with VCFtools v0.1.16[58]. Genetic distances between breeds/populations were calculated with the $F_{ST}$ estimates and ROH were analyzed using PLINK v1.9[59, 60] (Supplementary Note 2). For PCA and admixture analysis, we first filtered out SNPs with a minor allele frequency (MAF) < 0.01 and performed LD-based pruning for the genotype data using the --indep-pairwise 50 10 0.1 option of PLINK v1.9[57]. For PCA, we used the Smartpca program in EIGENSOFT v4.2[61]. The Tracy-Widom test was used to determine the significance level of the eigenvectors. ADMIXTURE v1.3.0 was used to quantify genome-wide admixture among cattle breeds/populations[62] and run for each possible group number (K = 2 to 8), where K was the assumed number of ancestries. The delta K method was used to choose the optimal K[62]. A

neighbor-joining (NJ) tree was constructed using the matrix of pairwise genetic distances calculated by PLINK v1.9[57]. A population-level phylogeny was reconstructed using the maximum likelihood (ML) in TreeMix[38].

### Detection of selection signatures shared by all indicine cattle groups
We screened genomic regions under selection with the largest differences in genetic diversity ($\theta_{\pi \ (indicine/taurine)}$ ratio) and $F_{ST}$ outliers between taurine (EUT, EUAT, NEAT, TBT, and AFT, n = 141) and indicine (SAI, EAI, and AFI, n = 309) cattle using VCFtools v0.1.16[58] (Supplementary Note 3). We also performed XP-EHH analysis using the default settings of selscan v1.1[63]. The π ratio, $F_{ST}$, and average normalized XP-EHH score were calculated for 50 kb windows with 20 kb steps. Top 1% windows were identified as significant genomic regions.

### Detection of selection signatures in South Asian, East Asian, and African indicine cattle
The CLR and $iHS$ were employed to detect the selection signatures in the SAI (n = 118), EAI (n = 80), and AFI (n = 111) genomes. The CLR was calculated for sites based on 50 kb windows with 20 kb steps using the SweepFinder2[64] command "SweepFinder2 -lu GridFile FreqFile SpectFile OutFile". The $iHS$ was calculated in selscan v1.1[63], and the proportion of SNPs with $|iHS| \geq 2$ was calculated in windows of 50 kb and steps of 20 kb. To perform $iHS$ and CLR computation, information on the ancestral and derived allele states is needed for each SNP. In our analysis, the ancestral allele was defined as the allele fixed in the swamp buffalo that was included in the genotype call set, while the SNPs failed in genotyping call for their ancestral state were discarded. To capture potential genes that were specifically selected for each indicine group, we also calculated the $F_{ST}$ between the target group and two other indicine groups. p values were calculated for the CLR, $|iHS|$, and $F_{ST}$ windows, and the overlap windows of p < 0.005 (Z test) of each method were considered candidate signatures of selection.

Considering that the EAI genomes were affected by banteng/gaur introgression, we used the PBS[65] in 50 kb windows with 20 kb steps to scan for genomic regions highly differentiated in EAI relative to SAI, AFI, and banteng (n = 4) genomes. Significant genomic regions were identified by a p < 0.005. In addition, $F_{ST}$ and $\theta_{\pi}$ methods were used to visualize the line chart of the top signals.

### Introgression analysis
TreeMix[38], the D statistic[40], and RFMix v2.02[39] were used to determine the gene flow between EAI and other bovine species. OptM was used to determine the optimal number of migration edges in the TreeMix. RFMix was used to identify regions introgressed from banteng or gaur into EAI cattle[39] (Supplementary Note 4). Pure taurine cattle, SAI cattle, banteng or gaur were selected as the reference panel according to D and $f_3$ statistics (Supplementary Note 4). We calculated the probability of banteng/gaur introgressed tracts in EAI cattle due to incomplete lineage sorting (ILS)[66]. We let r be the recombination rate per generation per base pair (bp) in indicine cattle, m be the length of the introgressed tracts, and t be the length of other bovine species (banteng and gaur) and cattle branches since divergence[10]. The expected length of a shared ancestral sequence was L = 1/(r × t) = 206.52 bp. The probability of a length of at least m was 1-GammaCDF (m, shape = 2, r = 1/L), in which GammaCDF is the gamma distribution function. We applied the probability of ILS < 0.05 to filter short introgressed segments in the RFMix results (Supplementary Data 6 and 7). A total of 79 topological trees were used to confirm banteng or gaur introgression and visualized by DensiTree[67] (Supplementary Fig. 16). Second, we used the statistics $U20_{SAI, \ EAI, \ banteng \ or \ gaur}$ (1%, 20%, and 100%)[41] and $U50_{SAI, \ EAI, \ banteng \ or \ gaur}$ (1%; 50%; 100%) to detect sites based on 50 kb windows with 20 kb steps where banteng or gaur had a particular allele at a frequency of 100%, while the frequency was less than 1% in SAI

but greater than 20% or 50% in EAI cattle (Supplementary Tables 9 and 10).

## Functional enrichment analyses

As source of annotation, we used the source *Bos taurus* Annotation Release 106 (GCF_002263795.1_ARS-UCD1.2_genomic.gtf) based on the NCBI assembly of GCF_002263795.1. Gene set enrichment analyses were carried out with GO categories and KEGG pathways for KOBAS v3.0[68]. The value was calculated using a hypergeometric distribution and corrected for the FDR. To adjust for multiple testing, pathways with $p < 0.05$ were considered significantly enriched.

## Paternal analysis

The X-degenerate region that consists of single-copy genes within the male-specific part of the Btau_5.0.1 Y chromosome reference sequence (GCF_000003205.7) was selected (Supplementary Note 5). After removing sites with missing genotypes in 10% of the samples, 1389 SNPs were extracted. Fasta sequence files were used to generate haplogroup trees. Phylogenetic construction was performed using BEAST v2.6.0[69]. To further explore the migration of Y3A haplotypes in China, we extracted 26 indicine Y haplotypes representing 11 hybrid breeds from North-Central China in previous studies[9]. We genotyped 26 individuals according to these 1389 SNPs (Supplementary Data 8).

Bayesian age estimates of haplogroups and Bayesian skyline plots (BSPs) were generated using BEAST v2.6.0[70]. A maximum clade credibility tree was generated using a 10% burn-in with TreeAnnotator and drawn with FigTree v1.4.3[71]. The BSPs of the indicine sub-haplogroup Y3 and its sub-haplogroup Y3A3 were generated. The following parameters were applied in both runs: HKY substitution model with gamma-distributed rates, a log-normal relaxed clock, coalescent Bayesian skyline analysis, a mutation rate per generation of $1.26 \times 10^{-8}$, and a generation time of 6 years[72]. The node age of sub-haplogroup Y3A3 (5.57 kya) was used as the only a priori parameter. We ran 100,000,000 iterations for Y3 and 50,000,000 iterations for Y3A3, with samples collected every 5000 steps, and visualized the BSPs obtained with Tracer v1.7[73]. LogCombiner was used to perform 10% burn-in. The results were calibrated with a generation time of 6 years, and BSP plots were plotted using the *ggplot2* in R v4.1.0[74].

## Mitogenome phylogeny

We assembled and selected 344 mitogenomes and aligned them to 18 bovine mitogenomes (Supplementary Note 5). Phylogenetic relationships were inferred from the mtgenomes using RAxML v8.2.9[75] with the following parameters: -f a -x 123 -p 23 -# 100 -k -m GTRGAMMA. The final tree topology was visualized using FigTree v1.4.3[71]. The median-joining network was constructed using NETWORK v5.0.1.1[71]. We extracted mitogenomes representing 13 hybrid breeds from North-Central China to further explore the migration of I1a sub-haplogroup in East Asia[9] (Supplementary Data 8).

Bayesian age estimates of haplogroups and BSPs were generated using BEAST v2.6.0[70]. BEAST runs were performed on three datasets with mitogenome coding regions (all 362 mitogenomes, 243 indicine mitogenomes, and 119 taurine mitogenomes). We used the HKY substitution model (with gamma-distributed rates) with a log-normal relaxed clock. We applied an evolutionary rate of $2.043 \pm 0.099 \times 10^{-8}$ base substitutions per nucleotide per year[76]. For each dataset, we performed ten independent BEAST runs with the chain length established at 20,000,000 iterations, samples collected at every 5000 MCMC steps and applying a 10% burn-in. The runs were then combined using the LogCombiner utility within the BEAST package by applying another 10% burn-in. A maximum clade credibility tree was drawn with FigTree v1.4.3[71]. BSP data were obtained with Tracer v1.7.1[73] using default parameters and calibrated using a generation time of 6 years[77]. The BSPs were plotted using the *ggplot2* in R v4.1.0[74].

## Estimation of effective population size and divergence time using autosomal SNPs

The multiple sequential coalescent Markovian model 2 (MSMC2)[78] method was used to model the population history of the three core indicine groups (EAI, SAI, and AFI) and to infer historical changes in their effective population size and population separation (Supplementary Note 5).

## Data availability

The newly whole-genome sequences for 297 samples data generated in this study have been deposited at the National Center for Biotechnology Information BioProject database (https://www.ncbi.nlm.nih.gov/bioproject) under the Bioproject accession number of PRJNA658727. The details of data mentioned above and other downloaded publicly available data used in this study are provided in Supplementary Data 1. The publicly available sequences were downloaded from the NCBI BioProject and China National Center for Bioinformation with the following project accession numbers: PRJCA002681 (Thawalam), PRJEB28185 (Yakutian and Finn cattle), PRJEB31621 (Lagune, Somba, and *Bos gaurus*), PRJNA176557 (Angus, Gelbvieh, Hereford, and Holstein), PRJNA210519 (Hanwoo), PRJNA256210 (Simmental), PRJNA277147 (Nelore and Gir), PRJNA285834 and PRJNA285835 (*B. grunniens*), PRJNA312138 (Ndama, Ogaden, Kenya Boran, and Kenana), PRJNA312492 (*Bison bonasus*), PRJNA318089 (Jersey), PRJNA321590 (*Bison bison* and *B. bonasus*), PRJNA324822 (Brahman), PRJNA325061 (*B. bison* and *B. javanicus*), PRJNA350833 (*Bubalus bubalis*), PRJNA379859 (Kazakh, Chaidamu, Yanbian, Tibetan, Tharparkar, Hariana, Sahiwal, Dianzhong, Wenshan, Dabieshan, Jinjiang, Guangfeng, Ji'an, Wannan, and Leiqiong), PRJNA386202 (Muturu), PRJNA396672 (Yanbian, Dehong, Wenling, and Minnan), PRJNA427536 (*B. gaurus*), PRJNA565271 (Yanbian), and PRJNA598339 (Mongolian). The known variant file (ARS1.2PlusY_BQSR_v3.vcf.gz) for base quality score recalibration was provided by the 1000 Bull Genomes Project (http://www.1000bullgenomes.com/).

## Code availability

All code and software sources used in our paper are listed in the "Methods" section with corresponding cite of references.

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

## Acknowledgements

This project was supported by grants from the National Key Research and Development Program of China (2021YFD1200400, SQ2021YFF1000041, and 2022YFF1000100) (W.L., N.C., and Y.J.), the earmarked fund for China Agriculture Research System of MOF and MARA (CARS-37) (C.L. and B.H.), the Yunnan Expert Workstations (202305AF150156), the National Natural Science Foundation of China (31872317), Foreign Young Talents Program (QN2022172008L) (C.L.), fellowships of the China Postdoctoral Science Foundation (2021T140564 and 2020M683587), the National Natural Science Foundation of China (32102523), the Shaanxi Youth Science and Technology New Star (2022KJXX-77), and High-end Foreign Experts Recruitment Plan (G2022172032L) (N.C.), Shaanxi Postdoctoral Science Foundation (2023BSHEDZZ132) (X.X.), the Program of Yunling Scholar and Yunling Cattle Special Program of Yunnan Joint Laboratory of Seeds and Seeding Industry (202205AR070001) and Construction of Yunling Cattle Technology Innovation Center and Industrialization of Achievements (2019ZG007) (B.H.), the National Natural Science Foundation of China (32072720) (Y.M.), the Chinese Government's contribution to the CAAS-ILRI Joint Laboratory on Livestock and Forage Genetic Resources in Beijing (2023-YWF-ZX-02) (J.H.), the International Livestock Research Institute's Livestock Genetics Program that was supported by the CGIAR Research Program on Livestock (CRP livestock) and sponsored by the CGIAR funding contributors to the Trust Fund (http://www.cgiar.org/about-us/our-funders/), partly by the Bill and Melinda Gates Foundation and UK aid from UK Foreign, Commonwealth and Development Office (Grant Agreement OPP1127286) under the auspices of the Centre for Tropical Livestock Genetics and Health (CTLGH), established jointly by the University of Edinburgh, SRUC (Scotland's Rural College) and ILRI (O.H.). This project was also supported by the Addis Ababa University, Ethiopia, the Italian Ministry of Education for Universities and Research (MUR) for PRIN2017 20174BTC4R (A.A.), the "Fondazione Adriano Buzzati–Traverso for the Luigi Luca Cavalli-Sforza fellowship (N.R.M.), the Carlsberg Foundation (CF20-0355) (M.H.S.S.), and the National Natural Science Foundation of China International Cooperative Research and Exchange Program (31861143014) (W.L.). Finally, we thank the High-Performance Computing (HPC) Center of Northwest A&F University (NWAFU) and Hefei Advanced Computing Center for providing computing resources and Lucia Mazzocchi for her contribution to the Y chromosome computational analysis.

## Author contributions

J.H., O.H., Y.J., and C.L. designed and supervised the project. N.C. and X.X. performed the majority of analysis with contributions from Q.H.,

F.Z., R.D., B.H., Y. Lyu, X. Luo, H.Y., S.W., F.W., J.C., X.G., Y. Liu, S.L., L.J., P.W., L.S., N.R.M., G.C., O.S. and A.A. Q.H., J.Z., J.L., K.Q., Y.C., J.S., Y. Liao, Z.X., M.C., L.M., A.Z.S., M.A., S.M., M.E.B., T.H., G.L.L.P.S., N.A.G., E.T., G.B., A.T., T.Z., M.G.G., Y.M., Y.W., Y.H., X. Lan, H. Chen, H. Cheng, W.L., C.L., J.H., and O.H. prepared the modern samples. N.C. and X.X. wrote the manuscript with input from all authors, whereas C.L., J.H., O.H., H.Z., A.A., M.H.S.S., and J.A.L. revised the manuscript.

## Competing interests

N.C., X.X., C.L., H. Chen, H.Y., and X. Lan are inventors on a patent application related to this work submitted on 29 October 2021 by Northwest A&F University (Patent no. ZL202111277432.3). All authors declare that they have no other competing interests.

## Additional information

Ningbo Chen [1,31], Xiaoting Xia[1,31], Quratulain Hanif [2,3,31], Fengwei Zhang[1,31], Ruihua Dang[1,31], Bizhi Huang[4,31], Yang Lyu[1], Xiaoyu Luo[1], Hucai Zhang[5], Huixuan Yan[1], Shikang Wang[1], Fuwen Wang[1], Jialei Chen[1], Xiwen Guan[1], Yangkai Liu[1], Shuang Li[1], Liangliang Jin [1], Pengfei Wang[1], Luyang Sun[1], Jicai Zhang[4], Jianyong Liu[4], Kaixing Qu[6], Yanhong Cao[7], Junli Sun[7], Yuying Liao[8], Zhengzhong Xiao[7], Ming Cai [4], Lan Mu[9], Amam Zonaed Siddiki[10], Muhammad Asif[2], Shahid Mansoor[2], Masroor Ellahi Babar[11], Tanveer Hussain [12], Gamamada Liyanage Lalanie Pradeepa Silva[13], Neena Amatya Gorkhali[14], Endashaw Terefe [15,16], Gurja Belay[17], Abdulfatai Tijjani[16,18], Tsadkan Zegeye [19], Mebrate Genet Gebre[20], Yun Ma[21], Yu Wang[1], Yongzhen Huang[1], Xianyong Lan[1], Hong Chen[1], Nicola Rambaldi Migliore [22], Giulia Colombo [22], Ornella Semino [22], Alessandro Achilli [22], Mikkel-Holger S. Sinding [23], Johannes A. Lenstra [24], Haijian Cheng[1,25], Wenfa Lu[26], Olivier Hanotte [16,27] ✉, Jianlin Han [3,28,29] ✉, Yu Jiang [1,30] ✉ & Chuzhao Lei [1] ✉

[1]Key Laboratory of Animal Genetics, Breeding and Reproduction of Shaanxi Province, College of Animal Science and Technology, Northwest A&F University, Yangling 712100, China. [2]National Institute for Biotechnology and Genetic Engineering, Faisalabad 38000, Pakistan. [3]CAAS-ILRI Joint Laboratory on Livestock and Forage Genetic Resources, Institute of Animal Science, Chinese Academy of Agricultural Sciences (CAAS), 100193 Beijing, China. [4]Yunnan Academy of Grassland and Animal Science, Kunming 650212, China. [5]Institute for Ecological Research and Pollution Control of Plateau Lakes, School of Ecology and Environment Science, Yunnan University, Kunming 650500, China. [6]Academy of Science and Technology, Chuxiong Normal University, Chuxiong 675000, China. [7]Guangxi Vocational University of Agriculture, Nanning 530007, China. [8]Guangxi Veterinary Research Institute, Guangxi Key Laboratory of Veterinary Biotechnology, Nanning 530001, China. [9]College of Landscape and Horticulture, Southwest Forestry University, Kunming 650224, China. [10]Genomics Research Group, Department of Pathology and Parasitology, Faculty of Veterinary Medicine, Chattogram Veterinary and Animal Sciences University (CVASU), Chattogram 4225, Bangladesh. [11]The University of Agriculture, Dera Ismail Khan, Khyber Pakhtunkhwa 29050, Pakistan. [12]Department of Molecular Biology, Virtual University of Pakistan, Islamabad 44100, Pakistan. [13]Department of Animal Science, University of Peradeniya, Peradeniya 20400, Sri Lanka. [14]National Animal Breeding and Genetics Centre, National Animal Science Research Institute, Nepal Agriculture Research Council, Khumaltar, Lalitpur 45200, Nepal. [15]College of Agriculture and Environmental Science, Department of Animal Science, Arsi University, Asella, Ethiopia. [16]International Livestock Research Institute (ILRI), P.O. Box 5689, 1000 Addis Ababa, Ethiopia. [17]College of Natural and Computational Sciences, The School of Graduate Studies, Addis Ababa University, 1000 Addis Ababa, Ethiopia. [18]The Jackson Laboratory, Bar Harbor, ME 04609, USA. [19]Mekelle Agricultural Research Center, P.O. Box 258, 7000 Mekelle, Tigray, Ethiopia. [20]School of Animal and Rangeland Science, College of Agriculture, Haramaya University, 2040 Haramaya, Oromia, Ethiopia. [21]Key Laboratory of Ruminant Molecular and Cellular Breeding of Ningxia Hui Autonomous Region, School of Agriculture, Ningxia University, Yinchuan 750000, China. [22]Department of Biology and Biotechnology "Lazzaro Spallanzani", University of Pavia, 27100 Pavia, Italy. [23]Section for Computational and RNA Biology, Department of Biology, University of Copenhagen, DK-1350 Copenhagen, Denmark. [24]Faculty of Veterinary Medicine, Utrecht University, 3584 CM Utrecht, The Netherlands. [25]Institute of Animal Science and Veterinary Medicine, Shandong Academy of Agricultural Sciences, Shandong Key Lab of Animal Disease Control and Breeding, Jinan 250100, China. [26]College of Animal Science and Technology, Jilin Agricultural University, Changchun 130118, China. [27]School of Life Sciences, University of Nottingham, Nottingham NG7 2RD, UK. [28]Livestock Genetics Program, International Livestock Research Institute (ILRI), 00100 Nairobi, Kenya. [29]Yazhouwan National Laboratory, Sanya 572024, China. [30]Key Laboratory of Livestock Biology, Northwest A&F University, Yangling 712100, China. [31]These authors contributed equally: Ningbo Chen, Xiaoting Xia, Quratulain Hanif, Fengwei Zhang, Ruihua Dang, Bizhi Huang. ✉ e-mail: o.hanotte@cgiar.org; h.jianlin@cgiar.org; yu.jiang@nwafu.edu.cn; leichuzhao1118@nwafu.edu.cn

