## [Peer Review File · Nature Communications]

Global genetic diversity, introgression and evolutionary adaptation of indicine cattle revealed by whole genome sequencingREVIEWER COMMENTS

Reviewer #1 (Remarks to the Author):

General comments

Chen et al. present the results of whole-genome resequencing of 355 indicine cattle genomes and 141 taurine cattle genomes from 57 and 17 populations, respectively. The quantity and quality of the material studied here, the methods used and the results obtained are of a high standard and deserve publication in a high-ranking journal. However, the main problem concerns the core zebu breed group and the outgroups.

Regarding the core Zebu group: The Zebu cattle breeds from India were missing in this study. The surrogates used for this purpose are Gir, Nellore and Brahman, which were bred outside India for centuries in new environments and with sporadic interchange with *Bos taurus* breeds. Introgression of *Bos taurus* into the WGS of Gir, Nellore and Brahman used here is neither ruled out nor investigated by the authors. Brahman in particular is known to be a synthetic breed with *Bos taurus* content. Many countries such as Sri Lanka, Pakistan and Nepal (SAI group) use *Bos taurus* breeds to improve local cattle breeds. Therefore, the purity of any reference used for introgression studies should be tested first.

With regard to outgroups: Besides bison and yak, two Bantengs and two Bours are used for important introgression analyses in this study. Again, there is a possibility of introgression of cattle into Banteng and/or Bours. This possibility has not been ruled out and has not been investigated. There are some unpublished results confirming cattle DNA segments in Bantengs. I do not have comparable information for Gaur, but this is possible. As shown in the study analysing cattle introgression in Yak, RFMix can produce false positive signals if the reference population is itself introgressed.

The next problem is the sample size of the outgroups (Banteng and Guar) in the introgression analysis. I expect that the relatively small sample size of the outgroups (2 samples each) will affect the performance of RFMix and the determination of ancestral alleles and their frequencies in the U20 and U50 analyses. In fact, the original paper of RFMix already describes how the sample size of the reference panel affects its performance (<https://www.ncbi.nlm.nih.gov/pmc/articles/PMC3738819/figure/fig6/>). How did the authors determine the robustness of their results?

The results of the D-statistics are confounded by the structure of the ancestral population as well as incomplete lineage sorting. I recommend that the authors perform other independent approaches, e.g. those based solely on branch length and tree discordance, to validate these results.

How did the authors determine the optimal number of migration edges in the TreeMix? Ideally, they should also include the matrix of residual values to show how integrating migration edges improved the % of variance explained.

How did the authors determine the expected length of introgressed haplotypes? How did they make sure that it is significantly different from the expected length because of its shared ancestral origin? Does the length of introgressed haplotypes tell us anything about the age of introgression?

Each figure and table, including supplementary figures and tables, should be considered as a stand-alone object, i.e. the readers should be able to understand it without reading other objects. For this reason, the legend of the figure or table must be informative enough to explain the main message of this stand-alone object. This is not the case. I make a specific comment for one figure only (mentioned in the comments below), but it applies to all and is therefore a general comment.

Specific comments

Line 87-88, “..resilient to local pathogens”: consider replacing with tick resistance as also mentioned in the reference paper cited here.

Line 106-108: Please keep in mind that there is a small but significant *Bos indicus* introgression also into almost all south-eastern European cattle breeds up to the southern Alps (<https://doi.org/10.1186/s12711-020-00560-8>).

Line 108-109, “...Gujarat in Western India”: While the reference paper do discuss the hypothesis about possible wild introgression in southern Indian zebu, the same is not mentioned for Gujarat, consider rewriting this sentence.

Line 144-145, “thirteen whole genome sequencing data...”: The authors should mention here the number of WGS used for each species separately.

Line 166-167, “...the PCA and a phylogenetic tree..”: The authors should describe here how SWCI fits into the overall phylogenetic reconstruction.

Line 176-178, “We identified candidate selected genomic regions...”: Why author only combined the three indicine groups (SAI, AFI, EAI) and left the other two (SEAI and SWCI) for this analysis?

Line 175ff: The authors use reverse genomics to detect candidate genes. They constructed two groups of animals/populations: 1) taurine and 2) SAI, AFI and EAI indicine populations. Both groups lived in environments that differed by a variety of factors, including temperature. The authors found numerous significantly differentiated regions and attribute them all to adaptation to high temperatures. The idea of reverse genomics is to find the outliers and then look for possible causes. Only after some causality analysis can we conclude that some of the outliers are an adaptation to high temperatures or something else. The authors should consider significant selection signatures as anonymous candidates. These anonymous candidates are subject to positive selection into a group of animals that share a specific environment. These anonymous candidates could, but need not, be caused

by one specific environmental factor.

Line 182-183, "In addition, the expression level...": As the tissue and animals (with respect to gender and age) are quite heterogenous, I will take this result with pinch of salt, moreover, how many other genes were differentially expressed and not under positive selection?

Line 192-195: Again, the authors link LIPH to a possible contribution to heat tolerance in indicine cattle by controlling hair length and/or thickness. However, we (readers) are not informed whether indicine and taurine cattle differ in hair length and thickness? I cannot find any such information in this manuscript.

Line 202ff: The SAI cattle group consists of breeds from Sri Lanka, Pakistan and Nepal (a very broad area), but all outliers are defined as candidates for adaptation to semi-arid environmental stress.

Line 223, "The genome-wide analysis shows...SAI origin": This is well-established fact; therefore, I would remove it or declare as confirmation.

Line 319-321: With regard to mtDNA distribution, a comparable situation can be observed in AFI and EAI. However, the presence of haplogroup T in Africa was not interpreted as T having emerged as a new haplogroup during the indicine westward migration, but as a haplogroup that was already present in the area before the indicine introgression. The authors should consider both the time of coalescence and the time of migration when drawing their conclusions.

Line 346-361: Samples originating from India and sampled there are missing in this study. This could be a reason why the west-east cline is missing and an abrupt transition is observed. I did not understand this section of the text. There are several problems in the text itself and in the corresponding figures. The legends of the figures are generally not informative enough. Let's just take Supplementary Figure 17 as an example. The reader cannot understand this figure from its legend. Even more, there are many trees labelled with the same A, B, C,.... and we do not know which tree represents which haplotype, etc. I also do not know what to conclude from figure 50, i.e. the legend does not help me to understand the message from line 355-357.

Line 414: (in brackets)???

Line 447-458: Two recent and comprehensive studies on paternal and maternal haplogroups in cattle could be helpful here (DOI: 10.1111/eva.13315 and doi: 10.1111/age.13104).

Line 459-470: This can be understood to mean that East Asia was a land without cattle, so that *Bos indicus* spread with the single matriline I1a in this empty area. This was also the case with the spread of domestic sheep in Europe. However, for sheep, there is

archaeological evidence for a sheep-less Europe before the Neolithic. Do you have any evidence that East Asia was a cattle-less region before the spread of *Bos indicus*? Africa was not cattle-less either, and as a result of paternal introgression we only observe haplogroup T. Please clarify.

Line 470-475: The reason behind this could be lack of sampled population along the inland trading routes from South to East Asia. Therefore, the integrated genetic analyses as mentioned in the paper may have difficulty supporting inland trading even if this is correct.

Line 558-559, “Treemix and Dsuite...”: Dsuite is a tool and not any method, please clarify this.

Supplementary table 1:

If possible, the country of origin should also be mentioned, also verify that breeds with Name “SriLanka” exists.

Supplementary table 2:

Haryana is declared as an Indian breed and the sample used here was sequenced in Chen et al. 2018 and should be from India. We know that India is very restrictive when it comes to foreign use of biological resources. There is no Indian collaborator in Chen et al. 2018, so I do not see any legal way to analyze Indian biological material outside India or without an Indian collaborator. Was this Haryana cattle sampled in India or elsewhere? Please clarify and correct if necessary in Supplementary Table 2.

Reviewer #2 (Remarks to the Author):

This manuscript is interesting and provides data from a unique sample of individuals that had not been previously characterized. Overall, I find the manuscript well written, although it appears to have been done in parts by different groups of individuals with different writing styles and slightly different nomenclature. This is inevitable with a project of this size. I would encourage the corresponding authors to go through the entire manuscript to make it more uniform. I do have a number of concerns related to the details that were presented, or not presented, that make a final determination not possible at this time.

I always start my reviews with the materials and methods but since there was such a substantial amount of information in the supplemental information, I will start there. Unfortunately, the supplemental document does not have line nor page numbers, so I'll quote specific lines.

Whole-genome resequencing

“Additional detailed information on the mapping rate and sequencing depth are provided in Supplementary Table 1” In Sup. Table 1 you use the SRR* ID as the sample. This is incorrect

because these are run IDs. You should use the BioSample ID. I downloaded this table and compared these SRR to the NCBI database and you have two duplicate samples. SRR2016752 and SRR2016754 are the same SAMN3387026 individual. With only a single duplicate, it is unlikely that any of the results would have been significantly affected. However, this table needs to be corrected to present the BioSample and if data reanalysis is performed, one of these two should be removed.

Population genomic analyses Genetic diversity

“Runs of homozygosity (ROH) were identified using the `--homozyg` option implemented in PLINK v.1.9, which slides a window of 50 SNPs (`--homozyg-window-snp 50`) across the genome.” This is a very small window considering the density of markers. It is well recognized that results are sensitive to settings for this analysis (see <https://doi.org/10.1186/s12864-020-6463-x>). I question whether or not the settings used were appropriate. Specifically, by using too small of a window size and other inappropriate settings, far too many ROH are detected. Evidence of this can be seen in Supplementary Fig. 2 where there are a significant number of samples that have estimated ROH totaling greater than 1/3 of the genome and a large number of samples with ROH greater than half the genome. This is unrealistic. This analysis needs to be redone with appropriate settings, which may need to be determined empirically given the data. It would also be useful at this time to examine any correlation between average genome coverage and number/length of ROH as one would predict that lower coverage samples will have heterozygous sites undercalled which may manifest as ROH when in fact there are not ROH if heterozygous positions were called accurately.

Principal component analysis (PCA)

“We removed “all LD” using...” First, please rephrase this. You did not remove “all LD”. I have the same objection here to the parameters used. First, this is a very small window and probably left a significant number of loci with $r^2 > 0.20$ which are more than 50kb apart. Given everything that we know regarding LD in bovine genomes, you should empirically find the appropriate setting for this analysis to achieve the desired thinning. This does not have to be done genome-wide and one could simply use 10Mb from any autosome to estimate what the appropriate parameters should be. My recommendation would be to use chr25 since we know there are no assembly issues. For a 10Mb region, calculate *all* the pairwise LD values using ‘-r2 yes really’ and examine the distribution of these values. This figure can be included in the supplement to provide justification for the settings chosen. As the manuscript is currently written, the settings are arbitrary and likely impacted the results and interpretation.

“PCA was performed on the genome-wide unlinked SNP dataset...” Please change the word “unlinked” throughout the manuscript. You do not have any unlinked variants. Perhaps use “LD pruned” as a replacement.

Structure analysis

“...200 bootstrap replicates were performed...” Why 200? Why not 10, or 10,000. Is the

default of 200 appropriate for your data?

“...used to determine the optimal ancestry number...” Exactly how did you determine the optimal number? There is quite a bit of literature on this, but nothing was stated or cited.

Neighbor-joining (NJ) and maximum likelihood (ML) phylogenetic trees

“...A window size of 1,000 SNPs was used to account for linkage disequilibrium...” This should be unnecessary if you properly LD pruned in the first place. This brings up another issue I have with the manuscript. You have many different analyses and it is difficult to tell exactly which set of loci you are using for which analysis. In this section, it appears that you are using the “raw” 67M variants. However, it seems to me that if the objective is to “identify closely related individuals” then the LD pruned dataset from the PCA section would be appropriate. However, it appears that these two analyses used different filtering. My recommendation would be to provide a supplemental table that lists the major analyses performed and exactly which data manipulations were done for each analysis. Since data manipulations may impact one analysis but not another, it would be helpful to the reader to understand exactly what was done and which analyses used the same loci. Furthermore, you can specify the total number of loci in this table for each dataset.

Detection of selection signatures shared by all indicine populations

“...were calculated for 50-kb windows with a 20 kb step across the autosomes...” Line 907 of the main manuscript says that a 100kb window was used, which is it? I have issues with what is specified here. Typically, these types of analyses are performed where the window size is an integer multiple of the step size. For instance 25 kb step and 50 kb window or 20kb step and 100kb window. This needs to be addressed throughout the manuscript and I would recommend using the same parameters for all analyses that use this type of setting (where appropriate). This will make comparisons between analyses much easier.

“...these windows harbored 117 candidate genes in indicine cattle...” I don’t think these 117 genes are specific to indicine cattle and you can probably just say cattle. By saying indicine cattle here one might think you are implying that these are not in taurine cattle, which isn’t the case.

“...the expression levels of these 117 candidate genes were validated in nine different tissues of taurine and indicine cattle.” I’m not sure exactly what you are validating? The fact that there are RNA-seq data in those tissues for these annotated genes simply validates that the annotation is correct.

“Candidate genes under selection were defined as those overlapped by sweep regions or within 20 kb of the signals.” 20kb seems arbitrary. What justification do you have for choosing 20kb? Why not 50kb, or 10 kb or simply require the gene to fall within the sweep region and not allow any overlap. It seems to me that allowing for a gene to be outside your sweep region, by some arbitrary distance, speaks to your confidence in defining the boundaries of sweep regions. Some justification should be given for whatever value you choose.

“...were plotted using a 10-kb sliding window...” Again, arbitrary. Why not use the same window size for plotting that you used for detection?

Detection of selection signatures in SAI, EAI, and AFI lineages

“...windows of 50 kb and a step of 20 kb...” Same comment on windows.

“...the overlap windows of P values less than 0.005 (Z test) of each method were considered candidate signatures of selection...” Why 0.005? Again, some justification is warranted. It seems to me that you have an extraordinarily large number of windows that were tested which implies that some adjustment needs to be considered for multiple testing.

Appropriately accounting for multiple testing here could significantly change the detected regions which in turn could significantly impact all downstream inference based on these results. As written, it is impossible to determine exactly what was done and what impact it may have on the results or interpretation.

“Only pathways or annotations with a Bonferroni-corrected $P < 10^{-2}$ were retained (Supplementary Tables 7-10).” This should probably be written as $P < 0.01$ to be consistent throughout the manuscript.

Introgression analysis

“Phylogenetic analyses of these segments confirmed the banteng or gaur introgression into specific EAI individuals.” Exactly how does this *confirm* introgression?

“Therefore, we used the statistic U_{20SAI} , EAI, banteng (1%, 20%, and 100%) 20, which was equal to the number of SNPs within a genomic window where a particular allele was fixed at a frequency of 100% in banteng but at a frequency less than 1% in SAI cattle or greater than 20% in EAI cattle.” For both the banteng and gaur analyses you only have two individuals of each species. This means that you have essentially no ability to estimate allele frequencies in gaur and banteng. This means that all of your results from analyses of this type are suspect, which in turn means that inference based on these results is suspect. There are more than two of each of these publicly available in SRA and your analysis would be significantly strengthened by using what is available to increase your sample size, which will allow you to better estimate allele frequencies in these species.

Paternal analysis

“...(i) only present in at least two males but not in females; (ii) hemizygous...” Are these not represented as homozygous in the actual data? In reality, the males are hemizygous but they are represented as homozygous when variant calling.

“BEAGLE was used to impute missing alleles.” Were samples set to homozygous for imputation and you simply pretended as if this was an autosome? Exactly how was Beagle run for Y-specific loci? The underlying model includes recombination rate, which is zero for the Y. It’s unclear to me how this was performed.

Whole mitogenome phylogeny

“...samples with a depth of coverage lower than 100× were disregarded.” Why? The depth of coverage on the MT is a function of the overall average genome coverage and the tissue source for the DNA. This biases the analysis against samples that were sourced from semen. A more appropriate threshold would be to look at the average coverage of the MT relative to the average of the autosomes. What you will find is that tissue sourced from semen will have similar MT coverage to the autosomes while tissue sourced from anything other than semen will have exceedingly high MT coverage. Regardless of coverage, you can assemble the MT reads from all of the samples and then perform an evaluation of MT genome completeness versus coverage and set some (non-arbitrary) threshold for what samples you use for downstream analysis. In summary, I think you may be leaving information on the table by setting this arbitrary threshold.

Estimation of effective population size and divergence time

It is unclear which variants were used and how the phasing was done. Both of these details need to be fully documented since both impact the downstream analysis.

Supplementary Fig. 4 & 5

This shows the results and the percentages for the first and second PCs are very small, especially given the number of samples. Supplemental Fig 5A shows that the percentages get even smaller when PCA is performed on the indicine samples only. This data should be better filtered to get a more informative PCA. This raises further questions related to my comments under the PCA section. I'm not convinced that this was done properly and therefore the results and interpretation are suspect.

Main manuscript which I'll just refer to line numbers.

L102 Change disposal to dispersal.

Line 182 Fig.2. Personally, I hate these figures. They look pretty but show nothing of substantive value because of the resolution (genome coordinates). It is up to the authors as to whether to include it but I feel that it is a waste of space.

L182-183 It is unclear why this is important or even relevant. Differentially expressed genes implies that a regulatory variant has been selected differently between the two groups of animals. However, a coding region variant may have been selected differently between the two groups and not change expression levels. What if you would have selected a different set of tissues? My point is that this analysis does nothing to further define what may be the underlying cause of the region to be differentially selected between the two groups (assuming the region truly is a selection target) and simply adds noise to the discussion. I would argue that this same observation could be made for a large number of regions randomly selected from the genome. However, without quantifying this, you are making a lot of assumptions to relate gene expression differences to putative selection regions.

L185 It appears to me that this is actually two regions. In fact, you list it as two regions but

you describe it as a single region. This needs to be reconciled.

L192-198 All of this is speculation. While I am not opposed to some level of speculation, I am opposed to adaptive storytelling. If you truly identified a functionally plausible gene, and you believe your underlying sequence data, then you should be able to identify a plausible candidate mutation (or mutations) to explain your data. If you are going to state this, then why not do the follow-up analyses to try to identify the actual mutation(s)? I believe a deep dive into this would be far more valuable than what was done.

L242-243 The SEMA3F Val650Ala variant appears to be 22:50162746 which is at frequency 0.033 in the 1000 Bulls Run9 data and is at frequency of 0.034 in my own UMAG1 data based on ~5500 samples. There are a lot of other protein altering variants in this gene, this just happens to be one of them. It is unclear how you arrived at this particular variant. I think the manuscript would be significantly improved if the authors tried to dissect some of these and do a more thorough analysis rather than simply list gene names.

L262-265 I previously listed my concerns with this analysis. Adding to this, for almost all other analyses you used sliding windows. However, here you state that you used non-overlapping windows, why?

L267 Sup. Figs 18-21. For Supp. Fig 19 has banteng in one part but gaur in the other. This needs fixed.

For supp Fig 18 & 19, this looks like a lot of random noise to me and I suspect it has something to do with the banteng/gaur N=2 issue. Were consecutive windows merged? The size of these regions should be informative for the timing of the introgression but nothing was mentioned about this. I suspect due to the previous issues I've already raised.

For supp Fig 20 & 21, how was a p-value calculated? I could not find this.

L281 Exactly how does this provide validation?

L292-294 What is the call rate for these 11 loci in the raw data? Can you rule out any effect imputation may have had in this region? This is the most compelling evidence shown thus far. I would recommend a deeper dive in this region to try to further strengthen your inference.

L318 Supp Fig 45 legend says 2 panels but there is only 1. Fig 46 shows N_e increasing in recent times which is contrary to everything we know about cattle demography. Perhaps this relates to my next point.

L325-344 This paragraph leads off with MSMC but it is unclear to me how you can use the Y and MT in an MSMC analysis when the underlying model is based on recombination?

L402-446 This is all speculation. Again, some is useful but this is 1.5 pages. I would recommend that a deeper dive into any one or two of these would be more valuable to the

reader that speculating on all of them.

L439-440 Proposes that EAI cattle may have introgression from a Bos-like ghost species as an explanation for hemoglobin-related genes. Since hybridization and introgression from known Bos species is difficult, and these large numbers of divergent sequences appear in only this family of genes, is it possible that this 'ghost' group is a lost population of EAI-like Indicine cattle, or that these mutations were specific to the extant EAI clade without any introgression?

Methods

L491-493 were duplicate reads marked or removed? The 1000 Bulls spec is for them to be marked. I just want to be sure that what is stated is what was actually done. Along those lines, the spec has indel realignment and BQSR but that is not stated in the manuscript. Please accurately specify what was actually done.

L494 "...depth (for all individuals) > 1/3× and < 3×..." I have no idea what this represents, please clarify.

L500 Please specify BEAGLE parameters that were used. If defaults were used, then state that. Additionally, you specify SNPs here, were only SNPs used or did you also use indels. Please specify. Did you make any attempt to evaluate the accuracy of imputation? If so, you should state that. If not, I would encourage you to evaluate this and include this information in any filtering that you perform.

L502 This is based on an annotation version, in which case you should specify the exact file or annotation version that was used to make it reproducible.

L547 I've already discussed the multiple testing issue and choice of $p < 0.005$.

L557 Introgression analyses... How might a MAF threshold of 0.01 filtering affected these analyses?

L573 Same comment about the GTF version.

L583 Already commented on the Y chromosome imputation.

Figure 3 panels B-C and D-E appear to be switched relative to the legend. 10kb sliding window appears to be different than what is described in the M&M.

I waive anonymity for all manuscripts and grants that I review and sign my reviews, Robert Schnabel.

**Response to Reviewers' comments**

**[Comment of Reviewer #1]**

General comments

Chen et al. present the results of whole-genome resequencing of 355 indicine cattle genomes
and 141 taurine cattle genomes from 57 and 17 populations, respectively. The quantity and
quality of the material studied here, the methods used and the results obtained are of a high
standard and deserve publication in a high-ranking journal. However, the main problem
concerns the core zebu breed group and the outgroups. Regarding the core Zebu group: The
Zebu cattle breeds from India were missing in this study. The surrogates used for this purpose
are Gir, Nellore and Brahman, which were bred outside India for centuries in new environments
and with sporadic interchange with *Bos taurus* breeds. Introgression of *Bos taurus* into the WGS
of Gir, Nellore and Brahman used here is neither ruled out nor investigated by the authors.
Brahman in particular is known to be a synthetic breed with *Bos taurus* content. Many countries
such as Sri Lanka, Pakistan and Nepal (SAI group) use *Bos taurus* breeds to improve local
cattle breeds. Therefore, the purity of any reference used for introgression studies should be
tested first.

*Response: Thank you for this valuable comment, we apologize that we did not provide a clear*
*description of the reference populations used for introgression studies.*

*In this study, we used Gir, Nellore, and Brahman breeds, but they were grouped as American*
*indicine (AMI) and were not included in the South Asian indicine (SAI) group. Cattle in Sri*
*Lanka, Pakistan and Nepal are authentic SAI zebu breeds.*

*In our study, for the reference groups used for introgression analysis, we selected only 40 core*
*and pure indicine cattle that are included and verified in the results of admixture analysis. We*
*provided this information in Supplementary Table 1. We selected indicine individuals without*
*taurine ancestry. We apologize for not describing this in previous version of our manuscript.*
*According to your suggestions, we also added f_3 statistics to confirm that the selected indicine*
*cattle have a pure genomic background. Our results showed that the reference SAI cattle did*
*not carry either taurine or banteng/gaur ancestries.*

**[Comment of Reviewer #1]**

With regard to outgroups: Besides bison and yak, two Bantengs and two Bours are used for
important introgression analyses in this study. Again, there is a possibility of introgression of
cattle into Banteng and/or Bours. This possibility has not been ruled out and has not been
investigated. There are some unpublished results confirming cattle DNA segments in Bantengs.
I do not have comparable information for Gaur, but this is possible. As shown in the study
analysing cattle introgression in Yak, RFMix can produce false positive signals if the reference
population is itself introgressed.

*Response: Thank you for this valuable comment.*

*We totally agree with you on this point. Introgression of zebu into banteng or gaur has also*

*been documented (Wu et al., 2018. Ref.15; Gao et al., DOI:10.1038/s41598-017-16438-7). To*
*address your concern, we tested for introgression of cattle into banteng and gaur. The f_3*
*statistic showed that there was no introgression into the banteng or gaur samples included in*
*this study.*

*We added this information to the Supplementary Information. We further used the U20 statistic*
*and phylogenetic trees of specific genes to verify this point, which in our view provided*
*compelling evidence for introgression (Fig. 5D, Suppl. Figs. 16 to 28) and clustering of East*
*Asian indicine haplotypes within the banteng or gaur haplotypes.*

**[Comment of Reviewer #1]**

The next problem is the sample size of the outgroups (Banteng and Guar) in the introgression
analysis. I expect that the relatively small sample size of the outgroups (2 samples each) will
affect the performance of RFMix and the determination of ancestral alleles and their frequencies
in the U20 and U50 analyses. In fact, the original paper of RFMix already describes how the
sample size of the reference panel affects its performance (<https://www.ncbi.nlm.nih.gov/pmc/articles/PMC3738819/figure/fig6/>). How did the authors determine the robustness of their
results?

**Response:** *Thank you for this valuable comment.*

*Of course, we would like to have more samples, but for wild animals, this is very difficult*
*because of CITES regulations. However, we have added six banteng and three gaur samples to*
*increase the number of samples of our reference group. And we reanalyzed the introgression*
*from banteng or gaur using U20 and RFMix analyse.*

*To confirm banteng or gaur introgression, we also analyzed the tree topologies of banteng or*
*gaur fragments and their homologous sequences in other bovine species belonging. This*
*indicated unambiguously a banteng or gaur origin of the introgressed segments.*

**[Comment of Reviewer #1]**

The results of the *D*-statistics are confounded by the structure of the ancestral population as
well as incomplete lineage sorting. I recommend that the authors perform other independent
approaches, e.g. those based solely on branch length and tree discordance, to validate these
results.

**Response:** *Thank you for this valuable comment.*

*We have added more approaches to identify and verify the introgressions. We first ruled out*
*introgressed fragments that were likely caused by incomplete lineage sorting (ILS) according*
*to a probability calculation (see Supplementary Note 4). We also analyzed the distribution of*
*tree topologies of banteng or gaur fragments and their homologous sequences in other bovine*
*species. This confirmed the banteng and gaur origins of the introgressions (Supplementary Fig.*
*6).*

**[Comment of Reviewer #1]**

How did the authors determine the optimal number of migration edges in the TreeMix? Ideally,
they should also include the matrix of residual values to show how integrating migration edges
improved the % of variance explained.

*Response: Thank you for this valuable comment. We added a matrix of residual values in*
*Supplementary Fig. 14. We also added the optimal number of migrations using OptM.*

**[Comment of Reviewer #1]**

How did the authors determine the expected length of introgressed haplotypes? How did they
make sure that it is significantly different from the expected length because of its shared
ancestral origin? Does the length of introgressed haplotypes tell us anything about the age of
introgression?

*Response: Thank you for this concern.*

*According to your suggestion, we have added more banteng and gaur samples for inferring*
*their introgressions. In our revised version, we modeled the expected length of ancestral*
*sequence shared by indicine cattle, banteng and gaur on the basis of incomplete lineage sorting*
*(ILS). The expected length of a shared ancestral sequence is $L = 1 / (r \times t)$, where r is the*
*recombination rate of 1.23 cM/Mb (Weng et al. 2014), and t is the branch length between cattle*
*and Asian wild bovine species (banteng and gaur). We used a generation time of 6 years and a*
*divergence time of 1000 kya (Wu et al. 2018). We calculated the probability of ILS. The*
*probability of a shared haplotype length derived from ILS according to the algorithm is $1 -$*
*GammaCDF (m , shape = 2, rate = $1/L$), where GammaCDF is gamma distribution function.*
*We ruled out the introgressed fragments < 980 bp that were likely caused by ILS with a*
*probability < 0.05. We realize that this calculation, although according to the state-of-the-art,*
*assumes a fixed recombination rate and does not account for the plausible effects of divergence*
*between maternal and paternal haplotypes on the probability of recombinations. For this*
*reason, we did not try to infer an age of the introgressions.*

**[Comment of Reviewer #1]**

Each figure and table, including supplementary figures and tables, should be considered as a
stand-alone object, i.e. the readers should be able to understand it without reading other objects.
For this reason, the legend of the figure or table must be informative enough to explain the main
message of this stand-alone object. This is not the case. I make a specific comment for one
figure only (mentioned in the comments below), but it applies to all and is therefore a general
comment.

*Response: Thank you for this specific comment.*

*We have revised all supplementary figures and tables to ensure that could be easily understood.*

**[Comment of Reviewer #1]**

Specific comments

Line 87-88, “..resilient to local pathogens”: consider replacing with tick resistance as also
mentioned in the reference paper cited here.

***Response:** Thank you for this suggestion! We have corrected this sentence as suggested.*

**[Comment of Reviewer #1]**

Line 106-108: Please keep in mind that there is a small but significant Bos indicus introgression
also into almost all south-eastern European cattle breeds up to the southern Alps
(<https://doi.org/10.1186/s12711-020-00560-8>).

***Response:** Thank you for this valuable comment!*

*We have added this information to the Introduction as follow: “Modern DNA analyses have
well documented the male-mediated indicine admixture into African taurine cattle in the eastern,
western and southern areas of the continent ^{7,11} and small but significant indicine introgression
into almost all southeastern European cattle breeds ¹²”.*

**[Comment of Reviewer #1]**

Line 108-109, “...Gujarat in Western India”: While the reference paper do discuss the
hypothesis about possible wild introgression in southern Indian zebu, the same is not mentioned
for Gujarat, consider rewriting this sentence.

***Response:** Thank you for this suggestion! We have deleted this sentence for a rigorous quotation.*

**[Comment of Reviewer #1]**

Line 144-145, “thirteen whole genome sequencing data...”: The authors should mention here
the number of WGS used for each species separately.

***Response:** Thank you for this specific comment! We have added the number of whole genome
sequences used for each species as follows in the revised version: “We also used sequencing
data of 22 whole genomes from six other bovine species, including two bison, two wisent, five
gaur, eight banteng, two yak, and two water buffaloes, as outgroups or for introgression
analysis.*

**[Comment of Reviewer #1]**

Line 166-167, “...the PCA and a phylogenetic tree..”: The authors should describe here how
SWCI fits into the overall phylogenetic reconstruction.

***Response:** Thank you for this valuable comment!*

*We have revised this part and added this information to the description as follows: “SWCI is
genetically in an intermediate position between SEAI and EAI”. According to the comments of
Reviewer 2, we reanalyzed the SNP data for structure analysis, and we redefined the population
ancestries. The PCA and phylogenetic tree almost completely separated the three indicine
geographic lineages of SAI, AFI, and EAI. SWCI was in an intermediate position between SEAI
and EAI.*

**[Comment of Reviewer #1]**

Line 176-178, “We identified candidate selected genomic regions...”: Why author only
combined the three indicine groups (SAI, AFI, EAI) and left the other two (SEAI and SWCI)
for this analysis?

**Response:** *Thank you for this specific inquiry!*

*We reasoned that the SEAI and SWCI have hybrid SAI-EAI ancestries, so we did not select these*
*two groups. Thus, we believe that the three indicine groups of SAI, AFI, and EAI adequately*
*represent the indicine cattle ancestry.*

**[Comment of Reviewer #1]**

Line 175ff: The authors use reverse genomics to detect candidate genes. They constructed two
groups of animals/populations: 1) taurine and 2) SAI, AFI and EAI indicine populations. Both
groups lived in environments that differed by a variety of factors, including temperature. The
authors found numerous significantly differentiated regions and attribute them all to adaptation
to high temperatures. The idea of reverse genomics is to find the outliers and then look for
possible causes. Only after some causality analysis can we conclude that some of the outliers
are an adaptation to high temperatures or something else. The authors should consider
significant selection signatures as anonymous candidates. These anonymous candidates are
subject to positive selection into a group of animals that share a specific environment. These
anonymous candidates could, but need not, be caused by one specific environmental factor.

**Response:** *Thank you for this valuable comment and suggestion!*

*We agree that our selection signatures do not necessarily indicate a specific adaptation to high-*
*temperature. We have modified the headings of Lines 174 and 175 as follows:*

*“The ancestral adaptation of indicine cattle*

*Ancestral environmental adaptation of South Asian indicine”.*

*Furthermore, this text is added after line 198 as follows: “However, further research is*
*warranted to test their role in heat adaptation or other differences between indicine and taurine*
*cattle.”*

**[Comment of Reviewer #1]**

Line 182-183, “In addition, the expression level...”: As the tissue and animals (with respect to
gender and age) are quite heterogenous, I will take this result with pinch of salt, moreover, how
many other genes were differentially expressed and not under positive selection?

**Response:** *Thank you for this valuable comment!*

*To address your concerns and to avoid misleading results, we have deleted this part.*

**[Comment of Reviewer #1]**

Line 192-195: Again, the authors link LIPH to a possible contribution to heat tolerance in

indicine cattle by controlling hair length and/or thickness. However, we (readers) are not
informed whether indicine and taurine cattle differ in hair length and thickness? I cannot find
any such information in this manuscript.

**Response:** *Thank you for this valuable comment! We have added a description of the difference*
*between taurine and indicine cattle in the Introduction as follows:*

*“Indicine cattle are recognized by their thoracic hump, low metabolic rate, many large sweat*
*glands, large skin surface, and short smooth coat⁵.”*

**[Comment of Reviewer #1]**

Line 202ff: The SAI cattle group consists of breeds from Sri Lanka, Pakistan and Nepal (a very
broad area), but all outliers are defined as candidates for adaptation to semi-arid environmental
stress.

**Response:** *Thank you for this thoughtful comment! We agree that our selection signatures do*
*not necessarily indicate a specific adaptation. We have modified the headings of Lines 174 and*
*175:*

*“The ancestral adaptation of indicine cattle*

*Ancestral environmental adaptation of South Asian indicine”*

*After line 199 we added a sentence as follows: “However, further research is warranted to test*
*their roles in heat adaptation or other differences between indicine and taurine cattle”.*

**[Comment of Reviewer #1]**

Line 223, “The genome-wide analysis shows...SAI origin”: This is well-established fact;
therefore, I would remove it or declare as confirmation.

**Response:** *Thank you for this valuable suggestion. We have removed this sentence.*

**[Comment of Reviewer #1]**

Line 319-321: With regard to mtDNA distribution, a comparable situation can be observed in
AFI and EAI. However, the presence of haplogroup T in Africa was not interpreted as T having
emerged as a new haplogroup during the indicine westward migration, but as a haplogroup that
was already present in the area before the indicine introgression. The authors should consider
both the time of coalescence and the time of migration when drawing their conclusions.

**Response:** *Thank you for these important comment and suggestion!*

*Indeed, several studies have shown that the mtDNA of African indicine cattle originates from*
*taurine cattle that were already present in Africa prior to the introduction of indicine cattle (see*
*lines 455-458; ref. 7 (Kim et al., 2020, The mosaic genome of indigenous African cattle as a*
*unique genetic resource for African 691 pastoralism. Nature Genetics 52, 1099-1110); and ref.*
*11 (Kim et al., 2017, The genome landscape of indigenous African cattle. Genome Biology 18,*
*34).*

**[Comment of Reviewer #1]**

Line 346-361: Samples originating from India and sampled there are missing in this study. This
could be a reason why the west-east cline is missing and an abrupt transition is observed. I did
not understand this section of the text. There are several problems in the text itself and in the
corresponding figures. The legends of the figures are generally not informative enough. Let's
just take Supplementary Figure 17 as an example. The reader cannot understand this figure from
its legend. Even more, there are many trees labelled with the same A, B, C,.... and we do not
know which tree represents which haplotype, etc. I also do not know what to conclude from
figure 50, i.e. the legend does not help me to understand the message from line 355-357.

**Response:** *Thank you for this valuable comments!*

*We apologize for our mistakes! We have revised all legends of the figures and Supplementary*
*figures to ensure that they are sufficiently informative.*

*For Supplementary Figure 16, we provide tree topologies of banteng and gaur fragments across*
*species belonging to the bovine species to confirm banteng and gaur introgression.*

*For Supplementary Figure 17, we have revised the figure legend in order to explain that it*
*provides a geographic contour map of banteng/gaur introgression proportions in 16 East Asian*
*indicine (EAI) breeds. The proportions of banteng and gaur introgressions were calculated by*
*RFMix. The proportions of each breed were plotted according to its geographic origin and*
*visualized using the ArcMap component of the ArcGIS software suite. EAI cattle in the*
*southeastern coastal region of China show the highest level of banteng and gaur ancestries. We*
*also modified the legends of Supplemental figure 50 (now is Supplementary Fig. 29) in order to*
*explain how the geography with three large rivers in a mountainous regions impede the gene*
*flow between SEAI and SWCI.*

**[Comment of Reviewer #1]**

Line 414: (in brackets)???

**Response:** *Thank you for this specific comment! We have deleted "(in brackets)".*

**[Comment of Reviewer #1]**

Line 447-458: Two recent and comprehensive studies on paternal and maternal haplogroups in
cattle could be helpful here (DOI: 10.1111/eva.13315 and doi: 10.1111/age.13104).

**Response:** *Thank you for drawing our attention to these interesting papers on the diversity of*
*mitochondrial and Y-chromosomal variations in cattle. We now refer the first paper on taurine*
*mtDNA in line 124. Since the second paper of Escoufflaire et al. (2021) described Y-*
*chromosomal variation in French taurine cattle, we do not refer it due to its irrelevance to our*
*analyses.*

**[Comment of Reviewer #1]**

Line 459-470: This can be understood to mean that East Asia was a land without cattle, so that

Bos indicus spread with the single matriline I1a in this empty area. This was also the case with
the spread of domestic sheep in Europe. However, for sheep, there is archaeological evidence
for a sheep-less Europe before the Neolithic. Do you have any evidence that East Asia was a
cattle-less region before the spread of Bos indicus? Africa was not cattle-less either, and as a
result of paternal introgression we only observe haplogroup T. Please clarify.

**Response:** Thank you for this thought-provoking comment! In fact, East China harbored taurine
cattle prior to the arrival of indicine DNA (Feliuss et al., 2014, doi:10.3390/d6040705; Zhang
et al., 2018; doi.org/10.1186/s12863-018-0705-9). Approximately 20% of South Chinese cattle
still contain taurine mtDNA with high diversity (Gao et al., DOI:10.1038/s41598-017-16438-
7; Xia et al., 2018; doi: 10.1111/age.12749). This is now indicated in the text as follows:
“Indicine cattle may have entered East Asia between 3,500 and 2,500 YBP well after the arrival
of taurine cattle (ref. 10, Feliuss et al., 2014, doi:10.3390/d6040705; Zhang et al., 2018;
doi.org/10.1186/s12863-018-0705-9; Gao et al., doi:10.1038/s41598-017-16438-7; Xia et al.,
2018; doi: 10.1111/age.12749)”.

**[Comment of Reviewer #1]**

Line 470-475: The reason behind this could be lack of sampled population along the inland
trading routes from South to East Asia. Therefore, the integrated genetic analyses as mentioned
in the paper may have difficulty supporting inland trading even if this is correct.

**Response:** Thank you for this valuable comment!

We have revised our conclusion along this point as follows: “but our combined uniparental and
autosomal data support a coastal route for the first migration wave to Southeast Asia as the
main entry point of indicine cattle into East Asia”.

This is also supported by the geographic situation with the Himalayan Mountain range as well
as the rivers and mountainous areas in Southwest China, which does not favor an inland
migration of cattle (Supplemental Figure 39).

**[Comment of Reviewer #1]**

Line 558-559, “Treemix and Dsuite...”: Dsuite is a tool and not any method, please clarify this.

**Response:** Thank you for this specific comment! We have revised this sentence to include
TreeMix and the D statistic.

**[Comment of Reviewer #1]**

Supplementary table 1:

If possible, the country of origin should also be mentioned, also verify that breeds with Name
“SriLanka” exists.

**Response:** Thank you for this specific comment! We have added the countries of origins to
Supplementary Table 1 and added more information for all breeds. We have added the local
name of Sri Lankan cattle.

**[Comment of Reviewer #1]**

Supplementary table 2:

Haryana is declared as an Indian breed and the sample used here was sequenced in Chen et al.
2018 and should be from India. We know that India is very restrictive when it comes to foreign
use of biological resources. There is no Indian collaborator in Chen et al. 2018, so I do not see
any legal way to analyze Indian biological material outside India or without an Indian
collaborator. Was this Haryana cattle sampled in India or elsewhere? Please clarify and correct
if necessary in Supplementary Table 2.

**Response:** *Thank you for this specific concern! Dr. Daniel G Bradley was a collaborator in*
*Chen et al. 2018 and provided with Indian samples of Haryana, Sahiwal, and Tharparkar. These*
*animals have been in his collection since the early 1990s, and they were sampled at the Indian*
*Veterinary Research Institute, Izatnagar-243122, District Bareilly, Uttar Pradesh, India. The*
*first paper including these samples (Anim Genet. 1994; 25(4): 265-71. doi: 10.1111/j.1365-*
*2052.1994.tb00203.x.) did have a collaborator, D S Balain, from India.*

**Reviewer #2 (Remarks to the Author):**

**[Comment of Reviewer #2]**

This manuscript is interesting and provides data from a unique sample of individuals that had
not been previously characterized. Overall, I find the manuscript well written, although it
appears to have been done in parts by different groups of individuals with different writing
styles and slightly different nomenclature. This is inevitable with a project of this size. I would
encourage the corresponding authors to go through the entire manuscript to make it more
uniform. I do have a number of concerns related to the details that were presented, or not
presented, that make a final determination not possible at this time. I always start my reviews
with the materials and methods but since there was such a substantial amount of information in
the supplemental information, I will start there. Unfortunately, the supplemental document does
not have line nor page numbers, so I'll quote specific lines.

**[Comment of Reviewer #2]**

Whole-genome resequencing

Additional detailed information on the mapping rate and sequencing depth are provided in
Supplementary Table 1” In Sup. Table 1 you use the SRR* ID as the sample. This is incorrect
because these are run IDs. You should use the BioSample ID. I downloaded this table and
compared these SRR to the NCBI database and you have two duplicate samples. SRR2016752
and SRR2016754 are the same SAMN3387026 individual. With only a single duplicate, it is
unlikely that any of the results would have been significantly affected. However, this table needs
to be corrected to present the BioSample and if data reanalysis is performed, one of these two
should be removed.

**Response:** *Thank you for your careful review! We have removed SRR2016752 and reanalyzed*

*the results. We also provided the BioSample ID for all samples in Supplementary Table 1 and*
*removed one duplicated sample.*

**[Comment of Reviewer #2]**

Population genomic analyses Genetic diversity

“Runs of homozygosity (ROH) were identified using the --homozyg option implemented in
PLINK v.1.9, which slides a window of 50 SNPs (--homozyg-window-snp 50) across the
genome.” This is a very small window considering the density of markers. It is well recognized
that results are sensitive to settings for this analysis (see [https://doi.org/10.1186/s12864-020-](https://doi.org/10.1186/s12864-020-6463-x)
[6463-x](https://doi.org/10.1186/s12864-020-6463-x)). I question whether or not the settings used were appropriate. Specifically, by using too
small of a window size and other inappropriate settings, far too many ROH are detected.
Evidence of this can be seen in Supplementary Fig. 2 where there are a significant number of
samples that have estimated ROH totaling greater than 1/3 of the genome and a large number
of samples with ROH greater than half the genome. This is unrealistic. This analysis needs to
be redone with appropriate settings, which may need to be determined empirically given the
data. It would also be useful at this time to examine any correlation
between average genome coverage and number/length of ROH as one would predict that lower
coverage samples will have heterozygous sites undercalled which may manifest as ROH when
in fact there are not ROH if heterozygous positions were called accurately.

**Response:** *Thank you for these valuable concern and comment!*

*We reanalyzed the ROHs. We first filtered samples with the mapping depth < 10 × or 3 × genome*
*coverage < 90% and used 331 individuals for ROH analysis. We then used imputed SNPs to*
*detect ROHs with a minimum length of 100 kb and containing at least 50 SNPs using PLINK*
*v1.9 software. Additionally, three heterozygous SNPs were allowed per ROH.*

*First, the effect of scanning window size (--homozyg-window-snp) was investigated by*
*varying this setting from 10 to 200 SNPs (step size of 10 SNPs). The other parameters were set*
*to a minimum density threshold (50 SNPs), a large gap (1000 kb), a minimum length (50 kb), a*
*maximum number heterozygous SNPs in a scanning window (3), and a scanning window*
*threshold level (0.05). The results suggested that increasing scanning window size led to the*
*decrease in both number and total length of the estimated ROHs (Fig. 1), however, there are*
*still too many ROH; When the scanning window size was 100 and 150 Mb, the largest numbers*
*of ROHs were 7914 and 6735, respectively.*

 Fig. 1 Relationship between the number of runs of homozygosity (ROHs) and the total length (Mb) of ROHs
 for all individuals from each cattle geographic group. Each dot represents an individual.

*Therefore, we have increased the --homozyg-kb parameter (minimum length set to 100 Mb)*
 *to filter the small ROHs. The other parameters were set to a minimum density threshold (50*
 *SNPs), a large gap (1000 kb), a maximum number heterozygous SNPs in scanning window (3)*
 *and a scanning window threshold level (0.05). The results show that our settings get the*
 *expected number (maximum number is 4570) and total length (maximum length is 1,423,160*
 *Mb) of ROHs, which are consistent with the results of Kim et al. (Kim et al. 2020). We used*
 *these results in the Supplementary Fig. 2. And these results shows that the level of inbreeding*
 *measured by ROHs was lower in indicine cattle than in taurine cattle.*

 Fig. 2 Runs of homozygosity (ROHs) patterns of all individuals from each cattle geographic groups. (A) the

scanning window size is 50 SNPs. (B) the scanning window size is 100 SNPs. The minimum ROH length
was set to 100 Kb for excluding short ROHs.

*According to your suggestions, we introduced genomic coverage as an indication of the*
*validity of the ROH analysis (Meyermans et al. 2020). The scanning window size setting (--*
*homozyg-window-snp) was investigated by varying this setting from 10 to 200 SNPs (step size*
*of 10 SNPs). The unchanged parameter set to a minimum density threshold (50 SNPs), a large*
*gap (1000 kb), a minimum length (50 kb), a maximum number heterozygous SNPs in scanning*
*window (3) and a scanning window threshold level (0.05). Consequently, genome coverage was*
*higher than 99% for all breeds, which means that the given settings allowed ROH detection for*
*more than 99% of all autosomes (Fig. 3).*

Fig. 3 The effect the scanning window size on F_{ROH} (green) and genome coverage (red) estimates for six
breed/populations. (A to C) three taurine cattle breeds (Jersey, Mongolian, and Yanbian). (D to F) three
indicine cattle breeds (Longlin, Weizhou, and Red Sindhi).

**[Comment of Reviewer #2:]**

Principal component analysis (PCA)

“We removed “all LD” using...” First, please rephrase this. You did not remove “all LD”. I have
the same objection here to the parameters used. First, this is a very small window and probably
left a significant number of loci with $r^2 > 0.20$ which are more than 50kb apart. Given
everything that we know regarding LD in bovine genomes, you should empirically find the
appropriate setting for this analysis to achieve the desired thinning. This does not have to be
done genome-wide and one could simply use 10Mb from any autosome to estimate what the
appropriate parameters should be. My recommendation would be to use chr25 since we know
there are no assembly issues. For a 10Mb region, calculate *all* the pairwise LD values using
‘-r2 yes really’ and examine the distribution of these values. This figure can be included in the
supplement to provide justification for the settings chosen. As the manuscript is currently
written, the settings are arbitrary and likely impacted the results and interpretation.

**Response:** Thank you for this suggestion!

*According to your suggestions, we have calculated the LD decay of the cattle genome, and we*

provided these results in Supplementary Fig. 4. The half value of LD decay (r^2) is 0.11, so we
performed LD-based pruning for the genotype data using PLINK v1.9 with '-indep-pairwise 50
10 0.1'.

*Supplementary Fig. 4 Linkage disequilibrium (LD) decay in 29 autosomes of all 495 cattle. The*
*half value of LD decay is 0.13.*

*For PCA and admixture analysis, we used the '-indep-pairwise 50 10 0.1' and '-indep-pairwise*
*20 10 0.1' options to perform LD based pruning for the genotype data and used these data for*
*PCA and admixture analysis to compare the influence of different parameters on PCA and*
*admixture analysis (Supplementary Fig. 5).*

*Using these data, the results of PCA and admixture were similar. Therefore, we selected the '-*
*indep-pairwise 50 10 0.1' option for **LD pruning**.*

Supplementary Fig. 5 Principal component analysis (PCA) of all 495 cattle, illustrated by PC1 against PC2 (A) and PC1 against PC3 (B). A total of 2,996,368 LD pruned SNPs were used for PCA with the parameter ‘-indep-pairwise 50 10 0.1’.

Supplementary Fig. 5 Principal component analysis (PCA) of all 495 cattle, illustrated by PC1 against PC2 and PC1 against PC3. A total of 7,192,063 LD pruned SNPs were used for PCA with the parameter ‘-indep-pairwise 20 10 0.1’.

[Comment of Reviewer #2:]

“PCA was performed on the genome-wide unlinked SNP dataset...” Please change the word “unlinked” throughout the manuscript. You do not have any unlinked variants. Perhaps use “LD pruned” as a replacement.

Response: Thank you for this specific suggestion! We used “LD pruned” to replace “unlinked” throughout this revised version of our manuscript.

[Comment of Reviewer #2]

Structure analysis

"...200 bootstrap replicates were performed..." Why 200? Why not 10, or 10,000. Is the default
of 200 appropriate for your data? "...used to determine the optimal ancestry number..." Exactly
how did you determine the optimal number? There is quite a bit of literature on this, but nothing
was stated or cited.

**Response:** *Thank you for this valuable comment!*

*We reanalyzed the population genetic structure in Admixture using new data, and we used the*
*default setting, so bootstrap replicates were not used. We used the delta K method to choose the*
*optimal K, and we added the K values to the Supplementary Table 4. And a sentence is added*
*as follows:*

*"ADMIXTURE v.1.3.0 was used to quantify genome-wide admixture among cattle*
*breeds/populations⁸² and run for each possible group number (K = 2 to 8), where K is the*
*assumed number of ancestral populations. The delta K method was used to choose the optimal*
*K⁸²".*

**[Comment of Reviewer #2]**

Neighbor-joining (NJ) and maximum likelihood (ML) phylogenetic trees

"...A window size of 1,000 SNPs was used to account for linkage disequilibrium..." This
should be unnecessary if you properly LD pruned in the first place. This brings up another issue
I have with the manuscript. You have many different analyses and it is difficult to tell exactly
which set of loci you are using for which analysis. In this section, it appears that you are using
the "raw" 67M variants. However, it seems to me that if the objective is to "identify closely
related individuals" then the LD pruned dataset from the PCA section would be appropriate.
However, it appears that these two analyses used different filtering. My recommendation would
be to provide a supplemental table that lists the major analyses performed and exactly which
data manipulations were done for each analysis. Since data manipulations may impact one
analysis but not another, it would be helpful to the reader to understand exactly what was done
and which analyses used the same loci.

Furthermore, you can specify the total number of loci in this table for each dataset.

**Response:** *Thank you for this valuable suggestion!*

*We apologize for the misleading description. We have clarified the datasets used for NJ tree*
*and ML trees as follows: "To identify relationships among individual cattle, a total of*
*67,162,108 autosomal SNPs were used to construct a NJ tree with PLINK v.1.9 based on the*
*matrix of pairwise genetic distances⁶ (Fig. 1). FigTree v.1.4.3 10 was used to visualize the NJ*
*tree. Then, we inferred a population-level phylogeny using the ML approach implemented in*
*TreeMix¹¹. We performed LD-based pruning for the genotype data of all 495 cattle and one yak*
*using the --indep-pairwise 50 5 0.1 option of PLINK v.1.9⁶. A total of 15,228,801 LD pruned*
*SNPs and the "-global -root yak" parameter were used to generate the ML tree (Supplementary*
*Fig. 8)." We have summarized all information on datasets used for different analyses in the*
*Supplementary Table 3.*

**[Comment of Reviewer #2]**

Detection of selection signatures shared by all indicine populations
"...were calculated for 50-kb windows with a 20 kb step across the autosomes..." Line 907 of
the main manuscript says that a 100kb window was used, which is it? I have issues with what
is specified here. Typically, these types of analyses are performed where the window size is an
integer multiple of the step size. For instance 25 kb step and 50 kb window or 20kb step and
100kb window. This needs to be addressed throughout the manuscript and I would recommend
using the same parameters for all analyses that use this type of setting (where appropriate). This
will make comparisons between analyses much easier.

**Response:** *Thank you for this valuable comment!*

*We apologize for the mistakes. Population genetic differentiation (F_{ST}) was calculated using a*
*sliding window approach with windows of 50 kb and a step size of 20 kb.*

*For Line 907 (now Line 825), there was a typo in our manuscript: the text should indicate 50*
*kb and a step size of 20 kb.*

*According to your suggestion, we have checked our methods. For F_{ST} , the $\theta\pi$ ratio (indicine/*
*taurine), F_{ST} , XP-EHH, and PBS, we used 50 kb and a step size of 20 kb. For U20 and U50*
*analyses, we reanalyzed the data and used a 50 kb window and a step size of 20 kb too.*

**[Comment of Reviewer #2]**

"...these windows harbored 117 candidate genes in indicine cattle..." I don't think these 117
genes are specific to indicine cattle and you can probably just say cattle. By saying indicine
cattle here one might think you are implying that these are not in taurine cattle, which isn't the
case.

**Response:** *Thank you for this specific suggestion! We have corrected this sentence and deleted*
*the reference to indicine cattle. The sentence is revised as follows: "These windows harbored*
*117 candidate genes."*

**[Comment of Reviewer #2]**

"...the expression levels of these 117 candidate genes were validated in nine different tissues
of taurine and indicine cattle." I'm not sure exactly what you are validating? The fact that there
are RNA-seq data in those tissues for these annotated genes simply validates that the annotation
is correct.

**Response:** *Thank you for this valuable comment! To avoid misleading results, we have deleted*
*this part from the manuscript.*

**[Comment of Reviewer #2]**

"Candidate genes under selection were defined as those overlapped by sweep regions or within
20 kb of the signals." 20kb seems arbitrary. What justification do you have for choosing 20kb?
Why not 50kb, or 10 kb or simply require the gene to fall within the sweep region and not allow
any overlap. It seems to me that allowing for a gene to be outside your sweep region, by some
arbitrary distance, speaks to your confidence in defining the boundaries of sweep regions. Some

justification should be given for whatever value you choose.

**Response:** *Thank you for this valuable comment!*

*We apologize for the misleading description around this topic. In fact, the candidate genes were*
*taken from the selected regions identified with the three methods, without extending the signal*
*by 20 kb. This sentence has been revised as follows: “The candidate genes selected in all*
*indicine cattle were defined as the genes with overlapped signals in any two of the three*
*selection methods ($\theta\pi$ ratio (indicine/taurine), F_{ST} , and XP-EHH)”.*

**[Comment of Reviewer #2]**

“...were plotted using a 10-kb sliding window...” Again, arbitrary. Why not use the same
window size for plotting that you used for detection?

**Response:** *Thank you for this specific suggestion!*

*The window shows three small regions (Supplementary Figs. 10-12). The region sizes are 0.67*
*Mb, 8.21 Mb, and 0.33 Mb. For the definition of sliding window, we tried different sizes. If we*
*use the detection window (50 kb), there are too few SNPs to draw the graph, and the real signal*
*for the target region will be overlooked.*

**[Comment of Reviewer #2]**

Detection of selection signatures in SAI, EAI, and AFI lineages“...windows of 50 kb and a step
of 20 kb...” Same comment on windows.

**Response:** *Thank you for this specific comment!*

*All the methods used to detect selection signatures were based on calculations using a 50 kb*
*window with a 20 kb step size across the autosomes, including CLR, F_{ST} , and XP-EHH.*

**[Comment of Reviewer #2]**

“...the overlap windows of P values less than 0.005 (Z test) of each method were considered
candidate signatures of selection...” Why 0.005? Again, some justification is warranted. It
seems to me that you have an extraordinarily large number of windows that were tested which
implies that some adjustment needs to be considered for multiple testing. Appropriately
accounting for multiple testing here could significantly change the detected regions which in
turn could significantly impact all downstream inference based on these results. As written, it
is impossible to determine exactly what was done and what impact it may have on the results
or interpretation.

**Response:** *Thank you for this kind concern!*

*P values were estimated based on Z-transformed values using the standard normal distribution*
*and were further corrected by multiple testing using the Benjamin-Hochberg false discovery*
*rate (FDR) method. We hope this justified.*

**[Comment of Reviewer #2]**

“Only pathways or annotations with a Bonferroni-corrected $P < 10^{-2}$ were retained

(Supplementary Tables 7-10).” This should probably be written as $P < 0.01$ to be consistent
throughout the manuscript.

*Response: Thank you for this specific comment. We have revised $P < 10^{-2}$ to $P < 0.01$*
*throughout the manuscript.*

**[Comment of Reviewer #2]**

Introgression analysis

“Phylogenetic analyses of these segments confirmed the banteng or gaur introgression into
specific EAI individuals.” Exactly how does this *confirm* introgression? “Therefore, we used
the statistic U20SAI, EAI, banteng (1%, 20%, and 100%) 20, which was equal to the number
of SNPs within a genomic window where a particular allele was fixed at a frequency of 100%
in banteng but at a frequency less than 1% in SAI cattle or greater than 20% in EAI cattle.” For
both the banteng and gaur analyses you only have two individuals of each species. This means
that you have essentially no ability to estimate allele frequencies in gaur and banteng. This
means that all of your results from analyses of this type are suspect, which in turn means that
inference based on these results is suspect. There are more than two of each of these publicly
available in SRA and your analysis would be significantly strengthened by using what is
available to increase your sample size, which will allow you to better estimate allele frequencies
in these species.

*Response: Thank you for this valuable suggestion!*

*The phylogeny of specific genes provided evidence for introgression (Fig. 5D, Suppl. Figs. 16*
*to 27) and support that introgressed segments of East Asian haplotypes were clustered within*
*bangteng or gaur haplotype groups. According to your and reviewer 1’s suggestions, we have*
*added six banteng samples and three gaur samples, and we recalculated the U20 and U50*
*statistics. We have added the number of whole genome sequences used for each species as*
*follows in the revised version: “We also used sequencing data of 22 whole genomes from six*
*other bovine species, including two bison, two wisent, five gaur, eight banteng, two yak, and*
*two water buffaloes, as outgroups or for introgression analysis.”*

**[Comment of Reviewer #2]**

Paternal analysis “...(i) only present in at least two males but not in females; (ii) hemizygous...”
Are these not represented as homozygous in the actual data? In reality, the males are
hemizygous but they are represented as homozygous when variant calling.

*Response: Thank you for this valuable comment! We apologize for our misleading description.*
*We have revised the text. In fact, no Y-SNP is heterozygous. This sentence has been revised as*
*follows: “Only the SNPs called in the MSY region that met the following criteria were retained:*
*(1) present in at least two males but not in females and (2) no heterozygous site.”*

**[Comment of Reviewer #2]**

“BEAGLE was used to impute missing alleles.” Were samples set to homozygous for
imputation and you simply pretended as if this was an autosome? Exactly how was Beagle run

for Y-specific loci? The underlying model includes recombination rate, which is zero for the Y.
It's unclear to me how this was performed.

***Response:** Thank you for this concern! In our study, we imputed missing alleles under the*
*assumption that they may be belonged to autosomes. In this study, we used imputed data only*
*to construct a median-joining (MJ) network. To avoid misleading results, we reconstructed the*
*network using unimputed data. We deleted this imputation-related part to avoid misleading.*

**[Comment of Reviewer #2]**

Whole mitogenome phylogeny

"...samples with a depth of coverage lower than 100× were disregarded." Why? The depth of
coverage on the MT is a function of the overall average genome coverage and the tissue source
for the DNA. This biases the analysis against samples that were sourced from semen. A more
appropriate threshold would be to look at the average coverage of the MT relative to the average
of the autosomes. What you will find is that tissue sourced from semen will have similar MT
coverage to the autosomes while tissue sourced from anything other than semen will have
exceedingly high MT coverage. Regardless of coverage, you can assemble the MT reads from
all of the samples and then perform an evaluation of MT genome completeness versus coverage
and set some (non-arbitrary) threshold for what samples you use for downstream analysis. In
summary, I think you may be leaving information on the table by setting this arbitrary threshold.

***Response:** Thank you for this valuable comment!*

*You are correct: this biases the analysis against samples that were sourced from semen, as*
*semen will have MT coverage similar to that of the autosomes, which will not be sufficient to*
*assemble full mitogenomes. We apologize for our unclear description. We used stricter criteria*
*for mitogenomes, and we have revised and supplemented our methods.*

*In this study, we focused only on individuals for which mitogenomes could be assembled. We*
*first selected all indicine cattle in our dataset for mitogenome analysis, and then we selected*
*only mitogenomes that were successfully assembled by MIA software and filtered mitogenomes*
*with a gap length > 1 bp. According to these criteria, we compared our results with those of a*
*previous study. We used more criteria, and an additional 13 samples were filtered. A total of*
*329 mitogenomes assembled in this study and 18 reference mitogenomes were used for the final*
*analysis. We reanalyzed all mitogenomes in the manuscript. The results were similar to our*
*previous results.*

**[Comment of Reviewer #2]**

Estimation of effective population size and divergence time

It is unclear which variants were used and how the phasing was done. Both of these details need
to be fully documented since both impact the downstream analysis.

***Response:** Thank you for this concern!*

*We have added more information for the estimation of effective population size and divergence*
*time.*

*The section is rewritten as follows: “The multiple sequential coalescent Markovian model 2*
*(MSMC2) method was used to model the population history of the three core indicine groups*
*(EAI, SAI, and AFI) and to infer historical changes in their effective population size and*
*population separation. We applied this method to all groups with two deep-coverage ($>14 \times$)*
*individuals per group. All sample sets of filtered variant calls were used for imputation and*
*phasing using Beagle v4.1 with default parameters 4, except for “niterations” which was set to*
*10. For each individual, DR2 value in INFO column of the “phase.vcf” file was used to filter*
*SNPs, and SNPs with $DR2 > 0.9$ were retained. We also applied the genome mask as*
*recommended in the documentation of the software. For the calculation of effective population*
*size, the parameter of MSMC2 was set to “msmc2 -t 10 -p 1*2+25*1+1*2 -I 0,1,2,3” and*
*“msmc2 -t 10 -p 1*2+25*1+1*2 -I 4,5,6,7”. For the calculation of population separation, the*
*parameter of MSMC2 was set to “msmc2 -t 8 -P 0,0,0,1,1,1,1 -s -p 1*2+25*1+1*2”. For*
*effective population size inference, two individuals (4 phased haplotypes) from each population*
*were used. The time scale in generation time at $g = 6$ and a mutation rate per generation at μg*
*$= 1.26 \times 10^{-8}$ were used.”*

**[Comment of Reviewer #2]**

Supplementary Fig. 4 & 5

This shows the results and the percentages for the first and second PCs are very small, especially
given the number of samples. Supplemental Fig 5A shows that the percentages get even smaller
when PCA is performed on the indicine samples only. This data should be better filtered to get
a more informative PCA. This raises further questions related to my comments under the PCA
section. I’m not convinced that this was done properly and therefore the results and
interpretation are suspect.

**Response:** *Thank you for this valuable comment! According to your suggestions, we have*
*performed PCA again using new LD pruned data with the parameters “-indep-pairwise 50 10*
*0.1”. The new results are shown in Supplementary Figs. 4 and 5.*

**[Comment of Reviewer #2]**

Main manuscript which I’ll just refer to line numbers.

L102 Change disposal to dispersal.

**Response:** *Thank you for this specific correction! We have corrected “disposal” to “dispersal”.*

**[Comment of Reviewer #2]**

Line 182 Fig.2. Personally, I hate these figures. They look pretty but show nothing of
substantive value because of the resolution (genome coordinates). It is up to the authors as to
whether to include it but I feel that it is a waste of space.

**Response:** *Thank you for this concern! We have deleted Fig. 2 in the main text and moved it to*
*the Supplementary Fig. 9.*

**[Comment of Reviewer #2]**

L182-183 It is unclear why this is important or even relevant. Differentially expressed genes
implies that a regulatory variant has been selected differently between the two groups of animals.
However, a coding region variant may have been selected differently between the two groups
and not change expression levels. What if you would have selected a different set of tissues?
My point is that this analysis does nothing to further define what may be the underlying cause
of the region to be differentially selected between the two groups (assuming the region truly is
a selection target) and simply adds noise to the discussion. I would argue that this same
observation could be made for a large number of regions randomly selected from the genome.
However, without quantifying this, you are making a lot of assumptions to relate gene
expression differences to putative selection regions.

*Response: Thank you for this concern! Accordingly, we have removed this section.*

**[Comment of Reviewer #2]**

L185 It appears to me that this is actually two regions. In fact, you list it as two regions but you
describe it as a single region. This needs to be reconciled.

*Response: Thank you for this correction!*

*We have revised this sentence as follows: "The top selection signatures are in two regions on*
*BTA7, both together spanning 4.46 megabases (Mb) (43.04-44.67 and 50.14-52.97 Mb)."*

**[Comment of Reviewer #2]**

L192-198 All of this is speculation. While I am not opposed to some level of speculation, I am
opposed to adaptive storytelling. If you truly identified a functionally plausible gene, and you
believe your underlying sequence data, then you should be able to identify a plausible candidate
mutation (or mutations) to explain your data. If you are going to state this, then why not do the
follow-up analyses to try to identify the actual mutation(s)? I believe a deep dive into this would
be far more valuable than what was done.

*Response: Thank you for this valuable comment! We agree that our statements are at best*
*tentative, and we have adapted our text as follows:*

*"Seven of the other 75 genes located in the topmost significant sweeps are functionally*
*associated with heart development, blood circulation, DNA damage, and light response.*
*Further research is warranted to test their roles in heat adaptation or other differences between*
*indicine and taurine cattle."*

**[Comment of Reviewer #2]**

L242-243 The SEMA3F Val650Ala variant appears to be 22:50162746 which is at frequency
0.033 in the 1000 Bulls Run9 data and is at frequency of 0.034 in my own UMAG1 data based
on ~5500 samples. There are a lot of other protein altering variants in this gene, this just happens

to be one of them. It is unclear how you arrived at this particular variant. I think the manuscript
would be significantly improved if the authors tried to dissect some of these and do a more
thorough analysis rather than simply list gene names.

***Response:** Thank you for this valuable comment! We have deleted this part from the manuscript.*

**[Comment of Reviewer #2]**

L262-265 I previously listed my concerns with this analysis. Adding to this, for almost all other
analyses you used sliding windows. However, here you state that you used non-overlapping
windows, why?

***Response:** Thank you for this valuable inquiry! To be consistent with the window and step sizes
of other methods, we used the same standards (50 kb window and 20 kb step size) for
recalculation in the revised manuscript.*

**[Comment of Reviewer #2]**

L267 Sup. Figs 18-21. For Supp. Fig 19 has banteng in one part but gaur in the other. This needs
fixed. For supp Fig 18 & 19, this looks like a lot of random noise to me and I suspect it has
something to do with the banteng/gaur N=2 issue. Were consecutive windows merged? The size
of these regions should be informative for the timing of the introgression but nothing was
mentioned about this. I suspect due to the previous issues I've already raised.

***Response:** Thank you for this valuable comment!*

*We have fixed the mistake in the Supplementary Fig. 19 legend (now Supplementary Fig. 18).*

*We reanalyzed the U20 and U50 statistics using increased sample sizes of eight banteng and
five gaur samples, and we merged consecutive windows. As we used only the results of the U20
statistic, we plotted only adaptive introgressed regions with higher frequencies; please see the
results.*

**[Comment of Reviewer #2]**

For supp Fig 20 & 21, how was a p-value calculated? I could not find this.

***Response:** Thank you for this valuable comment! P values were estimated based on Z-
transformed values using the standard normal distribution and were further corrected by
multiple testing using the Benjamin-Hochberg false discovery rate (FDR) method. We have
added this information to the manuscript.*

**[Comment of Reviewer #2]**

L281 Exactly how does this provide validation?

***Response:** Thank you for this inquiry! We apologize that we did not describe this clearly.*

*We used phylogenetic analysis to support the introgression from banteng or gaur into East
Asian indicine cattle. Phylogenetic analyses of haplotypes representing banteng, East Asian
indicine cattle, and other bovine species clustered East Asian indicine cattle with banteng or*

*gaur*, thus confirming *banteng* or *gaur* introgression into the East Asian indicine genomes.

**[Comment of Reviewer #2]**

L292-294 What is the call rate for these 11 loci in the raw data? Can you rule out any effect
imputation may have had in this region? This is the most compelling evidence shown thus far.

I would recommend a deeper dive in this region to try to further strengthen your inference.

**Response:** *Thank you for this inquiry!*

*The call rate of 80 East Asian indicine cattle for these 11 loci in the raw data is in the table,*
*and the missing rate is 0.01 to 0.21.*

*Table 1 The call rate for 11 loci in 80 East Asian indicine cattle*

Chromosome	Position	Alleles	Number of missing alleles	Missing rate
25	216559	160	0	0
25	216571	160	0	0
25	216581	160	0	0
25	216850	160	16	0.1
25	219613	160	2	0.0125
25	219625	160	4	0.025
25	219635	160	10	0.0625
25	222783	160	0	0
25	222849	160	0	0

**[Comment of Reviewer #2]**

L318 Supp Fig 45 legend says 2 panels but there is only 1. Fig 46 shows N_e increasing in recent
793 times which is contrary to everything we know about cattle demography. Perhaps this relates to
794 my next point.

**Response:** *Thank you for this valuable observation! We have deleted the description of panel B*
*in Supplementary Fig. 45 (now Supplementary Fig. 33). For Supplementary Fig. 46 (now*
*Supplementary Fig. 36), demographic increases in the population size of indicine cattle can be*
*associated with postdomestication expansion, as observed in other domesticated species, and*
*with a more recent diffusion of *I1a* within East Asia.*

**[Comment of Reviewer #2]**

L325-344 This paragraph leads off with MSMC but it is unclear to me how you can use the Y
and MT in an MSMC analysis when the underlying model is based on recombination?

**Response:** *Thank you for kind inquiry!*

*The Bayesian skyline plots (BSPs) of mtDNA and Y chromosome data were generated using*
*BEAST v2.6.0, as reported in the Methods section. We have specified this at the beginning of*
*the paragraph to avoid misunderstanding: “Using an empirical Bayesian approach with*
*BEAST v2.6.0, we detected...”.*

**[Comment of Reviewer #2]**

L402-446 This is all speculation. Again, some is useful but this is 1.5 pages. I would recommend
that a deeper dive into any one or two of these would be more valuable to the reader than
speculating on all of them.

***Response:** Thank you for this valuable comment! We have revised this part and reduced
speculation. We agree that this text is too detailed for a discussion of our tentative results. It
has been condensed to 17 lines. We also made the preceding part of the Discussion more to the
point.*

**[Comment of Reviewer #2]**

L439-440 Proposes that EAI cattle may have introgression from a Bos-like ghost species as an
explanation for hemoglobin-related genes. Since hybridization and introgression from known
Bos species is difficult, and these large numbers of divergent sequences appear in only this
family of genes, is it possible that this 'ghost' group is a lost population of EAI-like Indicine
cattle, or that these mutations were specific to the extant EAI clade without any introgression?

***Response:** Thank you for this valuable comment! We believe that the phylogenetic position of
the 'ghost' species is not compatible with an EAI origin but instead indicates a sister species of
gaur. We also used an ancient kouprey genome (2× coverage) and gayal genomes to genotype
this region. The polygenetic tree showed that EAI was influenced not only by banteng and gaur
but by gayal and extinct kouprey species as well. Considering the low coverage of kouprey and
the hybrid ancestry of gayal, we did not analyze the gene flow between these two species and
EAI further.*

**[Comment of Reviewer #2]**

Methods

L491-493 were duplicate reads marked or removed? The 1000 Bulls spec is for them to be
marked. I just want to be sure that what is stated is what was actually done. Along those lines,
the spec has indel realignment and BQSR but that is not stated in the manuscript. Please
accurately specify what was actually done.

***Response:** Thank you for this valuable comment!*

*We apologize for the misleading description. We have rewritten the methods. We used BQSR to
map reads and we did not use indel realignment. We have revised the method in the manuscript.*

*We generated genotype data following the 1000 Bull Genomes Project Run 8 guideline
(<http://www.1000bullgenomes.com/>) (Supplementary Note 1). We removed low-quality bases
and artifact sequences using Trimmomatic v.0.39, and all clean reads were mapped to the cattle
reference assembly (ARS-UCD1.2) and Btau_5.0.1 Y BWA-MEM (v.0.7.13-r1126) with default
parameters¹. We then used SAMtools v.1.9 to sort bam files. For the mapped reads, potential
PCR duplicates were identified using 'MarkDuplicates' of Picard v.2.20.2*

(<http://broadinstitute.github.io/picard>). ‘BaseRecalibrator’ and ‘PrintReads’ of the Genome
Analysis Toolkit (GATK, v.3.8-1-0-gf15c1c3ef)² were used to perform base quality score
recalibration (BQSR) with the known variant file (ARSI.2PlusY_BQSR_v3.vcf.gz) provided by
the 1000 Bull Genomes Project.

For SNP calling, we created GVCF files using ‘HaplotypeCaller’ in GATK with the ‘-ERC
GVCF’ option. We called SNPs from combined GVCF files using ‘GenotypeGVCFs’ and
‘SelectVariants’, respectively. To avoid possible false-positive calls, we used VariantFiltration
as recommended: (1) SNP clusters with ‘-clusterSize 3’ and ‘-clusterWindowSize 10’ options;
(2) SNPs with mean depth (for all samples) $< 1/3\times$ and $> 3\times$ (\times , overall mean sequencing
depth across all samples); (3) quality by depth, $QD < 2$; (4) phred-scaled variant quality score,
$QUAL < 30$; (5) strand odds ratio, $SOR > 3$; (6) Fisher strand, $FS > 60$; (7) mapping quality,
$MQ < 40$; (8) mapping quality rank sum test, $MQRankSum < -12.5$; and (9) read position rank
sum test, $ReadPosRankSum < -8$ were filtered. We then filtered out nonbiallelic SNPs and SNPs
with missing genotype rates > 0.1 . The whole genome sequencing data from six other bovine
species were mapped in the same way. We used the 67,162,108 SNPs as a reference list to
genotype the combined set of 495 cattle samples and 22 samples of six other bovine species,
resulting in 67,145,163 SNP data with wild species data. The two final SNPs genotyping
datasets were imputed and phased using BEAGLE v.4.0³ with default parameters and filtered
by $DR2 < 0.9$ (Supplementary Table 3). The remaining SNPs were annotated according to their
positions using SnpEff v.4.3⁴. We also summarized samples and SNPs used for different
analyses in Supplementary Table 3.

**[Comment of Reviewer #2]**

L494 “...depth (for all individuals) $> 1/3\times$ and $< 3\times$...” I have no idea what this represents,
please clarify.

**Response:** Thank you for the inquiry! We have corrected this sentence as follows “SNPs with
mean sequencing depth (over all samples) $< 1/3\times$ and $> 3\times$ (\times , overall mean sequencing depth
across all samples) were removed”.

**[Comment of Reviewer #2]**

L500 Please specify BEAGLE parameters that were used. If defaults were used, then state that.
Additionally, you specify SNPs here, were only SNPs used or did you also use indels. Please
specify. Did you make any attempt to evaluate the accuracy of imputation? If so, you should
state that. If not, I would encourage you to evaluate this and include this information in any
filtering that you perform.

**Response:** Thank you for this valuable comment! We used only SNPs and default parameters

of BEAGLE v.4.0 for imputation. We also detected accuracy according to DR2. In this version,
we filtered sites with $DR2 < 0.9$, resulting in a total of 65,336,403 SNPs (Supplementary Table
3).

**[Comment of Reviewer #2]**

L502 This is based on an annotation version, in which case you should specify the exact file or
annotation version that was used to make it reproducible.

**Response:** Thank you for your advice. We specify that we used annotation version RefSeq
assembly GCF_002263795.1 for autosome annotation and RefSeq assembly
GCF_000003205.7 for Y chromosome annotation.

**[Comment of Reviewer #2]**

L547 I've already discussed the multiple testing issue and choice of $p < 0.005$.

**Response:** Thank you for the valuable comments.

*P* values were estimated based on Z-transformed values using the standard normal distribution
and were further corrected by multiple testing using the Benjamin-Hochberg false discovery
rate (FDR) method.

**[Comment of Reviewer #2]**

L557 Introgression analyses... How might a MAF threshold of 0.01 filtering affected these
analyses?

**Response:** Thank you for kind inquiry! We tried using the MAF threshold, but it reduced the
detection of rare alleles of wild bovine species.

**[Comment of Reviewer #2]**

L573 Same comment about the GTF version.

**Response:** Thank you for the valuable comments.

We specify that we used RefSeq assembly GCF_002263795.1 for autosome annotation and
RefSeq assembly GCF_000003205.7 for Y chromosome annotation.

**[Comment of Reviewer #2]**

L583 Already commented on the Y chromosome imputation.

**Response:** Thank you for this valuable comment! We have revised this part in the current
version of manuscript.

**[Comment of Reviewer #2]**

Figure 3 panels B-C and D-E appear to be switched relative to the legend. 10kb sliding window
appears to be different than what is described in the M&M.

**Response:** Thank you for this valuable observation! We have corrected the panels in Fig. 3.

REVIEWER COMMENTS

Reviewer #1 (Remarks to the Author):

General comments

The main message of this manuscript is that East Asian Indicine (EAI) cattle are descended from pure Indicine cattle originating in the South Asian core area (i.e. SAI, Indus Valley). Following the introduction of these pure Indicine animals via coastal routes, rapid and successful adaptation to hot and humid environments occurred through introgression of Banteng and/or Gaur genes. The age of introduction of the pure Indicine cattle to the new environment is estimated to be about 3000 years, i.e. 600 generations of cattle. That mean the first contact to and the introgression of Banteng and/or Gaur should be about 600 generations ago or younger. These results are supported by cumulative evidence from different analyses.

Adaptive introgression is a very important evolutionary mechanism, but currently it is becoming more popular and many signals are interpreted as introgression, often without an alternative hypothesis.

My main comments in the first report were about the quality of the reference groups, which are important for the above mentioned main message of this manuscript. I asked questions like:

- 1) Do the authors ensure that the SAIs are pure indicines without a genomic taurine segment?
- 2) Do the authors ensure that the Banteng samples are pure and do not contain a genomic segment from indicine or taurine cattle?
- 3) Do the authors ensure that the Gaur samples are pure and do not contain a genomic segment from indicine or taurine cattle?

In his rebuttal letter, the authors state that this is ensured, but there are some inconsistencies that worry me, and these inconsistencies are the main problem with this response.

As already mentioned by Reviewer #2 (first comment on the first submission), in a project of this magnitude it is inevitable that the material will be analysed by different groups of authors and that the results will also be described and interpreted by different groups of

authors. Therefore, Reviewer #2 asks the corresponding authors to go through the entire manuscript in order to make it more consistent.

In my opinion, this has not yet been done. I am even more disturbed by the fact that not possibly all analyses were carried out with the same material. Just a few examples:

- 1) Supplementary Table 1 lists 514 samples, but at the beginning of the Results section, lines 139-145, the authors mentioned 517 samples ($297+198+22=517$).
- 2) Eight Banteng are mentioned in the main text, but only five are listed in Table S1.
- 3) Table S11 lists eight banteng, but only B1, B2 and B3 are identical to Table S1. Banteng like banteng08, banteng09, banteng_ypt2224, banteng_ypt2225, banteng_ypt2230 are not described in the main material (Table S1).
- 4) Heading of the important table S11 is wrong (source group twice).
- 5) Cross-reference to Table S11 is wrong (see Supplementary Note 4 line 365).
- 6) Supplementary Note 4 mentions gaur04, but there is no gaur04 in Table S1.
- 7) Supplementary Note 4 mentions 40 core and pure Indicine cattle, but in Table S1 (last column) we find only 20 Indicine cattle. Table S11 mentions 17 pure SAI animals. Most probably we should add Sha3b, Thar1 and Har03 to this list and thus get 20 again and not 40. The Response to my first comment also mentions 40 Core and pure Indicine cattle.
- 8) Table S1 and Figure S7 do not use the same sorting. Therefore, the reader cannot assign the results of the Admixture analysis to a particular Biosample ID.

These are just inconsistencies I spotted because I was particularly interested in the use of pure reference panels in the introgression analyses. I am not in a position, and it is not my job, to check all the cross-references and all the figures, but from the above I get the impression that the authors do not consider Reviewer #2's first comment. The authors do not address it either.

With respect to ADMIXTURE, TreeMix, D- and f3-statistics.

All these analyses are performed at the genome or even population level, which does not necessarily indicate the purity of the animal at the locus level. The analyses that check for possible admixture at the locus/segment level are more appropriate. This is particularly important for the Banteng and Gaur reference population.

As mentioned in this paper (<https://www.nature.com/articles/s41467-018-05257-7>),

factors like large difference in the number of individuals sampled per breed, genetic drift in the population can affect its ancestry composition. Similarly, post-admixture drift can also distort the signals of f_3 -statistics; this is also mentioned by the authors themselves in the legend of Table S11. Further, in the description of f_3 -statistics (supplementary note 4), the authors incorrectly mentioned the threshold of Z-score as 3; for the test to be significant the Z-score of f_3 -tests should be less than -3. Next, f_3 -test is better suited to group of animals (increases the power of test) rather than performing on each individual separately. I would also recommend the authors to perform f_3 -tests on each SAI cattle breed as target with Banteng as P1 (one of the source population) and Zebu as P2 (another source population) to check the sensitivity of this test.

Methods used to identify signature of selection such as composite likelihood ratio (CLR) and iHS has increased power if the information about derived alleles is used. Since the authors have already aligned Gaur, Banteng, Bison, Wisent, Yak and Buffalo to the cattle assembly, they could have easily ascertain the ancestral alleles. Therefore, I am curious as to why the authors decided not to use ancestral allele information for this analysis. Further, the authors have used the results of Tajima's D to support their signature of selection analysis, however, in some cases (for instance, Fig 2 and 3), it is hardly negative. Usually, the significance of Tajima's D, whether or not the obtained value significantly deviates from 0, is obtained using beta distribution (table 2, of original Tajima's D paper). Therefore, I suggest that the authors either remove the results of Tajima's D or justify it.

Further, parameters used in D-statistics are not described in sufficient details. Please include it in Table S3 too.

There are 84 introgressed and then positively selected genes in 42 candidate regions (line 274, U50). These are validated by phylogenetic analysis. Supplementary Figures 20-26 show the phylogenetic analysis of 27 of these regions. All these phylogenetic trees indicate complete fixation of the introgressed segment. The authors show that EAI cattle are descended from pure SAI cattle, but in the 27 regions presented here, not a single EAI haplotype clustered with its native group of origin. Such 100% turnover could be expected for one or two strongly positively selected segments, but I would not expect this for all 27 segments shown in Figures S20-S26.

In what demographic scenario would such a 100% turnover be expected for 27 segments

presented?

How strong must positive selection be to observe the phylogenetic trees shown in Figures S20-S26?

In all 27 trees, EUT and SAI haplotypes form monophyletic groups, and these are paraphyletic with respect to EAI, and there is not a single EAI haplotype that cluster with its origin group SAI. Could the Funder effect cause the observed patterns?

In the main text (lines 279-288), the positively selected segment in the proximal region of BTA25 is presented and discussed (fig. S26 and fig. 4). According to this, EAI animals in most of the EAI subpopulations studied carry the native SAI haplotype with a frequency of >0.5. The haplotypes of presumed Banteng and Gaur origin show only an increased frequency. In my opinion, this strongly contradicts the phylogeny shown in Supplementary Figure 26, where no EAI haplotype was clustered with SAI.

In addition to Banteng and Gaur, the authors implicate Kouprey, Gayal (data origin not given) and some phantom species as explanations for the observed diversity in the proximal segment of BTA25. The authors should at least try to discuss an explanation other than the implausible and complicated multi-species introgression and subsequent positive selection? If I remember correctly, the globin gene family, and in particular the α -globin chain, is known for a complicated evolution of paralogous genes, including recurrent mutations, gene conversion, co-evolution of different family members, and so on. Authors should check if complicated evolution of paralogues gene families could affect their conclusions.

Figure S24 shows three chromosomal segments on BTA18. All three are located in the most complex region of the *Bos taurus* and *Bos indicus* genomes (see <https://doi.org/10.3168/jemandes2021-21625>). The segmental duplication behaves similarly to the gene families mentioned above, and the region on BTA18 is very difficult to reconstruct correctly. The authors should check whether segmental duplication (in general) could influence their conclusions.

Other minor comments

Line 76-77, the authors write ...“was a direct consequence of a small number of genomic regions.” While it is conceivable that introgressed segments in EAI cattle did provide adaptive advantage, this statement is too strong especially considering that authors themselves have identified genomic segments under selection that were not introgressed

from its wild relatives. Selection based on standing genetic variation should at least be discussed as a possibility. Further, authors should consider removing the phrase “direct consequence” as the authors did not provide any evidences showing direct and strong causality between introgressed segments and adaptation in EAI cattle.

Line 113, “..where have large local population are found” This sentence appears to be grammatically incorrect, please rephrase it.

Line 205, section, “Ancestral environmental adaptation of...” ◇ The authors should at least provide a sentence or two to give readers some information about the methods applied to identify signature of selection in this section. I am emphasizing this because when I started reading this section I was under the impression that the same set of methods were applied here as applied in the previous section, i.e. Fst, theta and XP-EHH, but it was not until the end of this section that it was mention that a different set of methods were applied, i.e. CLR and iHS. Moreover, the naming of the sections are also confusing, “the ancestral adaptation of indicine cattle”, “Ancestral environmental adaptation of south Asian indicine cattle”, “the indicine adaptation to the tropical, humid environment”. Ideally, signature of selection analysis should be performed without a priori expectation because it is very difficult to conclusively link the identified genes under selection with the breed characteristics; usually, it is done by functional genomics and by studying the literature extensively. Therefore, such naming of the sections is suitable to discussion but not to result section.

Reviewer #2 (Remarks to the Author):

Overall, the manuscript has improved significantly. I appreciate the author’s willingness to make changes and incorporate suggestions. There are three documents as I see it, the main manuscript, the supplement and the response to previous review, all of which are lengthy. I have comments for each, which I’ll address individually, but the most pressing issue relates to the ROH analysis under Supplementary Note 2: Genetic diversity. I’ll start there because I think this needs addressed.

Response L358-400 and supplement L273-280. I still do not believe your ROH analysis is accurate. Even with your new analysis, you show a large number of individuals with a total genome length contained within ROH to be >1 Gb, or over a third of the genome. That is

simply not realistic. There has been a large amount of prior literature on the subject, largely from SNP-chip data, and all of them show the “extreme” samples having under 1 Gb in ROH. I believe this [1] is an informative publication.

[1] <https://bmcmgenomdata.biomedcentral.com/articles/10.1186/1471-2156-13-70>

I believe Figure 5 from [1] is particularly useful in that it shows the same trend you are seeing in your data but the scale is quite different. This is particularly important because if the current manuscript is published it will present conflicting values because I believe you are including far too much of the genome in ROH. It is generally accepted that these types of ROH analysis are measuring recent inbreeding, which you acknowledge on L160 of the manuscript, where the size of ROH regions is proportional to the inbreeding coefficient. As the size of a ROH decreases you are estimating IBD further back in time. This may be appropriate in some cases, even perhaps the current manuscript, but your ability to differentiate old IBD regions from random chance or other population forces requires increased sample sizes, which you do not have. As I said, on its face, I do not believe your ROH results. The question is, what is causing them to be elevated? In order to try to figure this out, I extracted 187 samples from your study that are also present within my UMAG1 call set, which is based on ~5500 genomes and VQSR. For these 187 samples, I extracted two sets of variants, A) all bi-allelic SNPs that passed VQSR and B) the positions contained on the Illumina Bovine HD and GGPF250 genotyping assays in order to compare with the results from [1].

A) “bcftools view --threads \$CPU -f PASS -m2 -M2 -v snps -S \$InputSamples --force-samples -Ob -o \$c.\$Prefix.bcf \$InputBcf”

B) “bcftools view --threads \$CPU -R \$c.variants2get.list --trim-alt-alleles -Ob -o \$c.\$Prefix.bcf \$InputBcf”

Dataset A produced 125M variants for these 187 samples.

bcftools +counts ChenAll.bcf

Number of samples: 187

Number of SNPs: 125813861

Number of INDELS: 0

Number of MNPs: 0

Number of others: 0

Number of sites: 125813861

Dataset B produced ~850K variants.

```
bcftools view -S chen_sample.list --force-samples ChenChip.bcf -Ou | bcftools +counts
```

Number of samples: 187

Number of SNPs: 850772

Number of INDELS: 64337

Number of MNPs: 0

Number of others: 0

Number of sites: 897036

The bcf files were then converted to plink bed/bim/bam:

```
plink --bcf ChenAll.bcf --const-fid 0 --chr-set 29 --allow-extra-chr --out ChenAll
```

```
plink --bcf ChenChip.bcf --const-fid 0 --chr-set 29 --allow-extra-chr --out ChenChip
```

Finally, the ROH analysis was performed using various parameter values.

The first run used the values specified on L279-280 of main paper and L389-396 of Response. I will refer to this as “ChenParameters”.

```
plink --bfile ChenAll --chr-set 29 --chr 1-29 --homozyg --homozyg-gap 1000 --homozyg-kb 100 --homozyg-snp 50 --homozyg-window-het 3 --homozyg-window-snp 50 --homozyg-window-threshold 0.05 --out ChenAll
```

The second run was the same as above but changed --homozyg-window-het 3 to 1. I refer to this as Chen1het.

The third run was the same as above but changed --homozyg-window-het 3  1, --homozyg-snp 50  200 --homozyg-window-snp 50  100 --homozyg-snp 50  200. I refer to this as ChenStrict.

The loci from the SNP-chips were run the same as the second analysis above which used the author’s parameters but changed --homozyg-window-het 3  1. I refer to this as Chip1het. All of the results from the *.hom.indiv plink files were copied into Excel and visualized. See

file ChenROHsubmitted.xlsx which should be available with this review. It is clear from Figure 1 that I was able to recreate the trend that is shown in Supp. Figures 2 & 3 with my “ChenParameters” in blue. As I contend, these parameters still include too many ROH and encompass too much of the genome per individual. The “Chen1het” in orange only change --homozyg-window-het from 3 to 1 which produces values that more closely resemble prior literature and what we know about these populations. Therefore, I believe the root problem with the analyses, as presented, is this parameter. In hindsight, this makes sense as this parameter is meant to allow for genotyping error. By setting this parameter to 3 and requiring only 100 loci within a window, I believe you are effectively saying that you have little confidence in your genotype calls because you are allowing 3 errors (3% error rate) to still call a region as homozygous. If you are confident in your genotype calls, and new mutations are negligible (which I think is true here) then a value of 1 is more appropriate and in fact yields results closer to expectations. In order to more fully explore this, I also created a more strict dataset (ChenStrict) which required more loci while still maintaining 1 het position (grey). In order to provide a direct comparison to snp-chip data I also present the same as “Chip1het” in red. As you can see, the “Chip1het” is comparable to prior literature from using these assays. My final interpretation of these data is that the parameters used in the revision are still not appropriate, the main parameter to change is --homozyg-window-het and the correct values for the other parameters are somewhere between what was used and the “strict” values that I used. This is a fairly trivial analysis that the authors should perform and update the manuscript.

As before, I will provide comments on the Supplement by referring to SL as supplement line number.

SL261-264: “The whole genome sequencing data from six other bovine species were mapped in the same way. We used the 67,162,108 SNPs as a reference list to genotype the combined set of 495 cattle samples and 22 samples of six other bovine species, resulting in 67,145,163 SNP data with wild species data.”

It is unclear to me what you mean by “...as a referencelist...”? My initial interpretation of this is that you added additional samples, called genotypes on those additional samples, and then just extracted the 67M positions from the prior variant call set. Is this correct? The proper way to perform this analysis would be to recall genotypes from **all** of your samples

starting with the combined GVCFs from SL253. This confusion is exacerbated by the third description in Supp. Table 3 which says “A total of 64,475,272 SNPs called from all 495 cattle and 22 genomes from other six bovine species.” Please clarify exactly what you did here.

SL266 Supplementary Table 3 This table is very informative and helpful to the reader and this reviewer. However, now that I have this table, it presents new problems. For the second line of this table where it says “Imputed data of...” I assume this represents the phasing and imputation that you did with Beagle, correct? If so, please add the word “phased” to this description. This is a very important point later.

None of these numbers seem to add up, going back to my comment directly above. It seems to me that the proper way to analyze all of these data is to do joint genotyping on all samples simultaneously, which would be your 514 samples presented in Supp. Table 1. (Note that the second line of this table, and in the manuscript and supplement you refer to 495+22 which is 517 but you only have 514 in Table1.) This will result in a total number of variants N. All analyses performed after this will involve some level of filtering or selection of loci but they all will go back to this original N variants. As it stands, this table helped clear up some issues but it presented more. For example the 6th description in the number of samples column, “All 495 cattle and one yak”. Table 1 lists 3 yak while SL237 says two yak. Table 1 has 5 banteng while SL237 says eight. Figure 4 (F) of the main manuscript mentions Kouprey but I do not see any in table 1 as Kouprey or Bos sauveli. Where did this sample come from? Again, many of these numbers are not consistent.

SL285 and 300 The PCA and admixture analysis used -indep-pairwise 50 10 0.1 while the NJ and ML phylogeny used --indep-pairwise 50 5 0.1. Why would you use different parameters for two different analyses to achieve the same objective of minimizing LD between the variants you use? By using different parameter sets you effectively make a direct comparison of these two analyses impossible because you changed the loci that you are using.

SL312 “...XP-EHH score using selscan v.1.1 with default settings.” Selscan software did not have the ability to use unphased data until v2.0. Therefore, I assume that you are relying on the Beagle phased and imputed data for these analyses, is that correct? If so, this is why it is

important to add the word phased to Supp. Table 1 because without doing the phasing in Beagle you could not have used selscan v.1.1.1.

SL329 "... using SweepFinder2."

From the manual, it appears that this requires a recombination rate file. What did you use as the recombination rate between loci? My assumption is that you assumed a constant recombination rate of 1 cM/Mb since you do not actually have a recombination map. If that is the case, please state that in the manuscript. This software also requires a "B-value" for each variant. Where did you get these values from?

SL354: Why were EAI grouped into a single population without looking into recent specific breed introgression as the potential source of admixed genotypes? Often times the direction of admixture events is difficult to estimate. Is the signal of introgression the result of the modern banteng/gaur samples having some amount of domestic cattle introgression in their genomes?

SL360-S365: The f3 data does not appear to be presented in Supplementary table 10, There appears to be a numbering issue of tables in the supplementary data document, with a shift of +1 after supplementary table 7.

Main manuscript referring to ML as manuscript line number:

ML 76 "...was a direct consequence of..." That's a fairly strong claim. I think you evidence consistent with a hypothesis but I question whether it represents a direct consequence.

ML113 remove the word have.

ML261 Supp. Figure 16 has multiple pages of figures with the same panel labels but only 1 legend. Please have a look at this as it appears it may have been an error when preparing the files for submission.

ML264-266 "To identify regions in the EAI genomes that were likely under selection, we used a statistic to detect positively selected and introgressed genes (PSIGs)" Are these actually detecting genes or regions? You use both terms in the same sentence.

ML351 Supp. Fig. 38 This is a much better figure than the previous version!

ML512 "...gene transfer format (GTF) (GCF_002263795.1) file." This is an accession for the genome assembly, it is NOT an accession for a gene set. I assume that you either used annotations from NCBI or Ensembl, both of which have many different versions which are regularly updated. You need to specify which annotations you used (NCBI or Ensembl) and exactly which version of the annotation.

**Responses to Reviewers' comments**

**[Comment of Reviewer #1]**

General comments

The main message of this manuscript is that East Asian Indicine (EAI) cattle are descended
from pure Indicine cattle originating in the South Asian core area (i.e. SAI, Indus Valley).
Following the introduction of these pure Indicine animals via coastal routes, rapid and
successful adaptation to hot and humid environments occurred through introgression of
Banteng and/or Gaur genes. The age of introduction of the pure Indicine cattle to the new
environment is estimated to be about 3000 years, i.e. 600 generations of cattle. That mean the
first contact to and the introgression of Banteng and/or Gaur should be about 600 generations
ago or younger. These results are supported by cumulative evidence from different analyses.
Adaptive introgression is a very important evolutionary mechanism, but currently it is
becoming more popular and many signals are interpreted as introgression, often without an
alternative hypothesis.

*Response: Yes, we automatically assume adaptive introgression. This is at least partially*
*supported by the GO and KEGG enrichment results but would of course need additional*
*evidence from functional studies.*

**[Comment of Reviewer #1]**

My main comments in the first report were about the quality of the reference groups, which are
important for the above mentioned main message of this manuscript. I asked questions like:

- 1) Do the authors ensure that the SAIs are pure indicines without a genomic taurine segment?
2) Do the authors ensure that the Banteng samples are pure and do not contain a genomic
segment from indicine or taurine cattle?
3) Do the authors ensure that the Gaur samples are pure and do not contain a genomic segment
from indicine or taurine cattle?

In his rebuttal letter, the authors state that this is ensured, but there are some inconsistencies
that worry me, and these inconsistencies are the main problem with this response.

*Response: Thank you for this valuable comment. Indeed, pure ancestry is never guaranteed for*
*a group of proximate cross-fertile species. However, according to your suggestions, we have*
*checked the D statistics among indicine cattle, taurine cattle, banteng and gaur samples to*
*select pure SAI samples without taurine, banteng or gaur introgression, select pure taurine*
*cattle without SAI cattle, banteng or gaur introgression, and select banteng or gaur without SAI*
*and taurine introgression. Finally, we selected 15 pure SAI cattle, 15 taurine cattle, 4 banteng,*
*and 2 gaur and used them for introgression analysis (Supplementary Table 10 and*
*Supplementary Dataset 1).*

*The D statistic was used to select pure SAI cattle, taurine cattle, banteng, and gaur for*
*introgression analysis. For SAI cattle, we used the three tree topologies of D (SAI individual,*
*SAI individual; taurine cattle, buffalo), D (SAI individual, SAI individual; banteng individual,*
*buffalo), and D (SAI individual, SAI individual, gaur, buffalo) to select the SAI samples without*

any gene flow from taurine cattle, banteng or gaur. For taurine cattle, we used three tree
topologies of *D* (taurine individual, taurine individual; SAI, buffalo), *D* (taurine individual,
taurine individual; banteng individual, buffalo), and *D* (taurine individual, taurine individual;
gaur individual, buffalo) to select taurine samples without any gene flow from SAI cattle,
banteng or gaur. For banteng, we used two tree topologies of *D* (banteng individual, banteng
individual; SAI individual, buffalo) and *D* (banteng individual, banteng individual; taurine
individual, buffalo) to select banteng samples without any gene flow from taurine or SAI cattle.
For gaur, we used two tree topologies of *D* (gaur individual, gaur individual; SAI individual,
buffalo) and *D* (gaur individual, gaur individual; taurine individual, buffalo) to select gaur
samples without any gene flow from taurine or SAI cattle. We used all combinations of
individuals to calculate the *D* statistic. We finally selected a panel of 15 pure SAI cattle, 15
taurine cattle, 4 banteng, and 2 gaur samples with a $|Z \text{ score}| < 3$ for RFmix analysis, *D* statistic,
*U*₂₀, and *U*₅₀ statistical calculation (Supplementary Data 1).

We repeated the analysis of introgression using these samples.

**[Comment of Reviewer #1]**

As already mentioned by Reviewer #2 (first comment on the first submission), in a project of
this magnitude it is inevitable that the material will be analysed by different groups of authors
and that the results will also be described and interpreted by different groups of authors.
Therefore, Reviewer #2 asks the corresponding authors to go through the entire manuscript in
order to make it more consistent.

In my opinion, this has not yet been done. I am even more disturbed by the fact that not possibly
all analyses were carried out with the same material. Just a few examples:

1) Supplementary Table 1 lists 514 samples, but at the beginning of the Results section, lines
139-145, the authors mentioned 517 samples (297+198+22=517).

**Response:** We apologize for our unclear description. We finally used 517 samples, including
495 cattle genomes and 22 wild bovine species genomes. We also used one gayal and one
ancient kouprey sample for the introgression region analysis of *BTA25*. We added this
information in the Supplementary Note 4.

“For the analysis of the introgressed region of *BTA25* (0.21-0.26 Mb), we also used a gayal
sample and an ancient kouprey sample to detect its origin. The coverage of gayal and kouprey
was 17.32× and 1.4×, respectively. Due to the hybrid origin of gayal and low coverage of the
kouprey genome, we did not examine possible introgression of gayal and kouprey. The publicly
available sequences were downloaded from China National GeneBank (CNGB) and the SRA
with the following project accession numbers: CRX165997 (gayal, YD4) and PRJNA764746
(kouprey).” Considering the low coverage of the kouprey genome and the hybrid ancestry of
gayal, we did not further analyze the gene flow between these two species and EAI cattle.

**[Comment of Reviewer #1]**

2) Eight Banteng are mentioned in the main text, but only five are listed in Table S1.

**Response:** We apologize for our unclear description. We included 8 banteng and 5 gaur in our

*dataset. According to your comment, we updated Supplementary Table S1 and added all*
*information on the 8 banteng, and we only used 4 banteng and gaur samples in the introgression*
*analysis.*

**[Comment of Reviewer #1]**

3) Table S11 lists eight banteng, but only B1, B2 and B3 are identical to Table S1. Banteng like
banteng08, banteng09, banteng_ypt2224, banteng_ypt2225, banteng_ypt2230 are not
described in the main material (Table S1).

*Response: We apologize for our mistakes. We updated our sample information, and we renamed*
*our samples to ensure that the names are consistent in all materials.*

**[Comment of Reviewer #1]**

4) Heading of the important table S11 is wrong (source group twice).

*Response: Thank you for your valuable comment. We have revised Supplementary Table 11,*
*and we selected pure SAI cattle, taurine cattle, banteng, and gaur for f_3 analysis.*

**[Comment of Reviewer #1]**

5) Cross-reference to Table S11 is wrong (see Supplementary Note 4 line 365).

*Response: We have double-checked the references to the Supplementary Tables 10 and 11.*

**[Comment of Reviewer #1]**

6) Supplementary Note 4 mentions gaur04, but there is no gaur04 in Table S1.

*Response: We have renamed the sample and ensured consistency in all materials.*

**[Comment of Reviewer #1]**

7) Supplementary Note 4 mentions 40 core and pure Indicine cattle, but in Table S1 (last column)
we find only 20 Indicine cattle. Table S11 mentions 17 pure SAI animals. Most probably we
should add Sha3b, Thar1 and Har03 to this list and thus get 20 again and not 40. The Response
to my first comment also mentions 40 Core and pure Indicine cattle.

*Response: We apologize for our mistakes. We finally selected 15 pure indicine cattle, 15 taurine*
*cattle, 4 banteng, and 2 gaur for introgression analysis. According to your suggestions, we have*
*checked the D statistic among indicine, taurine, banteng and gaur samples. We have corrected*
*our mistakes in Supplementary Note 4 and Supplementary Tables 10 and 11.*

**[Comment of Reviewer #1]**

8) Table S1 and Figure S7 do not use the same sorting. Therefore, the reader cannot assign the
results of the Admixture analysis to a particular Biosample ID.

*Response: Thank for your suggestion. We have sorted the samples in Supplementary Table 1*
*and Supplementary Figure 7 to ensure consistency.*

These are just inconsistencies I spotted because I was particularly interested in the use of pure
reference panels in the introgression analyses. I am not in a position, and it is not my job, to
check all the cross-references and all the figures, but from the above I get the impression that
the authors do not consider Reviewer #2's first comment. The authors do not address it either.

*Response: Thank you for your careful review of the consistency of Supplementary Figures and*
*Tables. We have double checked all figures and tables.*

**[Comment of Reviewer #1]**

With respect to ADMIXTURE, TreeMix, D- and f₃-statistics. All these analyses are performed
at the genome or even population level, which does not necessarily indicate the purity of the
animal at the locus level. The analyses that check for possible admixture at the locus/segment
level are more appropriate. This is particularly important for the Banteng and Gaur reference
population.

*Response: Thank for your suggestions. Accordingly, and to ensure the quality of the reference*
*groups, we calculated the D statistic and selected 15 pure South Asian indicine (SAI) cattle, 15*
*taurine cattle, 4 banteng, and 2 gaur for introgression analysis. We agree with your point, and*
*our results in Supplementary Figs. 20-25 show locus-specific trees, which show if a fragment*
*introgressed from, for example, banteng has a non-banteng origin.*

**[Comment of Reviewer #1]**

As mentioned in this paper (<https://www.nature.com/articles/s41467-018-05257-7>), factors like
large difference in the number of individuals sampled per breed, genetic drift in the population
can affect its ancestry composition. Similarly, post-admixture drift can also distort the signals
of f₃-statistics; this is also mentioned by the authors themselves in the legend of Table S11.
Further, in the description of f₃-statistics (supplementary note 4), the authors incorrectly
mentioned the threshold of Z-score as 3; for the test to be significant the Z-score of f₃-tests
should be less than -3. Next, f₃-test is better suited to group of animals (increases the power of
test) rather than performing on each individual separately. I would also recommend the authors
to perform f₃-tests on each SAI cattle breed as target with Banteng as P1 (one of the source
population) and Zebu as P2 (another source population) to check the sensitivity of this test.

*Response: Thank you for your suggestions. We agree that Admixture patterns should be*
*interpreted with caution. However, our Admixture pattern reproduce the divergence of*
*geographically separated clusters: European taurine, African taurine, African indicine, South*
*Asian indicine and East Asian indicine cattle. In addition, to ensure the quality of the reference*
*groups, we calculated the D statistic and selected 15 pure South Asian indicine (SAI) cattle, 15*
*taurine cattle, 4 bangteng, and 2 gaur samples. We also calculated f₃ statistics for each SAI*
*breed as targets with banteng, gaur, taurine, and other SAI breeds. For the f₃ statistic, if the Z*
*score ($Z \leq -3.0$) is significantly negative, test population C has admixture from both reference*
*populations A and B. All f₃ statistics were positive, indicating that there was no evidence of*
*admixture (Supplementary Table 11).*

**[Comment of Reviewer #1]**

Methods used to identify signature of selection such as composite likelihood ratio (CLR) and
iHS has increased power if the information about derived alleles is used. Since the authors have
already aligned Gaur, Banteng, Bison, Wisent, Yak and Buffalo to the cattle assembly, they
could have easily ascertain the ancestral alleles. Therefore, I am curious as to why the authors
decided not to use ancestral allele information for this analysis.

*Response:*

*Thank you for your suggestions. In iHS and CLR computation, information on the ancestral*
*and derived allele state is needed for each SNP. In our analysis, the ancestral allele was defined*
*as the allele fixed in the swamp buffalo that was included in the genotype call set, and the*
*ambiguous SNP was discarded. We also updated the iHS and CLR results in our manuscript.*
*We updated our results and added details in the Methods.*

**[Comment of Reviewer #1]**

Further, the authors have used the results of Tajima's D to support their signature of selection
analysis, however, in some cases (for instance, Fig 2 and 3), it is hardly negative. Usually, the
significance of Tajima's D, whether or not the obtained value significantly deviates from 0, is
obtained using beta distribution (table 2, of original Tajima's D paper). Therefore, I suggest that
the authors either remove the results of Tajima's D or justify it.

Further, parameters used in D-statistics are not described in sufficient details. Please include it
in Table S3 too.

***Response:** Thank you for your comment. We have removed Tajima's D. We also added the*
*parameters used in calculating the D statistic in Supplementary Table 3.*

**[Comment of Reviewer #1]**

There are 84 introgressed and then positively selected genes in 42 candidate regions (line 274,
U50). These are validated by phylogenetic analysis. Supplementary Figures 20-26 show the
phylogenetic analysis of 27 of these regions. All these phylogenetic trees indicate complete
fixation of the introgressed segment. The authors show that EAI cattle are descended from pure
SAI cattle, but in the 27 regions presented here, not a single EAI haplotype clustered with its
native group of origin. Such 100% turnover could be expected for one or two strongly positively
selected segments, but I would not expect this for all 27 segments shown in Figures S20-S26.
In what demographic scenario would such a 100% turnover be expected for 27 segments
presented? How strong must positive selection be to observe the phylogenetic trees shown in
Figures S20-S26? In all 27 trees, EUT and SAI haplotypes form monophyletic groups, and these
are paraphyletic with respect to EAI, and there is not a single EAI haplotype that cluster with
its origin group SAI. Could the Funder effect cause the observed patterns?

***Response:** Thank you for sharing this concern. We apologize for our unclear description. In*
*our previous revision, we used the same color for indicine or taurine haplotypes. In the indicine*
*group, we did find EAI haplotypes clustered with SAI, but we did not color them.*

*In the revised manuscript, we first repeated the U50 analysis, specified the length of the*
*introgressed regions, and used 5 SAI cattle, 5 taurine cattle, 4 banteng, and 2 gaur samples and*
*other wild species samples to construct phylogenetic trees and validate the introgression. We*
*excluded four complex regions, and finally obtained 23 regions. Our results showed that*
*introgressed haplotypes of EAI cattle were clustered with banteng or gaur, while no introgressed*
*haplotypes were clustered with South Asian indicine cattle (Supplementary Figs. 20-25).*

**[Comment of Reviewer #1]**

In the main text (lines 279-288), the positively selected segment in the proximal region of

BTA25 is presented and discussed (fig. S26 and fig. 4). According to this, EAI animals in most
of the EAI subpopulations studied carry the native SAI haplotype with a frequency of >0.5.
The haplotypes of presumed Banteng and Gaur origin show only an increased frequency. In my
opinion, this strongly contradicts the phylogeny shown in Supplementary Figure 26, where no
EAI haplotype was clustered with SAI.

***Response:** Thank you for this concern. We are very sorry that we did not explain this clearly.
We have revised Supplementary Figs.20-26 (new version Figs.20-25), we colored all EAI
haplotypes, and new figures showed that non-introgressed haplotypes of EAI cattle were
clustered with SAI haplotypes. In Supplementary Figs. 20-25, we first specified the length of
the introgressed regions and used 5 pure SAI cattle, 5 taurine cattle, 4 banteng, 2 gaur, 3 yak,
2 bison, 2 wisent, 2 buffalo, and 80 EAI samples to construct phylogenetic trees. Our results
showed that introgressed haplotypes of EAI were clustered with banteng or gaur, while no
introgressed haplotypes were clustered with SAI haplotypes. We also found that some EAI
haplotypes also clustered with taurine cattle, which showed that some EAI also have taurine
ancestry.*

*In some regions, a small number of markers may also explain why the position of the bison-
wisent-yak outgroup or gaur-banteng is variable. In some trees, banteng and gaur are
separated. We also checked that the genomic segments were demarcated accurately and
contained only the introgressed fragments. Please see Supplementary Figs. 20-25.*

**[Comment of Reviewer #1]**

In addition to Banteng and Gaur, the authors implicate Kouprey, Gayal (data origin not given)
and some phantom species as explanations for the observed diversity in the proximal segment
of BTA25. The authors should at least try to discuss an explanation other than the implausible
and complicated multi-species introgression and subsequent positive selection?

***Response:** Thank you for this specific comment. Introgression of different bovine species occurs
everywhere the species share territory. The cluster patterns indeed suggest sequences not
originating from zebu, banteng, gaur or kouprey, but this may reflect, especially for the gaur,
their limited sampling. For the selected genes, we have revised our manuscript and now refer
to the selected genes as frequently introgressed genes.*

**[Comment of Reviewer #1]**

If I remember correctly, the globin gene family, and in particular the α -globin chain, is known
for a complicated evolution of paralogous genes, including recurrent mutations, gene
conversion, co-evolution of different family members, and so on. Authors should check if
complicated evolution of paralogous gene families could affect their conclusions.

***Response:** Thank you for this specific comment. We checked that the bovine genome assembly
shows two HBA genes that encode identical HbA subunits. We did not find reports on HbA
rearrangements or paralogs that would invalidate our results.*

**[Comment of Reviewer #1]**

Figure S24 shows three chromosomal segments on BTA18. All three are located in the most

complex region of the *Bos taurus* and *Bos indicus* genomes (see
<https://doi.org/10.3168/jemandes2021-21625>). The segmental duplication behaves similarly to
the gene families mentioned above, and the region on BTA18 is very difficult to reconstruct
correctly. The authors should check whether segmental duplication (in general) could influence
their conclusions.

**Response:** *Thank you for this specific comment. The correct link to Dachs et al. (2021) is*
*<https://doi.org/10.3168/jds.2021-21625>. This paper indeed mentions SVs near a QTL region on*
*(BTA18, 57,816,000-59,430,000), close to the regions shown in Supplementary Fig. 24 (now in*
*Supplementary Fig. 23, 60240001-60740000). Thirty of the 31 SVs are upstream of the regions*
*we are studying, but one of their samples has a deletion of 59123315 to 61313922. This would*
*delete all segments on which the three BTA18 trees in Supplementary Fig. 23 are based, but*
*does not seem to influence the phylogenetic identification of introgression in other samples. In*
*the legend of Fig. S23 we have mentioned the occurrence of this SV with reference to Dachs et*
*al (2023).*

**[Comment of Reviewer #1]**

Other minor comments

Line 76-77, the authors write ...“was a direct consequence of a small number of genomic
regions.” While it is conceivable that introgressed segments in EAI cattle did provide adaptive
advantage, this statement is too strong especially considering that authors themselves have
identified genomic segments under selection that were not introgressed from its wild relatives.
Selection based on standing genetic variation should at least be discussed as a possibility.
Further, authors should consider removing the phrase “direct consequence” as the authors did
not provide any evidences showing direct and strong causality between introgressed segments
and adaptation in EAI cattle.

**Response:** *Thank you for this suggestion. We have corrected the Abstract as suggested.*

**[Comment of Reviewer #1]**

Line 113,“..where have large local population are found” This sentence appears to be
grammatically incorrect, please rephrase it.

**Response:** *Thank you for this valuable comment. We deleted “have”.*

**[Comment of Reviewer #1]**

Line 205, section, “Ancestral environmental adaptation of...” □ The authors should at least
provide a sentence or two to give readers some information about the methods applied to
identify signature of selection in this section. I am emphasizing this because when I started
reading this section I was under the impression that the same set of methods were applied here
as applied in the previous section, i.e. Fst, theta and XP-EHH, but it was not until the end of
this section that it was mention that a different set of methods were applied, i.e. CLR and iHS.

**Response:** *Thank you for this suggestion. We have added the description in the first and last*
*paragraphs.*

*First paragraph of the section:*

“Throughout the history of migration and admixture of indicine cattle, genomic regions under
selection might have been lost in specific indicine groups. We therefore also performed a test
for positive selection signatures in SAI cattle using $\theta\pi$, iHS , CLR , and F_{ST} estimates based on
the comparison between SAI and non-SAI groups.”

*Last paragraph:*

“Additionally, we compared the light- and dark-coated SAI breeds, such as the white-coated
Bhagnari and Dajal cattle from Pakistan, by using F_{ST} and $\theta\pi$ ratio estimates. We identified
shared selective sweeps around pigmentation loci, e.g., $LEF1$ and $ASIP$, in the light-coated
indicine breeds (Fig. 2). This selection pressure may have been favored or driven by high
temperatures and intense solar radiation and/or human preferences. Across the whole genomes,
the CLR and iHS analyses revealed 368 regions overlapping with 477 genes present in AFI and
SAI (Supplementary Table 8), supporting that the ancestral adaptations of SAI cattle were
equally important for AFI cattle”

Moreover, the naming of the sections are also confusing, “the ancestral adaptation of indicine
cattle”, “Ancestral environmental adaptation of south Asian indicine cattle”, “the indicine
adaptation to the tropical, humid environment”. Ideally, signature of selection analysis should
be performed without a priori expectation because it is very difficult to conclusively link the
identified genes under selection with the breed characteristics; usually, it is done by functional
genomics and by studying the literature extensively. Therefore, such naming of the sections is
suitable to discussion but not to result section.

**Response:** *We agree and have renamed the sections.*

**Reviewer #2 (Remarks to the Author):**

**[Comment of Reviewer #2]**

Overall, the manuscript has improved significantly. I appreciate the author's willingness to
make changes and incorporate suggestions. There are three documents as I see it, the main
manuscript, the supplement and the response to previous review, all of which are lengthy. I have
comments for each, which I'll address individually, but the most pressing issue relates to the
ROH analysis under Supplementary Note 2: Genetic diversity. I'll start there because I think
this needs addressed.

Response L358-400 and supplement L273-280. I still do not believe your ROH analysis is
accurate. Even with your new analysis, you show a large number of individuals with a total
genome length contained within ROH to be >1 Gb, or over a third of the genome. That is simply
not realistic. There has been a large amount of prior literature on the subject, largely from SNP-
chip data, and all of them show the "extreme" samples having under 1 Gb in ROH. I believe
this [1] is an informative publication.
[1] <https://bmcbgenomdata.biomedcentral.com/articles/10.1186/1471-2156-13-70> I believe
Figure 5 from [1] is particularly useful in that it shows the same trend you are seeing in your
data but the scale is quite different. This is particularly important because if the current
manuscript is published it will present conflicting values because I believe you are including
far too much of the genome in ROH. It is generally accepted that these types of ROH analysis
are measuring recent inbreeding, which you acknowledge on L160 of the manuscript, where
the size of ROH regions is proportional to the inbreeding coefficient. As the size of a ROH
decreases you are estimating IBD further back in time. This may be appropriate in some cases,
even perhaps the current manuscript, but your ability to differentiate old IBD regions from
random chance or other population forces requires increased samples sizes, which you do not
have. As I said, on it's face, I do not believe your ROH results. The question is, what is causing
them to be elevated? In order to try to figure this out, I extracted 187 samples from your study
that are also present within my UMAG1 call set, which is based on ~5500 genomes and VQSR.
For these 187 samples, I extracted two sets of variants, A) all bi-allelic SNPS that passed VQSR
and B) the positions contained on the Illumina Bovine HD and GGPF250 genotyping assays in
order to compare with the results from [1].

326 A) "bcftools view --threads \$CPU -f PASS -m2 -M2 -v snps -S \$InputSamples --force-samples
-Ob -o \$c.\$Prefix.bcf \$InputBcf"

B) "bcftools view --threads \$CPU -R \$c.variants2get.list --trim-alt-alleles -Ob -o \$c.\$Prefix.bcf
\$InputBcf"

Dataset A produced 125M variants for these 187 samples.

bcftools +counts ChenAll.bcf

Number of samples: 187

Number of SNPs: 125813861

Number of INDELS: 0

Number of MNPs: 0

Number of others: 0

Number of sites: 125813861

Dataset B produced ~850K variants.

bcftools view -S chen_sample.list --force-samples ChenChip.bcf -Ou | bcftools +counts
Number of samples: 187
Number of SNPs: 850772
Number of INDELS: 64337
Number of MNPs: 0
Number of others: 0
Number of sites: 897036
The bcf files were then converted to plink bed/bim/bam:
plink --bcf ChenAll.bcf --const-fid 0 --chr-set 29 --allow-extra-chr --out ChenAll
plink --bcf ChenChip.bcf --const-fid 0 --chr-set 29 --allow-extra-chr --out ChenChip
Finally, the ROH analysis was performed using various parameter values.
The first run used the values specified on L279-280 of main paper and L389-396 of Response.
I will refer to this as “ChenParameters”.
plink --bfile ChenAll --chr-set 29 --chr 1-29 --homozyg --homozyg-gap 1000 --homozyg-kb
100 --homozyg-snp 50 --homozyg-window-het 3 --homozyg-window-snp 50 --homozyg-
window-threshold 0.05 --out ChenAll
The second run was the same as above but changed --homozyg-window-het 3 to 1. I refer to
this as Chen1het.
The third run was the same as above but changed --homozyg-window-het 3  1, --homozyg-
snp 50  200 --homozyg-window-snp 50  100 --homozyg-snp 50  200. I refer to this as
ChenStrict.
The loci from the SNP-chips were run the same as the second analysis above which used the
author’s parameters but changed --homozyg-window-het 3  1. I refer to this as Chip1het.
All of the results from the *.hom.indiv plink files were copied into Excel and visualized. See
file ChenROHsubmitted.xlsx which should be available with this review. It is clear from Figure
1 that I was able to recreate the trend that is shown in Supp. Figures 2 & 3 with my
“ChenParameters” in blue. As I contend, these parameters still include too many ROH and
encompass too much of the genome per individual. The “Chen1het” in orange only change --
homozyg-window-het from 3 to 1 which produces values that more closely resemble prior
literature and what we know about these populations. Therefore, I believe the root problem with
the analyses, as presented, is this parameter. In hindsight, this makes sense as this parameter is
meant to allow for genotyping error. By setting this parameter to 3 and requiring only 100 loci
within a window, I believe you are effectively saying that you have little confidence in your
genotype calls because you are allowing 3 errors (3% error rate) to still call a region as
homozygous. If you are confident in your genotype calls, and new mutations are negligible
(which I think is true here) then a value of 1 is more appropriate and in fact yields results closer
to expectations. In order to more fully explore this, I also created a more strict dataset
(ChenStrict) which required more loci while still maintaining 1 het position (grey). In order to
provide a direct comparison to snp-chip data I also present the same as “Chip1het” in red. As
you can see, the “Chip1het” is comparable to prior literature from using these assays. My final
interpretation of these data is that the parameters used in the revision are still not appropriate,
the main parameter to change is --homozyg-window-het and the correct values for the other
parameters are somewhere between what was used and the “strict” values that I used. This is a
fairly trivial analysis that the authors should perform and update the manuscript.

**Response:** Thank you for these valuable concerns and comments. We very much appreciate
 your efforts! We agree that the “strict” values (ChenStrict) of the ROH detection method are
 more accurate, so we used the “strict” value parameters outlined in your comments to detect
 the ROH in our cattle data. The final parameters were set to a minimum length of 100 kb, a
 scanning window size of 100 SNPs, a minimum density threshold of 200 SNPs, a large gap of
 1000 kb, a maximum number of heterozygous SNPs in the scanning window of 1, and a scanning
 window threshold level of 0.05. These settings yielded expected number (maximum number was
 3259) and total length (maximum length was 1,138,710 Mb) of ROH (Supplementary Fig. 2).
 Finally, we have updated Supplementary Note 2 and Supplementary Fig. 2.

 *Supplementary Fig. 2 Runs of homozygosity (ROH) patterns of all individuals from each cattle*
 *geographic group.*

**[Comment of Reviewer #2]**

As before, I will provide comments on the Supplement by referring to SL as supplement line
 number.

SL261-264: “The whole genome sequencing data from six other bovine species were mapped
 in the same way. We used the 67,162,108 SNPs as a reference list to genotype the combined set
 of 495 cattle samples and 22 samples of six other bovine species, resulting in 67,145,163 SNP
 data with wild species data.”

It is unclear to me what you mean by “...as a referencelist...”? My initial interpretation of this
 is that you added additional samples, called genotypes on those additional samples, and then

just extracted the 67M positions from the prior variant call set. Is this correct? The proper way
to perform this analysis would be to recall genotypes from *all* of your samples starting with
the combined GVCFs from SL253. This confusion is exacerbated by the third description in
Supp. Table 3 which says “A total of 64,475,272 SNPs called from all 495 cattle and 22 genomes
from other six bovine species.” Please clarify exactly what you did here.

*Response: Thank you for these valuable concerns and comments. We have rephrased this*
*sentence: “We genotyped the combined set of 495 cattle samples and 22 samples of six other*
*bovine species, and then extracted the 67,162,108 SNPs. After filtering out the non-biallelic*
*SNPs, 67,145,163 autosomal SNPs were obtained.”*

**[Comment of Reviewer #2:]**

SL266 Supplementary Table 3 This table is very informative and helpful to the reader and this
reviewer. However, now that I have this table, it presents new problems. For the second line of
this table where it says “Imputed data of...” I assume this represents the phasing and imputation
that you did with Beagle, correct? If so, please add the word “phased” to this description. This
is a very important point later.

*Response: Thank you for these valuable concerns and comments. We indeed used phased and*
*imputed and phased data. We revised the text as follows: “We used phased and imputed SNP*
*data”.*

None of these numbers seem to add up, going back to my comment directly above. It seems to
me that the proper way to analyze all of these data is to do joint genotyping on all samples
simultaneously, which would be your 514 samples presented in Supp. Table 1. (Note that the
second line of this table, and in the manuscript and supplement you refer to 495+22 which is
517 but you only have 514 in Table1.) This will result in a total number of variants N. All
analyses performed after this will involve some level of filtering or selection of loci but they all
will go back to this original N variants. As it stands, this table helped clear up some issues but
it presented more. For example the 6th description in the number of samples column, “All 495
cattle and one yak”. Table 1 lists 3 yak while SL237 says two yak. Table 1 has 5 banteng while
SL237 says eight. Figure 4 (F) of the main manuscript mentions Kouprey but I do not see any
in table 1 as Kouprey or Bos sauveli. Where did this sample come from? Again, many of these
numbers are not consistent.

*Response: Thank you for this suggestion. We have double-checked our tables.*

*We have 517 samples, including 495 cattle and 22 other bovine species samples. We have*
*revised the number of samples. For the 6th description, we selected only one yak for the outgroup,*
*but to ensure consistency, we reanalyzed the data using three yak for the outgroup. We have*
*eight banteng and five gaur in our dataset, and we selected four pure banteng and 2 gaur*
*samples for the introgression analysis. We apologize for our mistakes in Supplementary Table 1*
*and we updated it, and we also added the ancient kouprey and gayal information to*
*Supplementary Note 4.*

*“For the analysis of the introgressed region of BTA25 (0.21-0.26 Mb), we also used a gayal*
*sample and an ancient kouprey sample to detect its origin. The coverage of gayal and kouprey*
*was 17.32× and 1.4×, respectively. Due to the hybrid origin of gayal and low coverage of the*

*kouprey genome, we did not examine possible introgression of gayal and kouprey. The publicly*
*available sequences were downloaded from China National GeneBank (CNGB) and the SRA*
*with the following project accession numbers: CRX165997 (gayal, YD4) and PRJNA764746*
*(kouprey).”*

**[Comment of Reviewer #2:]**

SL285 and 300 The PCA and admixture analysis used -indep-pairwise 50 10 0.1 while the NJ
and ML phylogeny used --indep-pairwise 50 5 0.1. Why would you use different parameters
for two different analyses to achieve the same objective of minimizing LD between the variants
you use? By using different parameter sets you effectively make a direct comparison of these
two analyses impossible because you changed the loci that you are using.

***Response:** Thank you for this specific suggestion. This is a typo. We used -indep-pairwise 50*
*10 0.1 for PCA and ML phylogenetic tree reconstruction, and all 67,162,108 autosomal SNPs*
*were used to construct the NJ tree.*

**[Comment of Reviewer #2]**

SL312 “...XP-EHH score using selscan v.1.1 with default settings.” Selscan software did not
have the ability to use unphased data until v2.0. Therefore, I assume that you are relying on the
Beagle phased and imputed data for these analyses, is that correct? If so, this is why it is
important to add the word phased to Supp. Table 1 because without doing the phasing in Beagle
you could not have used selscan v.1.1.

***Response:** Thank you for this valuable suggestion. We used phased and imputed data in selscan*
*software, and we have added this information to Supplementary Table 1.*

**[Comment of Reviewer #2]**

SL329 “... using SweepFinder2.”

From the manual, it appears that this requires a recombination rate file. What did you use as the
recombination rate between loci? My assumption is that you assumed a constant recombination
rate of 1 cM/Mb since you do not actually have a recombination map. If that is the case, please
state that in the manuscript. This software also requires a “B-value” for each variant. Where did
you get these values from?

***Response:** Thank you for this valuable comment.*

*In the CLR analysis, we did not use the recombination rate and B-value parameters. In*
*SweepFinder2 software's instructions, there are five alternative methods for the selection scan,*
*and the “recombination rate” and “B-value” are not required for each method. We applied*
*“Scan for selective sweeps with pre-computed empirical spectrum” in SweepFinder2, and the*
*parameter was “./SweepFinder2 --lu GridFile FreqFile SpectFile OutFile”.*

**[Comment of Reviewer #2]**

SL354: Why were EAI grouped into a single population without looking into recent specific
breed introgression as the potential source of admixed genotypes?

***Response:** Thank you for this valuable comment. We analyzed the breed averages of length and*
*sum and added the D values. The EAI breeds have similar values.*

**[Comment of Reviewer #2]**

Often times the direction of admixture events is difficult to estimate. Is the signal of

introgression the result of the modern banteng/gaur samples having some amount of domestic
cattle introgression in their genomes?

*Response: This indeed may happen, but such domestic->domestic introgression would not be*
*detected by RFMix or by phylogenetic analysis.*

**[Comment of Reviewer #2]**

SL360-S365: The f3 data does not appear to be presented in Supplementary table 10, There
appears to be a numbering issue of tables in the supplementary data document, with a shift of
+1 after supplementary table 7.

*Response: Thank you for this specific suggestion. We added more details on the D statistic and*
*f3 results to Supplementary Tables 10 and 11, and the Supplementary Dataset1. We have double-*
*checked the numbers of Supplementary tables and figures.*

**[Comment of Reviewer #2]**

Main manuscript referring to ML as manuscript line number:

ML 76 "...was a direct consequence of..." That's a fairly strong claim. I think you evidence
consistent with a hypothesis but I question whether it represents a direct consequence.

*Response: Thank you for this valuable comment. We have rephrased the text.*

**[Comment of Reviewer #2]**

ML113 remove the word have.

*Response: Thank you for this valuable comment. We have edited the text accordingly.*

**[Comment of Reviewer #2]**

ML261 Supp. Figure 16 has multiple pages of figures with the same panel labels but only 1
legend. Please have a look at this as it appears it may have been an error when preparing the
files for submission.

*Response: Thank you for this specific suggestion. We used DensiTree software to merge and*
*visualize all 79 trees in one figure. Each line represents a tree, so this figure includes 79 trees.*

*Reference: Bouckaert, R.R. DensiTree: making sense of sets of phylogenetic trees.*
*Bioinformatics 26, 1372-1373 (2010).*

**[Comment of Reviewer #2]**

ML264-266 "To identify regions in the EAI genomes that were likely under selection, we used
a statistic to detect positively selected and introgressed genes (PSIGs)" Are these actually
detecting genes or regions? You use both terms in the same sentence.

*Response: Thank you for this specific comment. We rephrased this sentence: "We used the U20*
*statistic to identify frequently introgressed genes in the EAI genomes"*

**[Comment of Reviewer #2]**

ML351 Supp. Fig. 38 This is a much better figure than the previous version!

*Response: Thank you for this kind comment.*

**[Comment of Reviewer #2]**

ML512 "...gene transfer format (GTF) (GCF_002263795.1) file." This is an accession for the
genome assembly, it is NOT an accession for a gene set. I assume that you either used

annotations from NCBI or Ensembl, both of which have many different versions which are
regularly updated. You need to specify which annotations you used (NCBI or Ensembl) and
exactly which version of the annotation.

***Response:*** *Thank you for this comment. We have updated the accessions for the gene set.*

*“As source of annotation, we used the source Bos taurus Annotation Release 106*
*(GCF_002263795.1_ARS-UCD1.2_genomic.gtf) based on the NCBI assembly of*
*GCF_002263795.1.”*

REVIEWERS' COMMENTS

Reviewer #1 (Remarks to the Author):

The authors have improved the manuscript considerably and have taken all my comments into account.

Therefore, I have no additional comments.

Reviewer #2 (Remarks to the Author):

I have reviewed all of the material the authors provided. In my opinion, the authors have addressed the main issues that I have raised that needed to be addressed. At this point, there seem to still be some issues regarding methodology, but those are unlikely to be resolved as they are differences of opinion on how something should be done. Likewise, I believe there may still be some differences of opinion regarding interpretation of results. However, I see nothing that would preclude this manuscript from being published, and any perceived issues addressed through the normal scientific process.

Robert Schnabel